# Host CDK-1 and formin mediate microvillar effacement induced by enterohemorrhagic *Escherichia coli*

Cheng-Rung Huang[1,2], Cheng-Ju Kuo [1,2], Chih-Wen Huang[1], Yu-Ting Chen[1], Bang-Yu Liu[1], Chung-Ta Lee[3], Po-Lin Chen [4,5], Wen-Tsan Chang[1,2], Yun-Wen Chen[2,6], Tzer-Min Lee[7,8], Hui-Chen Hsieh[1,2] & Chang-Shi Chen [1,2 ✉]

Enterohemorrhagic *Escherichia coli* (EHEC) induces changes to the intestinal cell cytoskeleton and formation of attaching and effacing lesions, characterized by the effacement of microvilli and then formation of actin pedestals to which the bacteria are tightly attached. Here, we use a *Caenorhabditis elegans* model of EHEC infection to show that microvillar effacement is mediated by a signalling pathway including mitotic cyclin-dependent kinase 1 (CDK1) and diaphanous-related formin 1 (CYK1). Similar observations are also made using EHEC-infected human intestinal cells in vitro. Our results support the use of *C. elegans* as a host model for studying attaching and effacing lesions in vivo, and reveal that the CDK1-formin signal axis is necessary for EHEC-induced microvillar effacement.

[1] Department of Biochemistry and Molecular Biology, College of Medicine, National Cheng Kung University, Tainan, Taiwan. [2] Institute of Basic Medical Sciences, College of Medicine, National Cheng Kung University, Tainan, Taiwan. [3] Department of Pathology, College of Medicine, National Cheng Kung University, Tainan, Taiwan. [4] Department of Medicine, College of Medicine, National Cheng Kung University, Tainan, Taiwan. [5] Department of Microbiology and Immunology, College of Medicine, National Cheng Kung University, Tainan, Taiwan. [6] Department of Pharmacology, College of Medicine, National Cheng Kung University, Tainan, Taiwan. [7] School of Dentistry, Kaohsiung Medical University, Kaohsiung, Taiwan. [8] Institute of Oral Medicine, College of Medicine, National Cheng Kung University, Tainan, Taiwan. ✉email: cschen@mail.ncku.edu.tw

Enterohemorrhagic *Escherichia coli* (EHEC) can cause severe diarrhea, hemorrhagic colitis (HC), and even hemolytic uremic syndrome in humans[1]. Three major groups of virulence determinants of EHEC have been identified, including the products of the virulence plasmid pO157, the products of the locus of enterocyte effacement (LEE), and the Shiga-like toxins (Stx) encoded by the temperate bacteriophages in chromosome[2]. Currently, there is no vaccine or targeted therapy for EHEC infection and the use of antibiotics is controversial[3]. Thus, understanding the pathogenesis of EHEC is necessary to move forward in developing novel therapeutics.

As a human attaching and effacing (A/E) pathogen, EHEC induces notable morphological changes in the intestinal epithelium, leading to the formation of A/E lesions[4]. A/E lesions are characterized by the effacement of intestinal microvilli firstly and then following the rearrangement of the host actin cytoskeleton to form pedestal-like structures for the attachment of EHEC. The ability to generate actin pedestals depends on the translocation of bacterial effectors into host enterocytes by a type III secretion system (T3SS) encoded in the LEE pathogenicity island. The T3SS effectors Tir, EspF, and EspH recruit host N-WASP and actin-related protein 2/3 (Arp2/3) to orchestrate actin polymerization beneath the contact point to form an intracellular pedestal, allowing tight attachment of the extracellular bacterium[4]. However, compared with the mechanism of the EHEC-induced attachment, which has been thoroughly studied over the past two decades, the underlying molecular mechanism of the EHEC-induced microvillar effacement remains largely unknown.

Given the lack of mouse models for studying A/E lesion induced by EHEC[5], we and others have applied the model organism *Caenorhabditis elegans* to study the infection of EHEC in vivo[6,7]. Our previous studies demonstrated that EHEC causes redistribution of ACT-5, a microvillus-specific actin, subcellular localization from the apical site to the basolateral cytoplasm in their intestinal cells (as shown in Fig. 1) and concomitant A/E lesion formation in the alimentary tract in *C. elegans*[7,8]. Five actin isoforms are encoded by *C. elegans*; however, only ACT-5 is essential for the stable morphogenesis of intestinal microvilli[9]. The microvillar ACT-5 proteins form a core bundle structure to support the normal morphology of the intestinal microvillus, and core microvillar components are in constant turnover, making microvilli highly dynamic structures[10]. Thus, we thought this EHEC-induced mislocalization of microvillar ACT-5 might be a biomarker for studying the microvillar effacement in *C. elegans*. Given the complexity and inherent dynamic nature of microvilli, the EHEC-induced microvillar effacement may likely involve yet unidentified cellular processes, including the key actin nucleators in a similar manner as N-WASP and Arp2/3 that are employed in the pedestal formation for EHEC attachment.

Three major classes of actin nucleators have been identified: the Arp2/3 complex, the formins, and the tandem actin-binding domain nucleators, such as Spire, Cobl, and leiomodin[11]. Formin, defined by the presence of a catalytic formin homology 2 (FH2) domain (see below), is a processive motor that requires profilin to accelerate elongation of the barbed and fast-growing ends of actin filaments[12,13]. Several reports have suggested that formin can be phosphorylated and regulated by the mitotic cyclin-dependent kinase 1 (CDK-1)[14–16]. As aforementioned, the A/E pathogen EHEC has developed strategies to subvert the host Arp2/3 complex and rearrange the actin cytoskeleton to form a pedestal for its attachment. However, whether EHEC can also hijack the same or the other actin nucleators, such as formin, to induce host microvillar effacement remains unknown.

Here, we show that the mitotic cyclin B3/CYB-3 is an important host factor activated by EHEC and is required for this EHEC-induced ACT-5 mislocalization in *C. elegans*. We also found that the CDK-1 activity is indispensable for this EHEC-mediated ACT-5 mislocalization. Moreover, we identified that the diaphanous-related formin 1/CYK-1 and profilin/PFN-1 act downstream of the CYB-3/CDK-1 signal to sequester ACT-5 in the basolateral cytoplasm, thus leading to the deformation of the microvillus structure induced by EHEC in *C. elegans*. In a human intestinal cell model, we also reconfirmed that cyclin B3/CCNB3 and CDK-1 regulates the diaphanous-related formin 1/DIAPH1 expression and are involved in microvillar effacement post-EHEC infection. In sum, our results not only reveal that the role of host CDK-1-formin signal for EHEC-induced microvillar effacement, but also suggests the potential for targeting this signal circuit as a therapeutic measure for EHEC infection.

## Results

**EHEC induces mislocalization of microvillus-specific actin and microvillar effacement in *C. elegans*.** Our previous study demonstrated that the EHEC strain EDL933 induces mislocalization of the intestinal microvillus-specific actin, ACT-5, in *C. elegans*[7]. In order to further characterize this EHEC-induced microvillar actin rearrangement in vivo, we used transgenic *C. elegans* expressing mCherry-tagged ACT-5 (mC::ACT-5)[17]. The microvillus-specific ACT-5 proteins are expressed exclusively in microvilli at the apical side of the intestinal cells[9] (Fig. 1a). In the *mC::ACT-5* worms fed with the non-pathogenic *E. coli* strain OP50 for 4 days, the mCherry signals were expressed exclusively at the apical side of the intestinal cells (see below). However, intriguingly we found that in the EHEC-infected animals (4 days post infection), the expression pattern of the mC::ACT-5 signals was abnormal, and ACT-5 signals redistributed their subcellular localization from the apical site and formed multiple punctae in the basolateral cytosol of intestinal cells in EHEC-infected animals (Fig. 1b). We thus gave each EHEC-infected animal with an arbitrary score from 0 to 3, and 3 showed the most severe phenotype, and we considered animals with a score of 2 or 3 as positive for the mC::ACT-5 mislocalization (Fig. 1b). The quantitative results are shown in Fig. 1c. These data suggested that the normal *C. elegans* microvilli morphology may be altered by EHEC. Moreover, in order to minimize the effect of the tagged fluorescent protein and the position and copy number effects of the *act-5* transgene. We have tested the EHEC-induced ACT-5 mislocalization by two additional independent transgenic animals, namely *GFP::ACT-5* and *YFP::ACT-5* (Supplementary Fig. S1a, b and, quantification in Supplementary S1c). The expression levels of these transgenes have also been examined (Supplementary Fig. S1d). Taken together, our results showed that EHEC can induce similar ACT-5 mislocalization phenotype in these three independent transgenic animals with different expression levels, suggesting the redistribution of ACT-5 subcellular localization induced by EHEC is not due to the effect of the tagged fluorescent proteins nor the position and copy number effects of the *act-5* transgenes in the *C. elegans* genome.

As mentioned above, EHEC can remodel the host actin cytoskeleton by T3SS and its injected effectors. In order to test whether this ACT-5 mislocalization phenotype is due to the alteration of intestinal cell polarity by EHEC, we used an apical membrane marker, PGP-1::GFP[18], to examine the polarity of intestinal cells. In contrast to in the EHEC-infected *mC::ACT-5* animals, we found that PGP-1::GFP signals were restricted to the apical side of intestinal cell in the *PGP-1::GFP* animals fed with OP50 or EHEC for 4 days (Fig. 1d). The quantitative results are shown in Fig. 1e. Moreover, the signals of another intestinal apical site marker, the intermediate filament protein IFB-2::CFP[19], were also limited to the apical side of intestinal cells in both the OP50 and EHEC-treated groups (images in Fig. 1d and

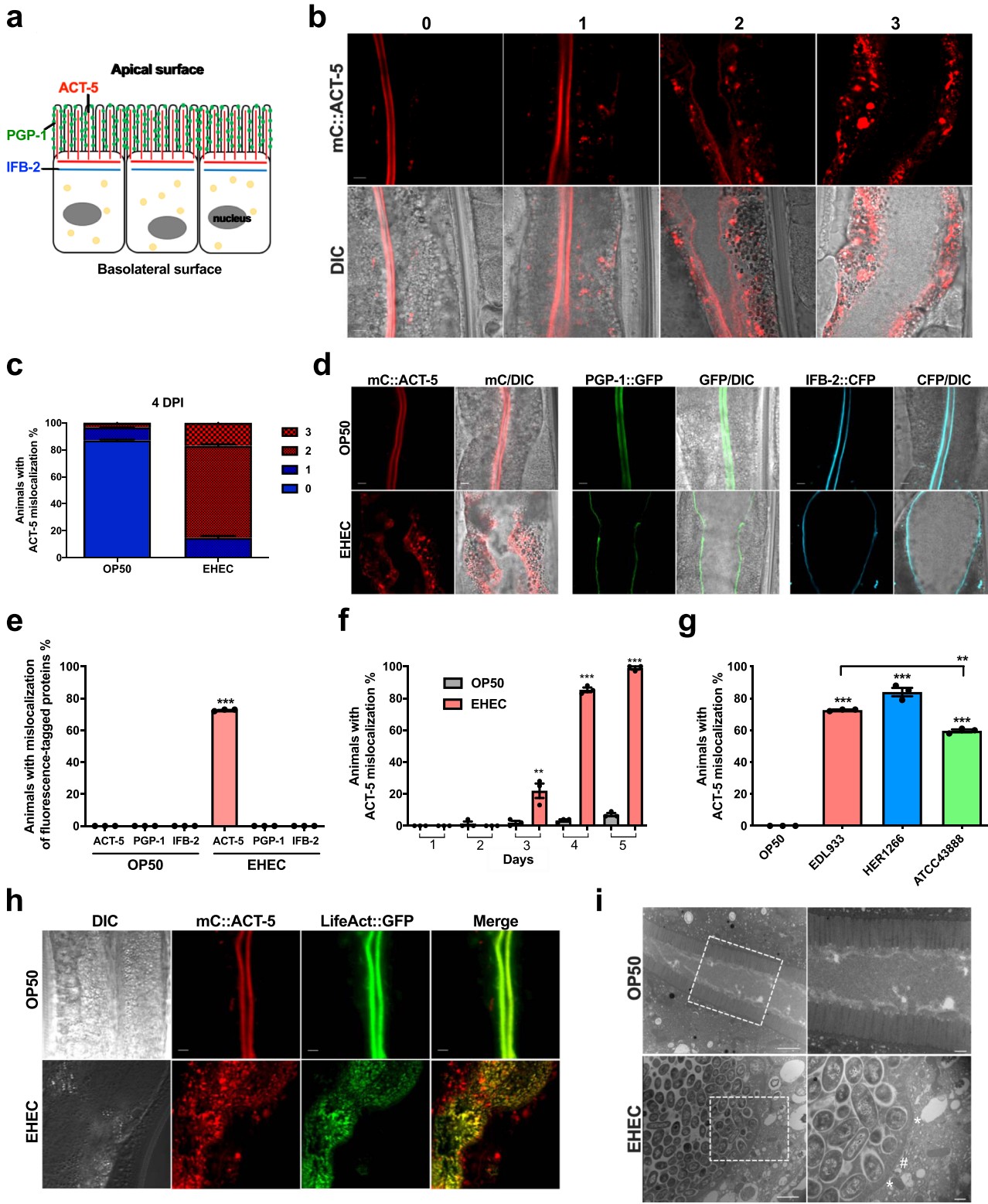

quantifications in Fig. 1e). Taken together, these results suggest that EHEC infection specifically induces relocalization of the intestinal microvillar ACT-5, but does not alter the polarity of intestinal cells in *C. elegans*.

To further investigate this EHEC-induced redistribution of ACT-5, we quantified the ratio of *mC::ACT-5* animals with this ACT-5 mislocalization phenotype over a period of time (Fig. 1f). Animals fed with OP50 exhibited a low percentage of ACT-5 mislocalization from day 1 to day 5. However, the percentages of

ACT-5 mislocalization in EHEC-infected worms were significantly increased from the third day and continued increase on the fourth and fifth days compared to OP50 control groups.

In order to determine whether this ACT-5 mislocalization phenotype is EHEC strain-specific, we also examined the *mC:: ACT-5* worms infected with the other two EHEC clinical isolates, HER1266 and ATCC43888. As shown in Fig. 1g, animals infected with these three different EHEC strains all exhibited a significantly increased percentage of ACT-5 mislocalization compared to the

**Fig. 1 EHEC induces rearrangement of microvillus-specific actin in *C. elegans*. a** A graphic representation of the anatomy of *C. elegans* intestinal cells and three proteins on the apical side, including the apical membrane transporter protein, PGP-1 (green), intermediate filaments, IFB-2 (blue), and microvillar actin, ACT-5 (red). **b** Representative confocal images of the four classes of EHEC-induced mC::ACT-5 mislocalization. Scale bars represent 5 μm. The quantitative results are shown in panel (**c**). **c** The quantitative result for the percentage of animals with EHEC-induced mC::ACT-5 mislocalizatoin in panel (**b**); $n = 97$, $N = 3$ in the OP50 group, and $n = 113$, $N = 3$ in the EHEC group. **d** Confocal images of the apical proteins, including mC::ACT-5, PGP-1:: GFP, and IFB-2::CFP, in the *mC::ACT-5*, *PGP-1::GFP*, and *IFB-2::CFP* animals, respectively, fed with OP50 or EHEC at 20 °C for 4 days are shown. Merged images (Merge) are the differential interference contrast (DIC) images overlaid with fluorescent images. The scale bars represent 5 μm. The quantitative results are shown in panel (**e**). **e** The quantitative result for the ectopic localization of fluorescent proteins in the *mC::ACT-5* ($n = 96$, $N = 3$), *PGP-1::GFP* ($n = 94$, $N = 3$), and *IFB-2::CFP* ($n = 93$, $N = 3$) worms fed with OP50, or in the *mC::ACT-5* ($n = 89$, $N = 3$, $P < 0.0001$), *PGP-1::GFP* ($n = 100$, $N = 3$, $P > 0.9999$), and *IFB-2::CFP* ($n = 91$, $N = 3$, $P > 0.9999$) worms fed with EHEC at 20 °C for 4 days in panel (**d**). Asterisks indicate statistically significant differences as compared to the OP50 control group (***$P < 0.001$). **f** The percentage of worms with mislocalization of *mC::ACT-5* after exposure to OP50 (1 day: $n = 134$, $N = 3$; 2 days: n = 134, $N = 3$; 3 days: $n = 129$, $N = 3$; 4 days: $n = 129$, $N = 3$; 5 days: $n = 118$, $N = 3$) or EHEC (1 day: $n = 137$, $N = 3$, $P > 0.9999$; 2 days: $n = 137$, $N = 3$, $P = 0.3739$; 3 days: $n = 125$, $N = 3$, $P = 0.0067$; 4 days: $n = 120$, $N = 3$, $P < 0.0001$; 5 days: $n = 95$, $N = 3$, $P < 0.0001$) at 20 °C for 5 days. Asterisks indicate statistically significant differences as compared to OP50 control group (**$P < 0.01$ and ***$P < 0.001$). **g** The percentage of animals with mC::ACT-5 mislocalization feeding on OP50 ($n = 141$, $N = 3$) and three different EHEC clinical isolates, including EDL933 ($n = 130$, $N = 3$, $P < 0.0001$), HER1266 ($n = 124$, $N = 3$, $P < 0.0001$), and ATCC43888 ($n = 123$, $N = 3$, $P < 0.0001$) for 4 days were examined. Asterisks indicate statistically significant differences as compared to OP50 control group (**$P < 0.01$ and ***$P < 0.001$). Moreover, ATCC43888 showed a significantly decreased percentage of ACT-5 mislocalization compared to EDL933 group ($P = 0.0002$). **h** The *mC::ACT-5; app-1p::LifeAct::GFP* animals were used to monitor F-actin in the intestinal cells. The representative confocal images are shown. Scale bars represent 5 μm. The experiment was performed independently three times. **i** The representative TEM images of N2 animals fed with OP50 or EHEC for 4 days at 20 °C. Enlarged images (right panels) are from the frame on the left panel images. The intestinal microvilli of EHEC-infected worms were much shorter as indicated by the hash mark (#). Moreover, the effacements of microvilli in the EHEC-treated animals are indicated by the asterisks (*). Scale bars in the left panels represent 2 μm. Scale bars in the right panels represent 0.5 μm. The experiment was performed independently three times. All quantitative data are presented as mean ± SEM, and each dot represented an independent result in the bar chart. All data statistics based on: **$P < 0.01$ and ***$P < 0.001$ by unpaired *t* test (two-tailed). Source data are provided as a Source Data file.

OP50 control. Of note, worms infected with the EHEC strain ATCC43888, an EHEC strain without the expression of Stx, showed a significantly decreased percentage of ACT-5 mislocalization compared to that of EDL933, which implied that the EHEC bacterial factor, Stx, might play a role in the induction of ACT-5 mislocalization.

The EHEC-induced mC::ACT-5 signals formed multiple punctae in the cytosol. We next investigated whether these mislocalized ACT-5 signals in the basolateral cytosol are F-actins in intestinal cells. LifeAct is a 17-amino-acid peptide that can bind F-actin without affecting actin dynamics in vivo[20]. We therefore generated a *LifeAct::GFP C. elegans* transgenic strain to monitor F-actin in the intestinal cells. As shown in Fig. 1h, the LifeAct::GFP signals completely colocalized with mC::ACT-5 signals at the apical site of the intestinal cells in the *mC:: ACT-5;LifeAct::GFP* worms fed with OP50, which reconfirmed that the ACT-5 bundles in the microvilli are F-actin structures. In the EHEC-treated worms, the mC::ACT-5 signals formed multiple punctae at the basolateral site and most of them were colocalized with LifeAct::GFP signals, which demonstrated that the mislocalized ACT-5 proteins are, at least in part, F-actins. However, we cannot exclude the possibility that some of the mislocalized ACT-5 are simply mC:ACT-5 protein aggregates. Nevertheless, our data also suggested that an, as yet unidentified, actin nucleator exists in the basolateral cytosol to polymerize these microvillus-specific ACT-5 proteins.

In order to more closely monitor the alteration of microvilli morphology induced by EHEC, we performed transmission electron microscopy (TEM) analysis to examine the microvilli morphology in EHEC-infected *C. elegans*. N2 WT animals fed with OP50 exhibited a complete microvillar brush border (Fig. 1i). However, animals infected with EHEC showed bloating of the intestinal lumen and accumulation of bacterial cells. Furthermore, the intestinal microvilli of EHEC-infected worms were much shorter, and effacement of microvilli could be observed. Taken together, our results demonstrated that EHEC causes mislocalization of microvillus-specific actin and concomitant microvillar effacement in *C. elegans*.

**cyb-3 is required for the EHEC-induced ACT-5 mislocalization in *C. elegans*.** In order to identify the host factors required for the EHEC-induced microvillar actin rearrangement, we performed an RNA-sequencing (RNA-seq) experiment to analyze the transcriptomic changes in the EHEC-infected *C. elegans*. We found that the messenger RNA (mRNA) levels of 97 genes in N2 WT animals were significantly upregulated and mRNA levels of 129 genes were significantly downregulated after EHEC infection compared to the OP50 control (Supplementary Data 1). We hypothesized that EHEC may induce the expression of some host factors that are required for the redistribution of microvillar ACT-5. Therefore, we subjected *mC::ACT-5* worms to RNA interference (RNAi) of those 97 upregulated genes and examined the ACT-5 mislocalization phenotype after EHEC infection for 4 days (Supplementary Fig. S1e and Supplementary Data 2). Our focused RNAi screen showed that silencing ten of these genes significantly reduced the percentages of ACT-5 mislocalization compared to the L4440 RNAi control (Fig. 2a and Supplementary S1e). Among them, knockdown of *cyb-3* by RNAi exhibited the lowest percentage of ACT-5 mislocalization. We, therefore, turned our focus to identify the mechanism of action of *cyb-3* in the EHEC-induced ACT-5 mislocalization in *C. elegans*.

First, we examined the mRNA expression level of *cyb-3* after EHEC infection by quantitative real-time polymerase chain reaction (qRT-PCR). We found that the *cyb-3* mRNA of N2 animals infected with EHEC was significantly increased compared to the OP50 control (Fig. 2b), which reconfirmed our RNA-seq results. We next imaged the *mC::ACT-5* animals with knockdown of *cyb-3* by RNAi and fed with OP50 or EHEC for 4 days. The mC::ACT-5 signals in animals subjected to L4440 control and *cyb-3* RNAi were all limited to the apical side of intestinal cells in the OP50 group (Fig. 2c). However, as shown in Fig. 2d, while L4440 control animals exhibited mislocalization of ACT-5 signals after EHEC infection, the ACT-5 signals were restricted at the apical site of intestinal cells in the *cyb-3* RNAi animals. The quantification results are shown in Fig. 2g. In order to confirm the role of *cyb-3* in the EHEC-induced ACT-5 mislocalization, we introduced the *cyb-3* overexpression transgene (*cyb-3 o/e*, from the

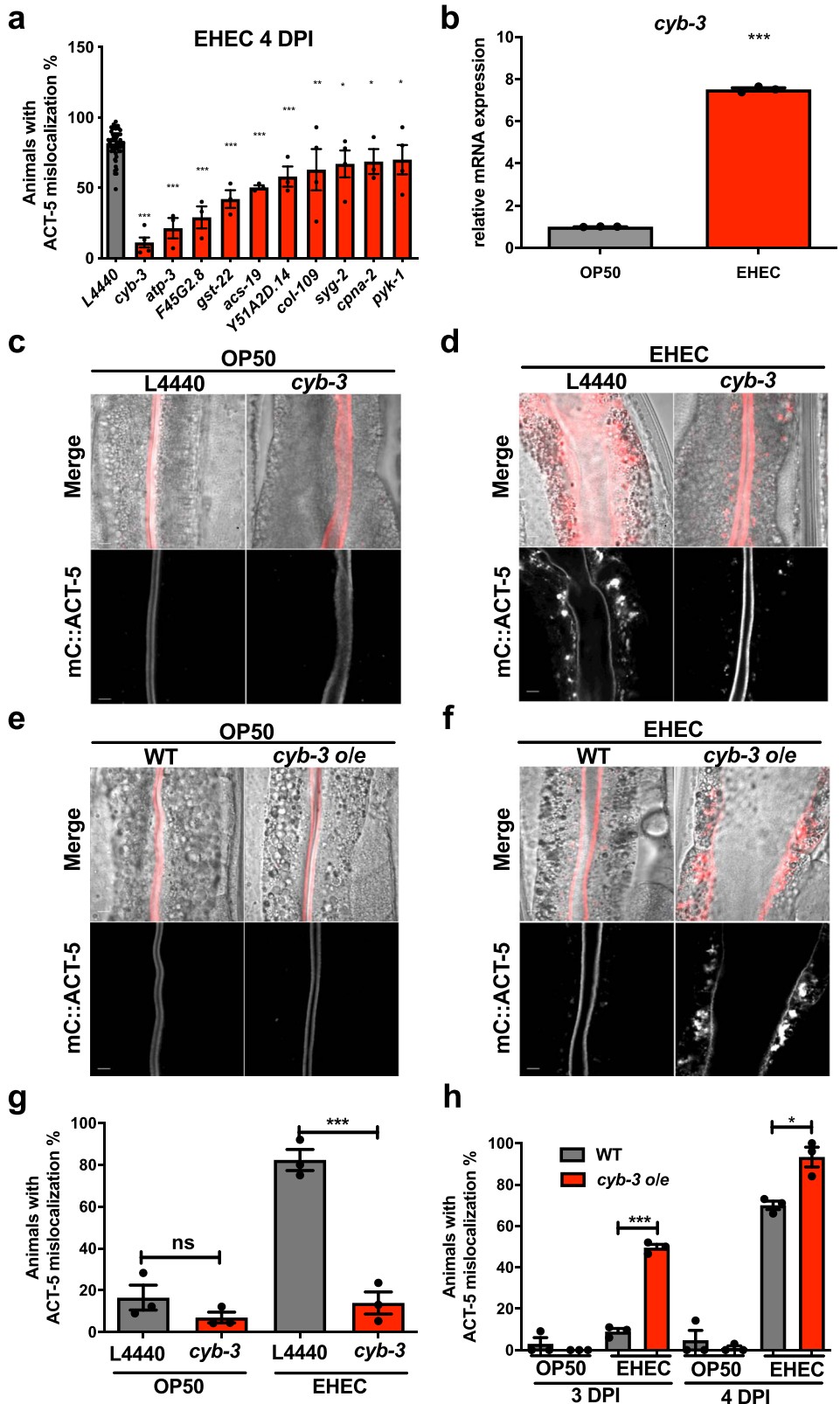

strain JNC100[21]) into the *mC::ACT-5* background. As shown in Fig. 2e, we did not observe ACT-5 mislocalization in the *cyb-3 o/e* animals fed with OP50 for 3 days. However, *cyb-3 o/e* animals infected with EHEC for 3 days exhibited early onset of the ACT-5 mislocalization phenotype compared to the WT worms (Fig. 2f). The quantification results are shown in Fig. 2h. *cyb-3 o/e* worms fed with EHEC for 3 or 4 days all showed a significant increase of

ACT-5 mislocalization compared to WT animals. Taken together, these data demonstrated that the host factor *cyb-3* is required for the EHEC-induced microvillar ACT-5 redistribution in *C. elegans*.

**CDK-1 activity is indispensable for the EHEC-induced microvillar actin rearrangement.** CYB-3 is a member of the cyclin B family, and CYB-3 binds to and activates the CDK-1 to

**Fig. 2 *C. elegans* cyb-3 is required for the ACT-5 mislocalization induced by EHEC. a** RNAi of *cyb-3* ($n = 198$, $N = 5$, $P < 0.0001$), *atp-3* ($n = 69$, $N = 3$, $P < 0.0001$), *F45G2.8* ($n = 77$, $N = 3$, $P < 0.0001$), *gst-22* ($n = 100$, $N = 3$, $P < 0.0001$), *acs-19* ($n = 144$, $N = 3$, $P < 0.0001$), *Y51A2D.14* ($n = 138$, $N = 3$, $P = 0.0002$), *col-109* ($n = 155$, $N = 4$, $P = 0.0032$), *syg-2* ($n = 129$, $N = 4$, $P = 0.0103$), *cpna-2* ($n = 137$, $N = 3$, $P = 0.0371$), and *pyk-1*($n = 179$, $N = 4$, $P = 0.0423$) suppressed the EHEC-induced ACT-5 mislocalization compared to empty vector L4440 ($n = 1790$, $N = 58$) on the fourth day post EHEC infection. Asterisks indicate statistically significant differences as compared to L4440 control group (*$P < 0.05$, **$P < 0.01$, and ***$P < 0.001$). **b** The qRT-PCR analysis of *cyb-3* mRNA in N2 wild-type animals exposed to OP50 ($N = 3$) or EHEC ($N = 3$, $P < 0.0001$) at 20 °C for 4 days. Asterisks indicate statistically significant differences as compared to OP50 control group (***$P < 0.001$). **c** The representative confocal images of *mC::ACT-5* worms fed with OP50 mixed with L4440 or *cyb-3* RNAi for 4 days. The merged images (Merge) are the DIC images overlaid with mCherry fluorescent images. The scale bar represents 5 µm. **d** The representative confocal images of *mC::ACT-5* worms infected with EHEC mixed with L4440 or *cyb-3* RNAi at 20 °C for 4 days. The merged images (Merge) are the DIC images overlaid with mCherry fluorescent images. The scale bar represents 5 µm. **e** The representative confocal images of wild-type (WT) and *cyb-3* overexpression (*cyb-3 o/e*) animals fed with OP50 at 20 °C for 3 days. The merge images (Merge) are the DIC images overlaid with mCherry fluorescent images. The scale bar represents 5 µm. **f** The representative confocal images of wild-type (WT) and *cyb-3 o/e* animals infected with EHEC at 20 °C for 3 days. The merged images (Merge) are the DIC images overlaid with mCherry fluorescent images. The scale bar represents 5 µm. **g** The quantification results of experiment in panels (**c**, **d**). OP50-L4440 group: $n = 151$, $N = 3$; OP50-*cyb-3* RANi: $n = 125$, $N = 3$, $P = 0.2166$; EHEC-L4440: $n = 134$, $N = 3$; EHEC-*cyb-3* RANi: $n = 95$, $N = 3$, $P = 0.0007$. Asterisks indicate statistically significant differences as compared to L4440 control group (***$P < 0.001$ and n.s., no significant difference). **h** The quantification results of ACT-5 mislocalization in wild-type (WT) and *cyb-3 o/e* animals exposed to OP50 or EHEC for 3 and 4 days. 3 DPI (OP50-WT animals: $n = 119$, $N = 3$; OP50-*cyb-3 o/e* animals: $n = 125$, $N = 3$, $P = 0.3739$; EHEC-WT animals: $n = 125$, $N = 3$; EHEC-*cyb-3 o/e* animals: $n = 129$, $N = 3$, $P < 0.0001$); 4 DPI (OP50-WT animals: $n = 117$, $N = 3$; OP50-*cyb-3 o/e* animals: $n = 125$, $N = 3$, $P = 0.4826$; EHEC-WT animals: $n = 118$, $N = 3$; EHEC-*cyb-3 o/e* animals: $n = 123$, $N = 3$, $P = 0.0112$). Asterisks indicate statistically significant differences as compared to WT group (*$P < 0.05$ and ***$P < 0.001$). All quantitative data are presented as mean ± SEM, and each dot represented an independent result in the bar chart. All data statistics based on: *$P < 0.05$, **$P < 0.01$, and ***$P < 0.001$ by unpaired *t* test (two-tailed). Source data are provided as a Source Data file.

regulate mitosis of the cell cycle[22,23]. We next aimed to examine whether the other cyclins and CDKs in the cell cycle are also involved in the EHEC-induced ACT-5 mislocalization in *C. elegans*. First, we subjected *mC::ACT-5* worms to RNAi of the G2/M-phase-specific CDK and cyclin genes, including *cdk-1*, *cyb-3*, and *cyb-1*, and treated them with OP50 or EHEC for 4 days (Fig. 3a, b). We found that knockdown of *cdk-1* also significantly suppressed the ACT-5 mislocalization in EHEC-infected worms compared to the L4440 control (Fig. 3a). The quantitative results are shown in Fig. 3b. Moreover, knockdown of the M-phase cyclins, including *cyb-3*/cyclin B3 and *cyb-1*/cyclin B1, all significantly suppressed the EHEC-induced ACT-5 mislocalization. These data suggested the mitotic CDK-1 activity, either activated by CYB-3 or CYB-1, is involved in the redistribution of ACT-5 in response to EHEC. However, we did not observe a significant difference in EHEC-induced ACT-5 mislocalization in *mC::ACT-5* animals subjected to specific RNAi for the CDKs or cyclins genes of G1, S, and G2 phases of the cell cycle compared to the L4440 control (Supplementary Fig. S2a–h). Taken together, our data suggested that only the mitotic CDK-1 activity is involved in the EHEC-induced ACT-5 mislocalization in *C. elegans*.

In order to reconfirm the role of *cdk-1*, we also utilized a *cdk-1* loss-of-function allele, *cdk-1(ne2257)*, and crossed it into the *mC::ACT-5* background. As shown in Fig. 3c, d, the *cdk-1(ne2257)* mutant animals exhibited a significantly lower percentage of ACT-5 mislocalization induced by EHEC compared to the WT animals. Given that both CYB-1 and CYB-3 can activate CDK-1, to test whether the *cyb-1* and *cyb-3* are interchangeable in the EHEC-induced ACT-5 mislocalization, we fed WT and *cyb-3 o/e* animals with *cyb-1* RNAi and EHEC for 3 days. Our results showed that *cyb-1* RNAi not only can significantly suppress the EHEC-induced ACT-5 mislocalization in WT animals (in agreement with Fig. 3b) but also inhibit the ACT-5 mislocalization in the *cyb-3 o/e* animals (Fig. 3e). These data demonstrated the non-redundant roles of *cyb-1* and *cyb-3* in the EHEC-induced ACT-5 mislocalization.

We then investigated whether the kinase activity of CDK-1 is crucial for the EHEC-induced ACT-5 mislocalization. When the RNAi of *cdc-25.1* and *cdc-25.2*, the endogenous activators of CDK-1[22], was used to suppress the kinase activity of CDK-1, animals showed significantly reduced percentages of the EHEC-induced

ACT-5 mislocalization compared to L4440 control (Fig. 3f). Moreover, knockdown of the endogenous CDK-1 inhibitors, including *cki-1*, *cki-2*, *cks-1*, and *wee-1.1*, induced early onset of the ACT-5 mislocalization phenotype in EHEC-infected animals on the third day compared to L4440 control (Fig. 3g). In order to reconfirm the notion that the enzyme activity of CDK-1 is indispensable for this EHEC-induced microvillar actin rearrangement, we used a specific CDK-1 inhibitor, RO3306, to directly inhibit the CDK-1 kinase activity in *C. elegans*. As shown in Fig. 3h, i, EHEC can significantly induce phospho-SUN-1$^{ser43}$ level, which is CDK-1-dependent[24,25], and RO3306 significantly inhibited the phosphorylation of SUN-1at Serine43 in worms. Moreover, the *mC::ACT-5* nematodes treated with RO3306 exhibited a decreased percentage of ACT-5 mislocalization compared to the untreated control worms (Fig. 3j). Taken together, our results suggested that CDK-1 kinase activity is required for the EHEC-induced ACT-5 mislocalization in *C. elegans*.

**CYK-1 and PFN-1 act downstream of the CYB-3/CDK-1 signal in the EHEC-induced microvillar actin rearrangement.** In order to identify the actin nucleator that acts downstream of the CYB-3/CDK-1 signal and is involved in the EHEC-induced microvillar ACT-5 polymerization in the basolateral cytosol, we performed another focused RNAi screen against various actin modifiers in the *mC::ACT-5* animals fed with OP50 or EHEC for 4 days (Fig. 4a). The results showed that knockdowns of the *arx-2*, *ani-1*, *cap-1*, and *cap-2* by RNAi confer significantly higher ACT-5 mislocalization in the *mC::ACT-5* animals fed with non-pathogenic OP50. These results reconfirmed that the *arx-2* (encoding the ortholog of Arp2 in *C. elegans*), *ani-1* (encoding the ortholog of anillin), *cap-1* (encoding the ortholog of capping actin protein subunit alpha 1), and *cap-2* (encoding the ortholog of capping actin protein subunit beta/CAPZB) are involved in the formation of normal microvillar actin bundles in the intestinal cells in *C. elegans*. Moreover, RNAi of the *cyk-1* and *pfn-1* both significantly reduced the ratios of ACT-5 mislocalization in *mC::ACT-5* animals fed with EHEC compared to the L4440 control. *cyk-1* encodes the ortholog of human diaphanous-related formin 1/DIAPH1 in *C. elegans*, and *pfn-1* encodes an actin-binding profilin, which accelerates the actin polymerization activity of CYK-1[26].

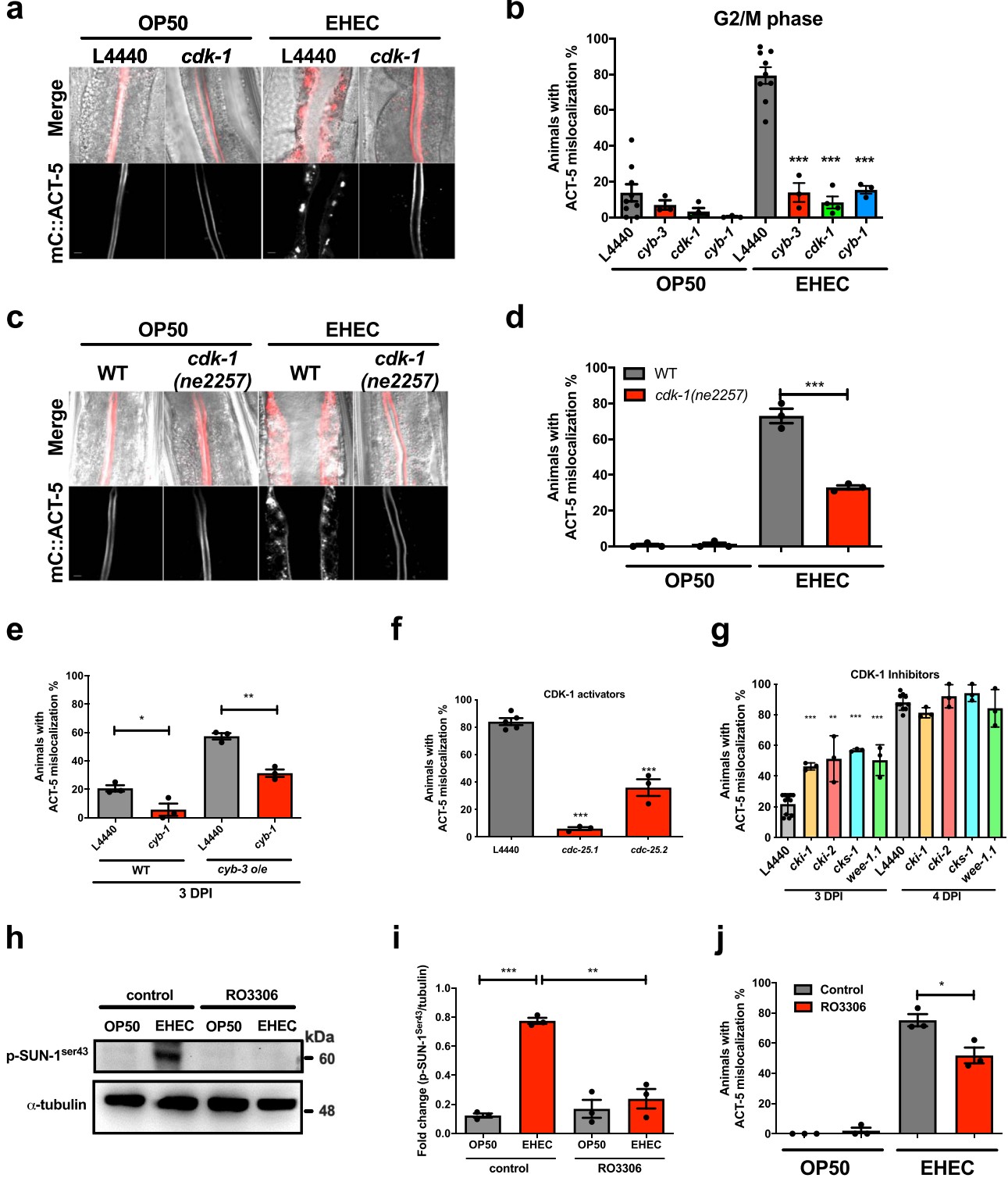

In order to reconfirm that this specific formin/profilin pair is required for the EHEC-induced microvillar effacement, we first treated the WT and *cyk-1* overexpression (*cyk-1 o/e*) animals with OP50 or EHEC for 3 and 4 days, respectively (Fig. 4b). The *cyk-1 o/e* animals infected with EHEC for 3 days exhibited significant early onset of the ACT-5 mislocalization phenotype compared to WT, and the percentage of ACT-5 mislocalization was also significantly increased in the *cyk-1 o/e* animals fed with EHEC on the fourth day. Moreover, the *mC::ACT-5* animals with the *pfn-1(ok808)* mutant background infected with EHEC for 4 days also exhibited a significantly lower percentage of the ACT-5 mislocalization compared to the WT (Fig. 4c). Taken together, our results reconfirmed that *cyk-1* and *pfn-1* are involved in the EHEC-induced ACT-5 mislocalization.

To test whether this specific formin/profilin pair acts downstream of the CYB-3/CDK-1 signal, we performed a genetic epistasis analysis in the *mC::ACT-5* animals with WT or *cyb-3 o/e* backgrounds (Fig. 4d, e). Our results showed that knockdown of *cyk-1* or *pfn-1* by RNAi in the *cyb-3 o/e* animals significantly suppressed the early onset of the EHEC-induced microvillar

**Fig. 3 CDK-1 activity is required for the EHEC-induced ACT-5 mislocalization. a** The representative confocal images of *mC::ACT-5* worms after RNAi of *cdk-1* and RNAi control L4440 and then fed with OP50 or EHEC for 4 days. The merged images (Merge) are the DIC images overlaid with the mCherry fluorescent images. The scale bars represent 5 μm. The experiment was performed independently for three times. **b** Knockdown of the G2/M-phase genes suppressed the EHEC-induced ACT-5 mislocalization in *C. elegans*. *mC::ACT-5* animals subjected to RNAi of the G2/M-phase-related genes, including *cyb-1* ($n = 116$, $N = 3$, $P < 0.0001$), *cyb-3* ($n = 135$, $N = 3$, $P < 0.0001$), and *cdk-1* ($n = 207$, $N = 4$, $P < 0.0001$), reduced the ACT-5 mislocalization in EHEC infection group compared to the L4440 control ($n = 453$, $N = 9$). **c** The representative confocal images of wild-type and *cdk-1(ne2257)* worms fed with OP50 or EHEC for 4 days. The merged images (Merge) are the DIC images overlaid with the mCherry fluorescent images. The scale bars represent 5 μm. The experiment was performed independently for three times. **d** The percentage of ACT-5 mislocalization of wild-type (WT) and *cdk-1(ne2257)* animals fed with OP50 or EHEC at 20 °C for 4 days were examined. WT-OP50 group: $n = 121$, $N = 3$; *cdk-1(ne2257)*-OP50: $n = 98$, $N = 3$; WT-EHEC group: $n = 115$, $N = 3$; *cdk-1(ne2257)*-EHEC: $n = 84$, $N = 3$, $P = 0.0007$. **e** The percentage of ACT-5 mislocalization of WT and *cyb-3 o/e* animals treated with *cyb-1* RNAi or L4440 RNAi control and EHEC at 20 °C for 3 days. WT-L4440: $n = 95$, $N = 3$; WT-*cyb-1* RNAi: $n = 104$, $N = 3$, $P = 0.035$; *cyb-3 o/e*-L4440: $n = 94$, $N = 3$; *cyb-3 o/e* -*cyb-1* RNAi: $n = 99$, $N = 3$, $P = 0.0016$. **f** The percentage of ACT-5 mislocalization of *mC::ACT-5* animals treated RNAi of the CDK-1 activator genes, *cdc-25.1* ($n = 134$, $N = 3$, $P < 0.0001$) and *cdc-25.2* ($n = 121$, $N = 3$, $P = 0.0001$), or L4440 RNAi control ($n = 226$, $N = 5$) after EHEC infection at 20 °C for 4 days. **g** The percentage of ACT-5 mislocalization of *mC::ACT-5* animals treated with RNAi of the CDK-1 inhibitor genes, including *cki-1*, *cki-2*, *cks-1*, and *wee-1.1*, after EHEC infection for 3 and 4 days. 3 DPI (L4440: $n = 364$, $N = 3$; RNAi *cki-1*: $n = 104$, $N = 3$, $P < 0.0001$; *cki-2*: $n = 123$, $N = 3$, $P = 0.0002$; *cks-1*: $n = 123$, $N = 3$, $P < 0.0001$; *wee-1.1*: $n = 132$, $N = 3$, $P < 0.0001$); 4 DPI (L4440: $n = 360$, $N = 3$; RNAi *cki-1*: $n = 86$, $N = 3$, $P = 0.0568$; *cki-2*: $n = 120$, $N = 3$, $P = 0.3013$; *cks-1*: $n = 116$, $N = 3$, $P = 0.1041$; *wee-1.1*: $n = 131$, $N = 3$, $P = 0.4199$). **h** The representative western blot of p-SUN-1$^{Ser43}$ and α-tubulin (loading control) in N2 animals fed with OP50 or EHEC and treated with RO3306 or control vehicle was shown. **i** The quantitative results of the western blot analysis in panel (h) was shown. **P < 0.001 and ***P < 0.001 by *t* test. **j** The percentage of ACT-5 mislocalization of *mC::ACT-5* animals fed with OP50 or EHEC with or without the treatment of 1 μM of the CDK-1 inhibitor, RO3306. **P < 0.001 compared to the vehicle control group. OP control group: $n = 97$, $N = 3$; OP-RO3306: $n = 99$, $N = 3$; EHEC control: $n = 96$, $N = 3$; EHEC-RO3306: $n = 99$, $N = 3$, $P = 0.0241$. All quantitative data are presented as mean ± SEM, and each dot represented an independent result in the bar chart. All data statistics were based on: *$P < 0.05$, **$P < 0.01$, and ***$P < 0.001$ by unpaired *t* test (two-tailed). Source data are provided as a Source Data file.

ACT-5 mislocalization compared to the L4440 control after EHEC infection for 3 days (Fig. 4e). Moreover, RNAi of *cyk-1* or *pfn-1* also significantly suppressed the EHEC-induced ACT-5 mislocalization in the WT and *cyb-3 o/e* animals fed with EHEC for 4 days. These data suggested that *cyk-1* and *pfn-1* act downstream of the CYB-3 signal in the EHEC-induced microvillar actin rearrangement. To further reconfirm these results, we performed another epistasis experiment using the *cyk-1 o/e* animals. RNAi of *cyb-3* and *pfn-1* in the *cyk-1 o/e* animals did not affect the expression pattern of ACT-5 when fed with OP50 for 3 and 4 days (Fig. 4f). In agreement with the results in Fig. 4b, *cyk-1 o/e* animals exhibited early onset of ACT-5 mislocalization in the L4440 control after EHEC infection for 3 days (Fig. 4g). Moreover, knockdown of *cyb-3* and *pfn-1* significantly suppressed the ACT-5 mislocalization in the *cyk-1 o/e* animals fed with EHEC for 3 or 4 days. Taken together, our epistasis analyses put the *cyk-1/pfn-1* in the same and downstream of the *cdk-1/cyb-3* signal pathway in the EHEC-induced microvillar actin rearrangement.

We next tested whether *C. elegans* with this EHEC-induced pathological phenotype, the redistribution of ACT-5 subcellular localization, is correlated with its susceptibility to EHEC infection. The survival ratios of N2 animals with knocked down *cdk-1*, *cyb-3*, *cyk-1*, or *pfn-1* genes and fed with EHEC were examined. Our result demonstrated that RNAi of *cdk-1*, *cyb-3*, *cyk-1*, and *pfn-1* all conferred animals with resistance to EHEC infection (Fig. 4h), and the lifespans of animals fed with OP50 and knocked down *cdk-1*, *cyb-3*, *cyk-1*, and *pfn-1* were similar to that of L4440 RNAi control (Supplementary Fig. S3a). Moreover, *cyb-3 o/e* and *cyk-1 o/e* animals, with significant early onset of the ACT-5 mislocalization phenotype, were more susceptible to EHEC infection compared to WT N2 (Fig. 4i). The *cdk-1(ne2257)* animals, with significantly reduced ACT-5 mislocalization phenotype, were significantly resistant to EHEC. Moreover, the *cyb-3 o/e* and *cyk-1 o/e* animals were not short-lived fed with OP50, and the median survival day of *cdk-1(ne2257)* animals was similar to that of N2 (Supplementary Fig. S3b). Taken together, our data unveiled not only the role of the CYB-3/CDK-1-CYK-1/PFN-1 signaling axis in the EHEC-induced microvillar actin rearrangement but also the correlation between the pathogenesis

of microvillar effacement and host susceptibility to EHEC infection.

**CDK-1-CYK-1 signal confers the EHEC-induced ACT-5 mislocalization in *C. elegans*.** We found that the actin nucleator CYK-1 is required for the CYB-3/CDK-1 signal in EHEC-induced microvillus-specific actin rearrangement. We next investigated whether CYK-1 may interact with ACT-5 to form F-actin in the basolateral cytosol of the intestinal cells during EHEC infection. To this end, we generated a *C. elegans* strain with the expression of mC::ACT-5 as well as CYK-1::GFP to monitor the interaction between these two proteins during EHEC infection in vivo. The results of confocal microscopy analysis are shown in Fig. 5a, b. The CYK-1::GFP signals were mostly colocalized with mC::ACT-5 in animals infected by EHEC, which suggests that CYK-1 might interact with ACT-5 in the basolateral cytosol to induce ACT-5 mislocalization during EHEC infection. Moreover, we observed that the fluorescent intensity of CYK-1::GFP is significantly increased in EHEC-infected animals compared to the OP50 control (Fig. 5a and quantitative analysis in Fig. 5c). To reconfirm that EHEC-induced CYK-1::GFP protein level, we performed a western-blotting analysis to examine the protein expression level of CYK-1::GFP in *C. elegans*. Our result demonstrated that the protein levels of CYK-1::GFP were significantly increased in EHEC-infected *C. elegans* (Fig. 5d and quantified in Fig. 5e). Interestingly, the mRNA level of *cyk-1::gfp* was not increased in EHEC-treated animals (Fig. 5f). Moreover, we also demonstrated that this EHEC-induced upregulation of CYK-1::GFP protein level is *cyb-3* dependent (Fig. 5g and quantified in Fig. 5h). Taken together, these results suggested that EHEC can induce CYK-1::GFP protein level in the basolateral cytoplasm by a, yet unknown, post-transcriptional regulation that dependent on the CYB-3/CDK-1 signal.

In order to test whether the *cyb-3*, *cdk-1*, *cyk-1*, and *pfn-1* are all involved in regulating ACT-5 polymerization in the basolateral cytosol of intestinal cells, we performed a fluorescence recovery after photobleaching (FRAP) assay. The fluorescence recovery rates of *mC::ACT-5* animals subjected to *cyb-3*, *cdk-1*, *cyk-1*, and *pfn-1* RNAi and infected by EHEC were all significantly reduced compared to L4440 control (Fig. 5i and quantified in Fig. 5j).

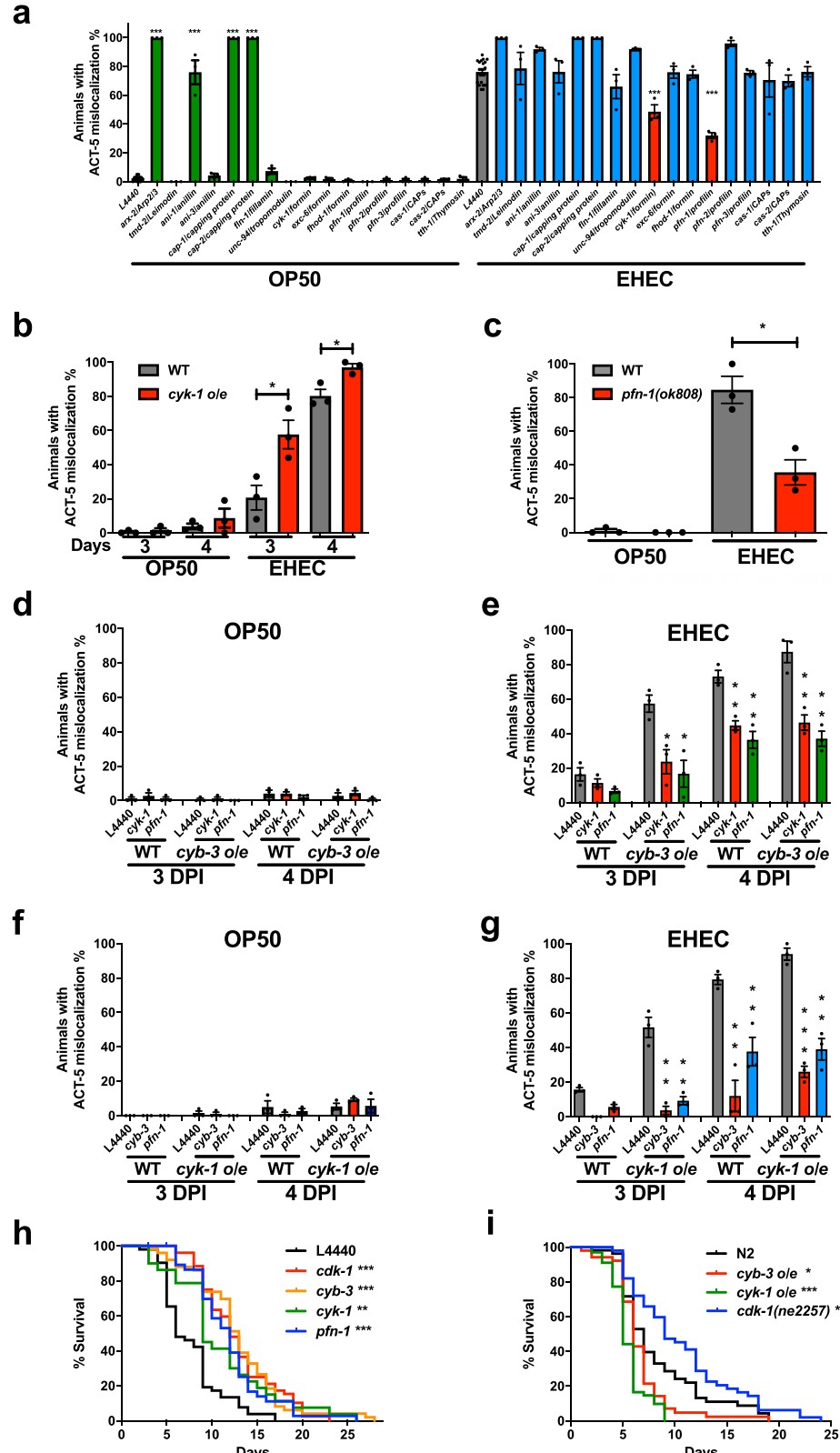

These results not only reconfirmed the dynamic nature of these basolateral *mC::ACT-5* punctae as F-actin structures but also reconfirmed that the CYB-3/CDK-1-CYK-1/PFN-1 signal is required for the EHEC-induced ACT-5 mislocalization.

DIAPH1/CYK-1 has been reported to be a substrate of the CDK-1 serine/threonine kinase[14–16]. We, therefore, generated a transgenic strain that expresses mC::ACT-5, CYK-1::GFP, and

CDK-1::BFP to simultaneously observe these proteins in vivo. We found that the signals of mC::ACT-5, CYK-1::GFP, and CDK-1::BFP colocalized upon EHEC infection by confocal microscopy analysis, but mC::ACT-5 signals were not colocalized to either CYK-1::GFP nor CDK-1::BFP in OP50-treated worms (Fig. 6a and quantitative analysis in Fig. 6b). These data, together with the above-mentioned reports that formin can polymerize actin and

**Fig. 4 CYK-1 and PFN-1 act downstream of the CYB-3 signal pathway in the EHEC-induced ACT-5 mislocalization. a** The percentage of ACT-5 mislocalization in *mC::ACT-5* animals subjected to RNAi of various actin modifiers and fed with OP50 or EHEC for 4 days were examined. **b** The quantification results of ACT-5 mislocalization in wild-type (WT) and *cyk-1* overexpressing (*cyk-1 o/e*) animals exposed to OP50 or EHEC for 3 and 4 days. 3 DPI (WT-OP50 group: *n* = 111, *N* = 3; *cyk-1 o/e*-OP50: *n* = 125, *N* = 3, *P* = 0.6089; WT-EHEC group: *n* = 116, *N* = 3; *cyk-1 o/e*-EHEC: *n* = 120, *N* = 3, *P* = 0.0289); 4 DPI (WT-OP50 group: *n* = 110, *N* = 3; *cyk-1 o/e*-OP50: *n* = 118, *N* = 3, *P* = 0.4531; WT-EHEC group: *n* = 108, *N* = 3; *cyk-1 o/e*-EHEC: *n* = 107, *N* = 3, *P* = 0.019). **c** The percentage of ACT-5 mislocalization of wild-type (WT);*mC::ACT* and *pfn-1(ok808)*;*mC::ACT-5* animals fed with OP50 or EHEC at 20 °C for 4 days were examined. WT-OP50 group: *n* = 109, *N* = 3; *pfn-1(ok808)*-OP50: *n* = 96, *N* = 3, *P* = 0.3739; WT-EHEC group: *n* = 91, *N* = 3; *pfn-1 (ok808)*-EHEC: *n* = 100, *N* = 3, *P* = 0.011. **d** The percentage of ACT-5 mislocalization was examined in the *mC::ACT-5* animals with wild-type (WT) and *cyb-3* overexpressing (*cyb-3 o/e*) background treated with *cyk-1* or *pfn-1* RNAi and fed with OP50 at 20 °C for 3 or 4 days post infection. 3 DPI (WT-L4440: *n* = 107, *N* = 3; WT-*cyk-1*: *n* = 120, *N* = 3, *P* = 0.4573; WT-*pfn-1*: *n* = 106, *N* = 3, *P* > 0.9999; *cyb-3 o/e*-L4440: *n* = 122, *N* = 3; *cyb-3 o/e*-*cyk-1*: *n* = 94, *N* = 3, *P* = 0.7953; *cyb-3 o/e*-*pfn-1*: *n* = 104, *N* = 3, *P* = 0.3739); 4 DPI (WT-L4440: *n* = 106, *N* = 3; WT-*cyk-1*: *n* = 106, *N* = 3, *P* > 0.9999; WT-*pfn-1*: *n* = 106, *N* = 3, *P* = 0.4353; *cyb-3 o/e*-L4440: *n* = 121, *N* = 3; *cyb-3 o/e*-*cyk-1*: *n* = 103, *N* = 3, *P* = 0.4929; *cyb-3 o/e*-*pfn-1*: *n* = 119, *N* = 3, *P* = 0.3486). **e** The percentage of ACT-5 mislocalization was examined in the *mC::ACT-5* animals with wild-type (WT) and *cyb-3* overexpressing (*cyb-3 o/e*) background treated with *cyk-1* or *pfn-1* RNAi and fed with EHEC at 20 °C for 3 or 4 days. 3 DPI (WT-L4440: *n* = 92, *N* = 3; WT-*cyk-1*: *n* = 97, *N* = 3, *P* = 0.325; WT-*pfn-1*: n = 123, *N* = 3, *P* = 0.0704; *cyb-3 o/e*-L4440: *n* = 107, *N* = 3; *cyb-3 o/e*-*cyk-1*: *n* = 84, *N* = 3, *P* = 0.0172; *cyb-3 o/e*-*pfn-1*: *n* = 113, *N* = 3, *P* = 0.0117); 4 DPI (WT-L4440: *n* = 84, *N* = 3; WT-*cyk-1*: *n* = 101, *N* = 3, *P* = 0.0033; WT-*pfn-1*: *n* = 104, *N* = 3, *P* = 0.0038; *cyb-3 o/e*-L4440: *n* = 84, *N* = 3; *cyb-3 o/e*-*cyk-1*: *n* = 82, *N* = 3, *P* = 0.0058; *cyb-3 o/e*-*pfn-1*: *n* = 102, *N* = 3, *P* = 0.0026). **f** The percentage of ACT-5 mislocalization were examined in the *mC::ACT-5* worms with WT and *cyk-1*-overexpressing (*cyk-1 o/e*) background and treated with *cyb-3* or *pfn-1* RNAi feeding on OP50 at 20 °C, for 3 or 4 days. 3 DPI (WT-L4440: *n* = 106, *N* = 3; WT-*cyb-3*: *n* = 106, *N* = 3, *P* > 0.9999; WT-*pfn-1*: *n* = 84, *N* = 3, *P* > 0.9999; *cyk-1 o/e*-L4440: *n* = 78, *N* = 3; *cyk-1 o/e*-*cyb-3*: *n* = 78, *N* = 3, *P* = 0.8512; *cyk-1 o/e*-*pfn-1*: *n* = 84, *N* = 3, *P* = 0.3739); 4 DPI (WT-L4440: *n* = 101, *N* = 3; WT-*cyb-3*: *n* = 105, *N* = 3, *P* = 0.3453; WT-*pfn-1*: *n* = 80, *N* = 3, *P* = 0.5807; *cyk-1 o/e*-L4440: *n* = 85, *N* = 3; *cyk-1 o/e*-*cyb-3*: *n* = 74, *N* = 3, *P* = 0.1234; *cyk-1 o/e*-*pfn-1*: *n* = 84, *N* = 3, *P* = 0.9415). **g** The percentage of ACT-5 mislocalization was examined in the *mC::ACT-5* worms with WT and *cyk-1* overexpressing (*cyk-1 o/e*) background and treated with *cyb-3* or *pfn-1* RNAi fed on EHEC at 20 °C for 3 or 4 days. 3 DPI (WT-L4440: *n* = 103, *N* = 3; WT-*cyb-3*: *n* = 111, *N* = 3, *P* = 0.0002; WT-*pfn-1*: *n* = 105, *N* = 3, *P* = 0.0061; *cyk-1 o/e*-L4440: *n* = 76, *N* = 3; *cyk-1 o/e*-*cyb-3*: *n* = 75, *N* = 3, *P* = 0.0016; *cyk-1 o/e*-*pfn-1*: *n* = 95, *N* = 3, *P* = 0.0025); 4 DPI (WT-L4440: *n* = 101, *N* = 3; WT-*cyb-3*: *n* = 108, *N* = 3, *P* = 0.0021; WT-*pfn-1*: *n* = 103, *N* = 3, *P* = 0.0086; *cyk-1 o/e*-L4440: *n* = 71, *N* = 3; *cyk-1 o/e*- *cyb-3*: *n* = 73, *N* = 3, *P* = 0.0001; *cyk-1 o/e*-*pfn-1*: *n* = 88, *N* = 3, *P* = 0.0015). **h** Survival curves of N2 animals with RNAi of *cdk-1*, *cyb-3*, *cyk-1*, or *pfn-1* genes and fed with EHEC. **i** Survival curves of wild-type N2, *cyb-3*-overexpressing (*cyb-3 o/e*), *cyk-1*-overexpressing (*cyk-1 o/e*), and *cdk-1(ne2257)* animals infected with EHEC at 20 °C were examined. For the survival analyses: *\*P* < 0.05, *\*\*P* < 0.01, and *\*\*\*P* < 0.001 by the Mantel–Cox log-rank test. The other quantitative data are presented as mean ± SEM, and each dot represented an independent result in the bar chart. Except for the survival analyses, all data statistics based on: *\*P* < 0.05, *\*\*P* < 0.01, and *\*\*\*P* < 0.001 by unpaired *t* test (two-tailed). Source data are provided as a Source Data file.

CDK-1 can phosphorylate formin, suggested a scenario that ACT-5, CYK-1, and CDK-1 might interact with one another during the EHEC-induced microvillar actin rearrangement in vivo.

However, the phosphorylation site of CYK-1 in this EHEC-induced ACT-5 mislocalization has not been confirmed. CDK-1 has a phosphorylation consensus sequence of (K/R)(S/T)PX or (S/T)PX, where X is any amino acid[27]. We identified three S/T-P sites in the important functional domains of CYK-1, that is, Threonine[611] in the FH3 domain, Threonine[1232] in the FH2 domain, and Threonine[1273] in the diaphanous autoregulatory domain (DAD) in CYK-1, as the potential phosphorylation sites of CDK-1 (Fig. 6c). We generated *C. elegans* strains with the expression of these phosphorylation-resistant CYK-1(T611A, T1232A, and T1273A), and examined whether these mutations affect the colocalization between CYK-1 and ACT-5 in vivo. From our confocal image analysis, we found that the GFP signals of CYK-1(T611A) and CYK-1(T1232A) remained colocalized with mC::ACT-5 signals in the EHEC-infected worms (Fig. 6d, and quantification of confocal colocalization shown in Fig. 6e). Moreover, animals with an expression of CYK-1, CYK-1, CYK-1 (T611A), and CYK-1(T1232A) transgenes, driven by the intestine-specific promoter *app-1p*, all exhibited early onset of EHEC-induced ACT-5 mislocalization 3 days post infection (Fig. 6f). By contrast, the CYK-1(T1273A)::GFP signals did not colocalize with mC::ACT-5 signals (Fig. 6d and quantification of confocal colocalization shown in Fig. 6e), and animals with the CYK-1(T1273A)::GFP transgenes conferred reduced ACT-5 mislocalization compared to the animals with the CYK-1(WT) transgenes 3 and 4 days post EHEC infection (Fig. 6f). We then generated the phosphomimetic CYK-1(T1273E #1 and #2) to reconfirm that the Threonine[1273] residue is required for the EHEC-induced ACT-5 mislocalization. The signals of CYK-1

(T1273E #1)::GFP colocalized with mC::ACT-5 signals (Fig. 6d and quantification of confocal colocalization shown in Fig. 6e), and CYK-1(T1273E) animals showed comparable percentages of EHEC-induced ACT-5 mislocalization compared to that of CYK-1(WT) after EHEC infection (Fig. 6f). Moreover, the expression levels of these CYK-1::GFP were analyzed and quantified by western blot (Fig. 6g and quantified in Fig. 6h). Our results clearly showed that EHEC can increase the protein level of CYK-1::GFP (WT #1) and CYK-1::GFP(T1273E #1), but not CYK-1::GFP (T1273A #1). In order to reconfirm that the significant upregulation of the EHEC-induced ACT-5 mislocalization in the animals with overexpression of CYK-1, driven by the intestine-specific promoter *app-1p*, is CDK-1-dependent, we repeated the above experiment with RNAi of *cdk-1* (Fig. 6i). Our results showed that the upregulation of the EHEC-induced ACT-5 mislocalization is significantly abolished by knockdown of *cdk-1* in the WT and CYK-1(WT #1) *o/e* group. However, animals with overexpression of CYK-1(T1273E #1) still exhibited the EHEC-induced ACT-5 mislocalization phenotype with RNAi of *cdk-1*. Taken together, these data suggested that ACT-5 mislocalization in the CYK-1::GFP(WT) *o/e* animals is CDK-1-dependent and phosphorylation of CYK-1, at least, on Threonine[1273] in the DAD is involved in this EHEC-induced pathology.

Finally, to examine whether EHEC induces phosphorylation of CYK-1, we performed phospho-tag gel analysis of the CYK-1:: GFP protein (Fig. 6j, and quantification in Fig. 6k). Our results showed that EHEC can significantly induce the level of phospho-CYK-1::GFP, and this phosphorylation signal of CYK-1::GFP induced by EHEC is *cdk-1*-dependent.

**RfaD plays a role in EHEC-induced microvillar actin efface-ment in *C. elegans* and human intestinal cells.** T3SS and its

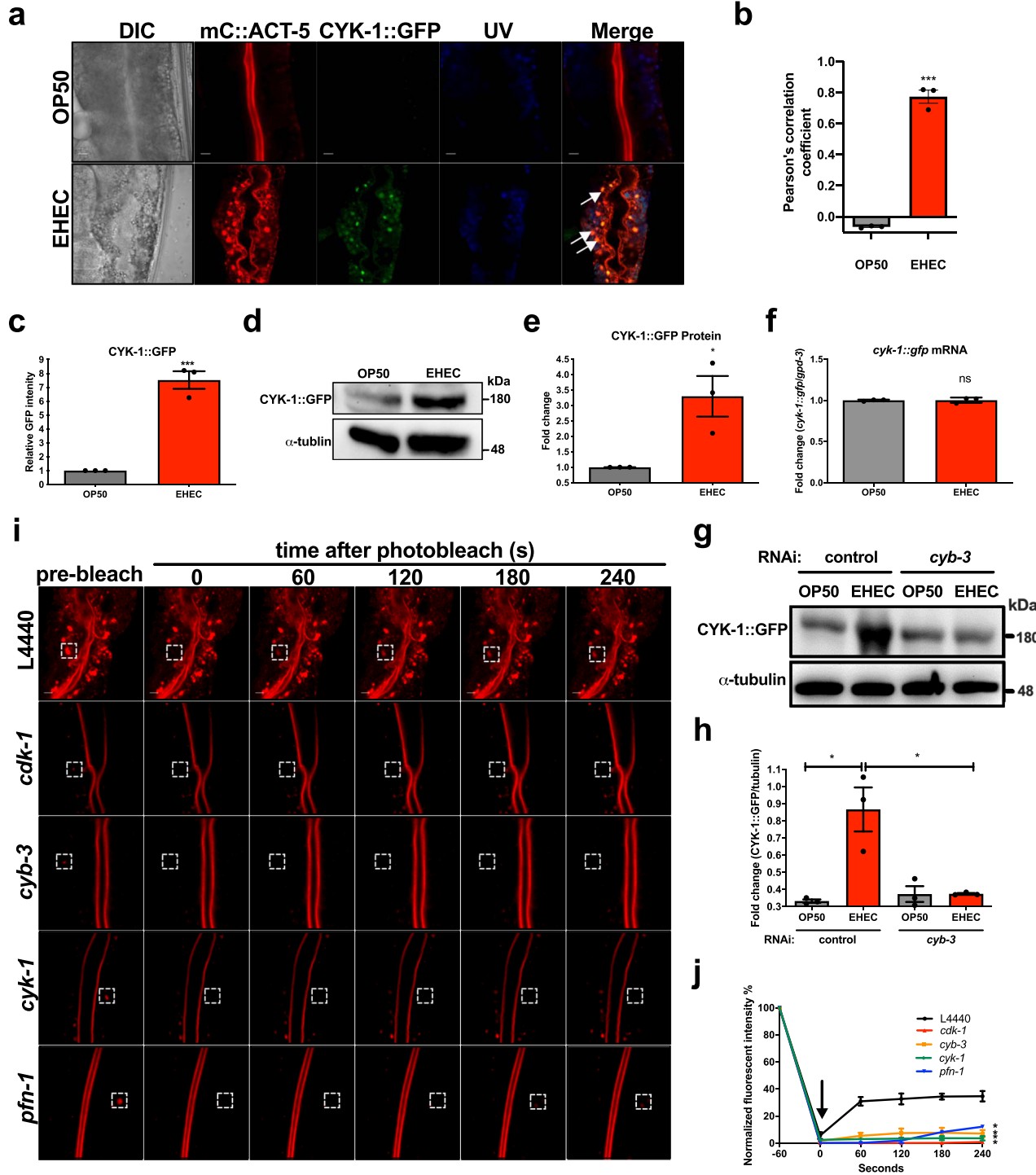

effectors have long been known to mediate A/E lesion formation. Moreover, our previous studies also demonstrated the Stx1 and lipopolysaccharide (LPS) are required for the full pathogenesis of EHEC in *C. elegans*[7,8]. We, therefore, aimed to test whether these EHEC virulence determinants are required for the EHEC-induced redistribution of microvillar ACT-5. To this end, we infected *mC::ACT-5* animals with these EHEC gene deletion mutants. We found that deletions of the T3SS system genes, including the master transcriptional regulator *ler*, the injectisome components *espA*, *espB*, and *espD*, and *escN*, and the secreted effectors *eae*, *espF*, *espFu*, *espG*, *espH*, *espM*, *map*, and *tir*, did not suppress the EHEC-induced ACT-5 mislocalization phenotype in *C. elegans* (Fig. 7a). Moreover, deletions of the toxin genes including the

*hlyA* (encoding for the hemolysin A), *hlyE* (encoding for the hemolysin E), and *stx2* (encoding for the Stx2), did not suppress EHEC to induce ACT-5 mislocalization in *C. elegans*. Interestingly, deletions of the *stx1* gene (encoding for the Stx1) and *rfaD* gene (encoding the ADP-L-glycero-D-manno-heptose-6-epimerase that functions in the biosynthesis of the LPS inner core) significantly diminished the EHEC-induced ACT-5 mislocalization phenotype in *C. elegans*. We next tested whether these two EHEC virulence determinants mediate in the mislocalization of ACT-5 through the CYB-3/CDK-1-CYK-1/PFN-1 signal axis. Our qRT-PCR analysis showed that the *cyb-3* mRNA in the EHEC Δ*rfaD* mutant infected worms is significantly lower compared to the WT-EHEC (Fig. 7b). However, the *cyb-3* mRNA

**Fig. 5 CYK-1 is required for the polymerization of microvillus-specific ACT-5 in the basolateral cytosol of intestinal cells. a** The representative confocal images of *mC::ACT-5;CYK-1::GFP* transgenic animals exposed to OP50 and EHEC for 4 days. The autofluorescence signals were monitored by UV channel (UV). The merged image (Merge) shows 3 overlaid fluorescent images. The arrowheads indicate that GFP signals colocalized with mCherry signals. The scale bars represent 5 μm. The experiment was performed independently three times. **b** The quantification analysis of the colocalization of the CYK-1::GFP with mC::ACT-5 signals in panel (**a**). $N = 3$ and $P < 0.001$ by $t$ test. **c** The quantification result of GFP intensity in the *mC::ACT-5;CYK-1::GFP* worms fed with OP50 ($n = 30$, $N = 3$) or EHEC ($n = 30$, $N = 3$, $P = 0.0005$ by $t$ test) for 4 days. **d** Western-blotting analysis of the CYK-1::GFP expression level by anti-GFP antibody in *mC::ACT-5;CYK-1::GFP* worms fed with OP50 or EHEC for 4 days. α-Tubulin served as a protein loading control. **e** The quantification result of CYK-1::GFP levels from western-blotting analyses in panel (**d**). $N = 3$ and $P < 0.05$ by $t$ test. **f** qRT-PCR analysis of the *cyk-1::gfp* mRNA in the *mC::ACT-5; CYK-1::GFP* animals treated with OP50 or EHEC for 4 days. $N = 3$ and n.s. indicated no significant difference by $t$ test. **g** Western-blotting analysis of the CYK-1::GFP expression level by anti-GFP antibody in *mC::ACT-5;CYK-1::GFP* worms fed with OP50 or EHEC and RNAi *cyb-3* for 4 days. α-Tubulin served as a protein loading control. **h** The quantification result of CYK-1::GFP levels from the western-blotting analyses in panel (**g**). $N = 3$ and * indicated $P < 0.05$ by unpaired $t$ test (two-tailed). **i** The representative time-lapse confocal images of fluorescence recovery after photobleaching (FRAP) experiment in *mC::ACT-5* worms subjected to *cdk-1*, *cyb-3*, *cyk-1* and *pfn-1* RNAi and fed with EHEC for 4 days. White dotted squares indicate the area of photobleaching. The scale bars represent 5 μm. The experiment was performed independently three times. **j** The normalized recovery curves of the FRAP analysis. The black arrowhead indicates the time point of photobleaching. All $N = 3$ and * indicated $P < 0.05$ by unpaired $t$ test (two-tailed). All quantitative data are presented as mean ± SEM, and each dot represented an independent result in the bar chart. All data statistics based on: *$P < 0.05$, **$P < 0.01$, and ***$P < 0.001$ by unpaired $t$ test (two-tailed). Source data are provided as a Source Data file.

level in worms infected by the EHEC Δstx1 mutant is similar to the WT-EHEC. Moreover, we also fed *cyb-3* o/e worms with EHEC-WT and the EHEC ΔrfaD mutant (Fig. 7c). Our results showed the EHEC-induced mC::ACT-5 mislocalization is significantly diminished in the EHEC ΔrfaD-treated worms. Taken together, our results suggested that the EHEC LPS may be a major virulence determinant that induces microvillar ACT-5 mislocalization through the CYB-3/CDK-1-CYK-1/PFN-1 signaling pathway, and Stx1 may induce the mislocalization of ACT-5 in a CYB-3-independent manner.

We next tested whether deletion of *rfaD* affected the EHEC-induced colocalization between ACT-5 and CYK-1 in *C. elegans*. Our confocal microscopy analysis showed that deletion of *rfaD* gene diminishes the colocalization between CYK-1::GFP and mC:: ACT-5 signals induced by EHEC infection (Fig. 7d). Moreover, the upregulation of CYK-1::GFP expression by EHEC was also significantly dampened in worms treated with the EHEC ΔrfaD mutant (Fig. 7d and quantitative analysis in Fig. 7e). Next, to reconfirm that the bacterial factors, *rfaD* and *stx1*, and the host factors, *cdk-1*, *cyb-3*, and *cyk-1*, are required for the microvillar effacement in vivo. The TEM analysis for N2 animals treated with OP50, EHEC, EHEC ΔrfaD, and EHEC Δstx1, and *rrf-3(pk1426)* animals treated with EHEC and RNAi control L4440, *cdk-1* RNAi, *cyb-3* RNAi, and *cyk-1* RNAi has been performed (Supplementary Fig. S4a–h). Our results showed that EHEC ΔrfaD does not induce microvillar effacement in *C. elegans* (Supplementary Fig. S4c). EHEC Δstx1 (Supplementary Fig. S4d) still can induce microvillar effacement, but is significantly less severe compared to EHEC-WT (Fig. S4b). Moreover, RNAi of *cdk-1*, *cyb-3*, and *cyk-1* can significantly suppress the EHEC-induced microvillar effacement compared to the L4440 RNAi control (Supplementary Fig. S4e–h).

Finally, to monitor the microvilli structure in mammals, we used cultured human Caco-2 intestinal epithelial cells (IECs) as our in vitro cell model. Our scanning electron microscopy (SEM) results showed that EHEC infection significantly effaces the microvilli on the Caco-2 cells (Fig. 7f and quantitative analysis in Fig. 7g). However, we observed that deletion of *rfaD* in EHEC significantly abolished its ability to induce microvillar effacement on Caco-2 cells compared to the WT-EHEC control. Interestingly, while the EHECΔler mutant failed to induce pedestals in HeLa cells (Supplementary Fig. S5a) and thus lost its ability to tightly attach to the host (Supplementary Fig. S5b), the EHEC Δler and ΔespA mutants could still significantly induce microvillar effacement on Caco-2 cells (Supplementary Fig. S5c and quantitative analysis in Supplementary Fig. S5d). These data suggested that the T3SS is required for the attachment phenotype,

but is dispensable for the effacement lesions induced by EHEC. Nevertheless, our data suggested that the EHEC LPS is required for the induction of microvillus-specific actin mislocalization in *C. elegans* and the effacement of microvilli on human Caco-2 intestinal cells in vitro.

To further reconfirm that the cyclin B3/CDK-1 signal is required for the EHEC-induced microvillar effacement in mammals, we first tested whether Cyclin B3/CCNB3 is also required for the EHEC-induced microvillar actin effacement in Caco-2 cells. Our SEM results showed that the EHEC-induced microvillar effacement was significantly abolished in the two independent *CCNB3* RNAi stable knockdown Caco-2 cells (Supplementary Fig. S6a–c). We next aimed to reconfirm the role of CDK-1 in the EHEC-induced microvillar effacement. Our SEM results showed that the CDK-1-specific inhibitor RO3306 significantly abolishes the EHEC-induced microvillar effacement on Caco-2 intestinal cells (Fig. 7h and quantitative analysis in Fig. 7i). The efficacy for the treatment of RO3306 in Caco-2 cells was monitored (Supplementary Fig. S5e and quantitative analysis in Supplementary Fig. S5f). Our results showed that EHEC can significantly induce phospho-PP1α$^{Thr302}$ level[28], which is CDK-1 phosphorylation substrate, and RO3306 significantly inhibited the p-PP1α$^{Thr302}$ level in Caco-2 cells. Our results suggested that the CDK-1 activity is required for the EHEC-induced microvillar effacement in mammals. Taken together, our data showed that the CCNB3/CDK-1 signal is required for the EHEC-induced microvillar effacement on human Caco-2 intestinal cells.

To further reconfirm that the diaphanous-related formin/CYK-1 signal is also activated and involved in the EHEC-induced microvillar effacement in mammals, we first analyzed the DIAPH1, DIAPH2, and DIAPH3, the orthologs of CYK-1, protein levels in the EHEC-infected Caco-2 cells. Our data showed that only the DIAPH1 protein level is significantly upregulated post EHEC infection (Supplementary Fig. S6d and quantitative analysis in Supplementary Fig. S6e). Moreover, the *DIAPH1* mRNA level in the EHEC-infected cells was similar to the control (Supplementary Fig. S6f), reconfirming the notion we found in *C. elegans* that the upregulation of DIAPH1 expression is through post-transcriptional regulation. Next, the pan-formin inhibitor SMIFH2 also significantly abolished the EHEC-induced microvillar effacement on Caco-2 cells (Supplementary Fig. S6g and quantitative analysis in Supplementary Fig. S6h). Finally, specifically knockdown of *DIAPH1* and *DIAPH2*, but not *DIAPH3*, by short hairpin RNA (shRNA) can significantly abolish the EHEC-induced microvillar effacement on Caco-2 cells (Fig. 7j and quantitative analysis in Fig. 7k; also in Supplementary

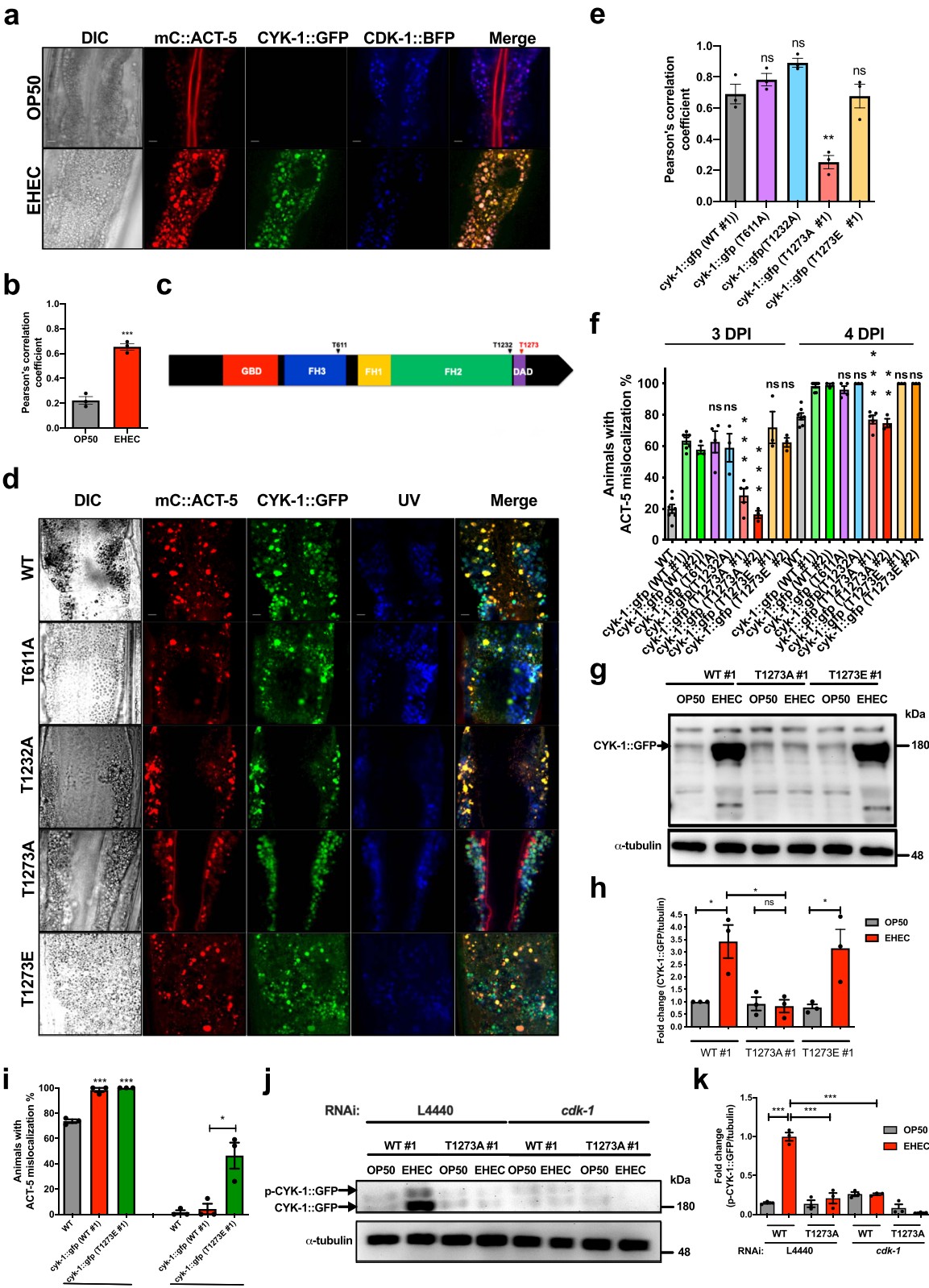

 Taken together, our results demonstrated that although only DIAPH1 protein level is subjected to be regulated by EHEC, the DIAPH1 and DIAPH2 are involved in the microvillar effacement on human Caco-2 cells induced by EHEC infection.

Finally, one potential caveat of our study of the EHEC-induced G2/M signal in the 20 intestinal cells of *C. elegans* is that all somatic cells in adult *C. elegans* are considered terminally differentiated and post-mitotic without any further cellular division. However, it has been reported that the intestinal cells can undergo nuclear endoreduplication, that is, in the first larval stage 14 additional nuclear divisions occur without cytokinesis, and a doubling of nuclear content accompanies each larval molt, resulting in 34 total nuclei in the same 20 intestinal cells that were

**Fig. 6 Phosphorylation of CYK-1 at Threonine[1273] is involved in EHEC-induced ACT-5 mislocalization. a** The representative confocal images of animals expressed mC::ACT-5, CYK-1::GFP, and CDK-1::BFP fed with OP50 or EHEC for 4 days. The scale bars represent 5 μm. **b** The quantification analysis of the colocalization of the CDK-1::BFP with CYK-1::GFP and mC::ACT-5 signals in panel (**a**). $N = 3$ and ***$P < 0.001$ by $t$ test. **c** Diagram of the predicted CDK-1 phosphorylated sites in CYK-1. GBD indicates the Rho GTPase-binding domain. FH1 (formin homology 1), FH2 (formin homology 2), and FH3 (formin homology 3) domains are also indicated. DAD indicates the diaphanous autoregulatory domain. **d** The representative confocal images of animals expressed with wild-type CYK-1::GFP (WT #1), phosphorylation-deficient CYK-1[T611A]::GFP (T611A), CYK-1[T1232A]::GFP (T1232A), CYK-1[T1273A]::GFP (T1273A #1), and phosphomimetic CYK-1[T1273E]::GFP (T1273E #1) infected with EHEC for 4 days. The confocal images of the CYK-1::GFP signals (CYK-1::GFP), the mCherry:: ACT-5 (mC::ACT-5) signals, the autofluorescence signals (UV), and the three fluorescence images overlaid over the DIC image in each group (Merge) are presented. Scale bars represent 5 μm. **e** The quantification analysis of the colocalization of the CYK-1::GFP with mC::ACT-5 signals in panel (**d**). $N = 3$ and **$P < 0.01$ by $t$ test. **f** The percentage of ACT-5 mislocalization of mC::ACT-5 animals with the CYK-1 wild-type (WT), phosphorylation-deficient CYK-1[T611A]::GFP (T611A), and CYK-1[T1232A]::GFP (T1232A), as well as CYK-1[T1273A]::GFP (T1273A) and phosphomimetic CYK-1[T1273E]::GFP (T1273E) driven by the app-1p promoter and fed with EHEC for 3 days (3 DPI) or 4 days (4 DPI) were analyzed. 3 DPI (WT: $n = 114$, $N = 7$; cyk-1::gfp (WT #1): $n = 87$, $N = 7$, $P < 0.0001$; cyk-1::gfp (WT #2): $n = 97$, $N = 3$, $P < 0.0001$; cyk-1::gfp (T611A): $n = 50$, $N = 4$, $P = 0.8944$; cyk-1::gfp (T1232A): $n = 38$, $N = 3$, $P = 0.4916$; cyk-1:: gfp (T1273A #1): $n = 72$, $N = 5$, $P < 0.0001$; cyk-1::gfp (T1273A #2): $n = 98$, $N = 3$, $P = 0.0003$; cyk-1::gfp (T1273E #1): $n = 39$, $N = 3$, $P = 0.2531$; cyk-1::gfp (T1273E #2): $n = 91$, $N = 3$, $P = 0.3022$); 4 DPI (WT: $n = 111$, $N = 7$; cyk-1::gfp (WT #1): $n = 88$, $N = 7$, $P < 0.0001$; cyk-1::gfp (WT #2): $n = 97$, $N = 3$, $P = 0.0004$); cyk-1::gfp (T611A): $n = 48$, $N = 4$, $P = 0.3314$; cyk-1::gfp (T1232A): $n = 37$, $N = 3$, $P = 0.3559$; cyk-1::gfp (T1273A #1): $n = 72$, $N = 5$, $P < 0.0001$; cyk-1::gfp (T1273A #2): $n = 98$, $N = 3$, $P = 0.0011$; cyk-1::gfp (T1273E #1): $n = 37$, $N = 3$, $P = 0.3559$; cyk-1::gfp (T1273E #2): $n = 91$, $N = 3$, $P = 0.3739$).
**g** Western-blotting analysis of the CYK-1::GFP expression level by anti-GFP antibody in cyk-1::gfp (WT #1), cyk-1::gfp (T1273A #1), and cyk-1::gfp (T1273E #1) worms fed with OP50 or EHEC for 4 days. α-Tubulin served as a protein loading control. **h** The quantification result of CYK-1::GFP levels from the western-blotting analyses in panel (**g**). **i** The percentage of ACT-5 mislocalization of mC::ACT-5 animals (WT), and mC::ACT-5 animals with the CYK-1 wild-type [cyk-1::gfp(WT)] and phosphomimetic CYK-1[T1273E]::GFP [cyk-1::gfp(T1273E)] knocked down cdk-1 by RNAi and treated with EHEC for 4 days were analyzed. L4440 (WT: $n = 69$, $N = 3$; cyk-1:gfp (WT #1): $n = 56$, $N = 3$, $P < 0.0001$; cyk-1:gfp (T1273E #1): $n = 58$, $N = 3$, $P < 0.0001$; cdk-1 (WT: $n = 56$, $N = 3$; cyk-1:gfp (WT #1): $n = 57$, $N = 3$, $P = 0.5839$; cyk-1:gfp (T1273E #1): $n = 65$, $N = 3$, $P = 0.0191$). **j** Phospho-tag gel and western-blotting analysis of the CYK-1::GFP by anti-GFP antibody in cyk-1:gfp (WT #1) and cyk-1:gfp (T1273A #1) worms fed with OP50 or EHEC and L4440 RNAi control or cdk-1 RNAi for 4 days. α-Tubulin served as a protein loading control. **k** The quantification result of p-CYK-1::GFP levels from the western-blotting analyses in panel (**j**). All quantitative data are presented as mean ± SEM, and each dot represented an independent result in the bar chart. All data statistics based on: *$P < 0.05$, **$P < 0.01$, and ***$P < 0.001$ by unpaired $t$ test (two-tailed). n.s. Not significant. Source data are provided as a Source Data file.

present at hatching in every *C. elegans* animal[29]. Our results show that EHEC can significantly induce nuclear endoreduplication in the intestinal cells in *C. elegans* (Supplementary Fig. S7a and quantitative analysis in Supplementary Fig. S7b), together with the upregulation of the CYB-3/CDK-1 signal suggesting that EHEC can still induce the G2/M signal in the post-mitotic intestinal cells in *C. elegans*. Moreover, we also reconfirmed that EHEC can induce various G2/M signals and DIAPH1 expression in human Caco-2 cells. Our flow cytometry results showed that G2/M-phase population was significantly increased after EHEC infection compared to the control (Supplementary Fig. S7c). During mitosis histone H3 is phosphorylated at Serine10 in all eukaryotes[30], we also demonstrated that the phosphorylation of Serine10 in histone 3 is significantly increased after EHEC infection (Supplementary Fig. S7d and quantitative analysis in Supplementary Fig. S7e). Moreover, by monitoring the phosphorylation of PP1α at Threonine320, a specific substrate of CDK-1[31], we demonstrated that the mitotic kinase activity of CDK-1 is increased after EHEC infection (Supplementary Fig. S7f and quantitative analysis in Supplementary Fig. S7g). Taken together, our data demonstrated that the G2/M-phase and CDK-1 activity are also induced in human intestinal cells post EHEC infection.

## Discussion

In response to A/E pathogens such as EHEC, IECs undergo a robust and coordinated remodeling of the apical cytoskeleton. Highly dynamic and organized core bundles of actin in the microvilli are disassembled (effacement) and newly synthesized actin pedestals conform to the apical membrane allowing tight attachment of bacterial cells (attachment). While the process for actin pedestal formation is well characterized, the molecular mechanisms of microvillar effacement induced by EHEC are largely unknown. In this study, we used a mCherry-labeled ACT-5 transgenic *C. elegans* to show the EHEC-induced mislocalization of microvillus-specific actin/ACT-5. We consider ACT-5

mislocalization to be a biomarker to monitor microvillar effacement, as it is significantly increased post-EHEC infection. Our results showed that EHEC infection induces the expression of mitotic cyclin B3/CYB-3 and leads to the activation of CDK-1/CDK-1 in *C. elegans*. This CYB-3-CDK-1 signal upregulates the diaphanous-related formin 1, DIAPH1/CYK-1, protein level probably through phosphorylation on Threonine[1273] in the DAD. However, we cannot exclude an alternative explanation that the Threonine[1273] to Alanine or Glutamate mutations may affect the open-close conformation and thus protein level of CYK-1::GFP. Nevertheless, our data suggested that CYK-1 then functions together with profilin/PFN-1 to sequester the microvillus-specific ACT-5 to form F-actin at the basolateral cytoplasm, rather than form the core bundle for the microvillus by Arp2/3 at the apical side of the intestinal cells, thus leading to the deformation of the microvillus structure induced by EHEC. We also found that human cyclin B3/CCNB3, CDK-1, DIAPH1, and DIAPH2 are required for the EHEC-induced microvillar effacement in human intestinal Caco-2 cells. Moreover, we found that two bacterial factors, namely RfaD and Stx1, are involved in the EHEC-induced microvillar effacement in Caco-2 cells and *C. elegans*. Taken together, our study not only clarifies the mechanism of EHEC-induced microvillar effacement but also highlights the potential for targeting the CDK-1-formin signal axis as a therapeutic measure for EHEC infection.

T3SS and its effectors are well known to specifically induce pedestal formation for the tight attachment of EHEC bacterial cells in A/E lesion formation. Interestingly, our data suggested that although T3SS participated in the pedestal formation in human HeLa cells (Supplementary Fig. S5a), it may not be involved in the EHEC-induced microvillar effacement in *C. elegans* and human Caco-2 cells (Fig. 7a and Supplementary Fig. S5c). Moreover, we found that two EHEC virulence factors RfaD and Stx1 and the host CDK-1-DIAPH1 signaling are required for the EHEC-induced microvillar effacement. Although further research is still necessary to identify the detailed mechanisms for

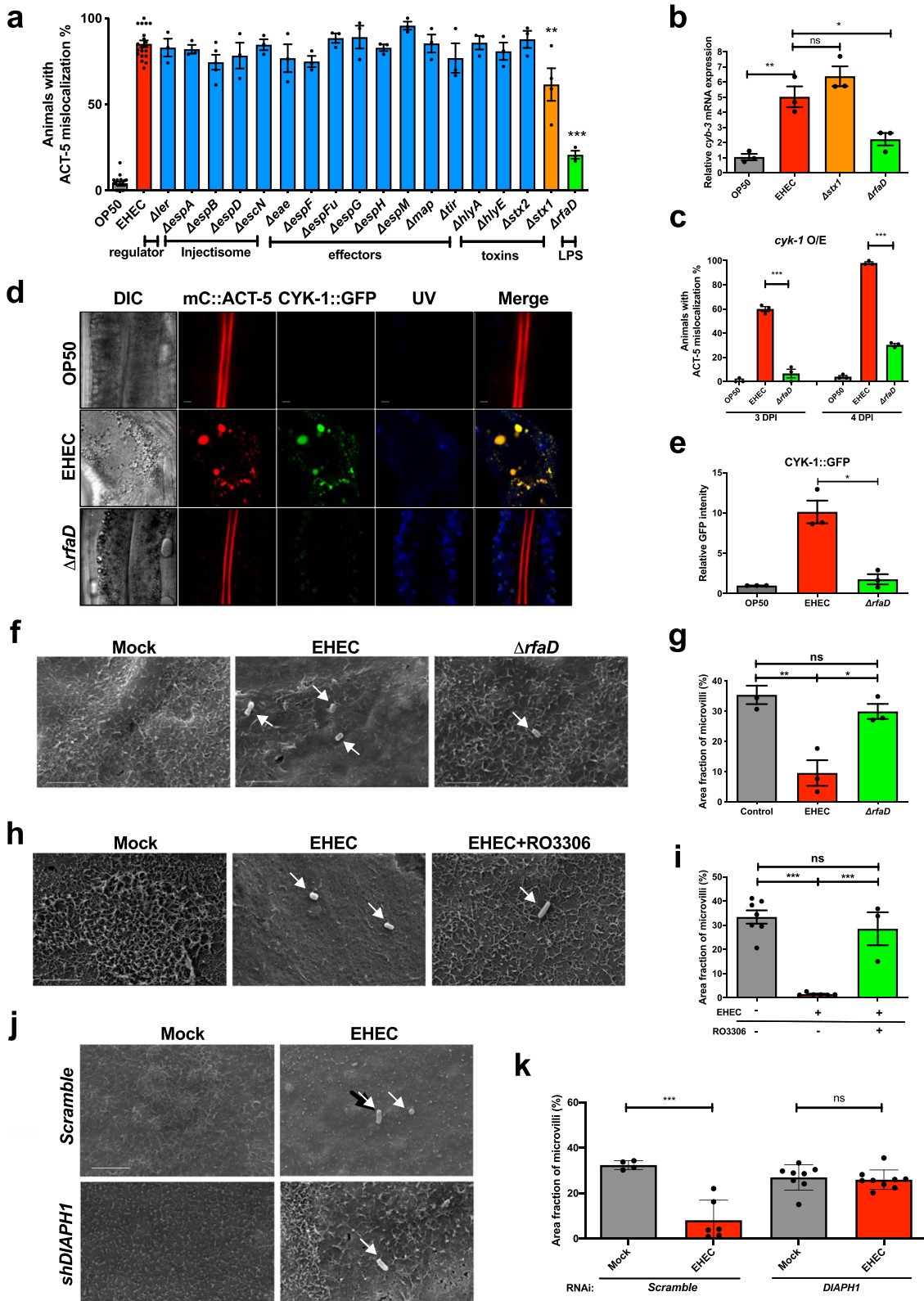

RfaD and Stx1 in this EHEC-induced pathology, our results suggest the notion that effacement and attachment are two distinct phenotypes mediated by different bacterial and host factors.

In summary, EHEC, an A/E pathogen, induces disruption of the actin cytoskeleton and leads to microvillar effacement and pedestal formation in host IECs. These morphological alterations are predicted to diminish the absorptive capacity as well as the

barrier function of IECs, thereby contributing to diarrhea and even severe HC. The major cellular consequences of infection by EHEC are undoubtedly, in part, due to the delivery of diverse effectors directly into host intestinal cells by T3SS encoded by the LEE pathogenicity island. Herein, we demonstrated that two non-LEE virulence determinants, Stx1 and LPS, released by the EHEC also play roles in the alteration of the microvillar brush border.

**Fig. 7 Bacterial RfaD is a major virulence determinant in EHEC-induced microvillar effacement in *C. elegans* and human intestinal cells. a** The percentage of ACT-5 mislocalization in *mCherry::ACT*-5 worms fed with OP50 ($n = 736$, $N = 20$) and EHEC wild-type ($n = 684$, $N = 20$) and different EHEC isogenic gene deletion mutants, including the T3SS regulator *ler* ($n = 99$, $N = 3$, $P = 0.6898$), the T3SS injectisome components *espA* ($n = 108$, $N = 3$, $P = 0.5503$), *espB* ($n = 158$, $N = 5$, $P = 0.6659$), and *espD* ($n = 100$, $N = 3$, $P = 0.2395$), and *escN* ($n = 91$, $N = 3$, $P = 0.9204$), and the T3SS effectors *eae* ($n = 98$, $N = 3$, $P = 0.164$), *espF* ($n = 105$, $N = 3$, $P = 0.0619$), *espFu* ($n = 100$, $N = 3$, $P = 0.528$), *espG* ($n = 97$, $N = 3$, $P = 0.4875$), *espH* ($n = 104$, $N = 3$, $P = 0.6639$), *espM* ($n = 112$, $N = 3$, $P = 0.0546$), *map* ($n = 96$, $N = 3$, $P = 0.9724$), and *tir* ($n = 99$, $N = 3$, $P = 0.173$), and toxins, including *hlyA* (hemolysin A) ($n = 92$, $N = 3$, $P = 0.8988$), *hlyE* (hemolysin E) ($n = 95$, $N = 3$, $P = 0.4307$), *stx2* ($n = 95$, $N = 3$, $P = 0.6309$), and *stx1* (Shiga-like toxin 1) ($n = 126$, $N = 4$, $P = 0.0005$), and *rfaD* (functions in the biosynthesis of LPS) ($n = 76$, $N = 3$, $P < 0.0001$) for 4 days. **b** The qRT-PCR analysis of *cyb-3* mRNA in N2 wild-type animals exposed to OP50, EHEC, the isogenic EHEC with *stx1* deletion (*Δstx1*), and the isogenic EHEC with *rfaD* deletion (*ΔrfaD*) at 20 °C for 4 days, respectively. **c** The percentage of ACT-5 mislocalization in the *cyk-1 o/e* worms fed with OP50, EHEC wild-type, or EHEC *ΔrfaD* bacteria for 3 and 4 days. 3 DPI (OP50: $n = 91$, $N = 3$; EHEC: $n = 97$, $N = 3$; EHEC *ΔrfaD*: $n = 71$, $N = 3$, $P = 0.0002$); 4 DPI (OP50: $n = 91$, $N = 3$; EHEC: $n = 95$, $N = 3$; EHEC *ΔrfaD*: $n = 70$, $N = 3$, $P < 0.0001$). **d** The representative confocal images of transgenic animal expressing mC::ACT-5 and CYK-1::GFP fed with OP50, EHEC, and the isogenic EHEC with *rfaD* deletion (*ΔrfaD*) for 4 days. The DIC images and confocal images of the CYK-1::GFP signals, the mC::ACT-5 signals, the autofluorescence signals (UV), and the three fluorescence images overlaid over in each group (Merge) are presented. Scale bars represent 5 μm. **e** The quantification result of GFP intensity in the *mC::ACT-5;CYK-1::GFP* worms fed with OP50, EHEC, and the isogenic EHEC with *rfaD* deletion (*ΔrfaD*) for 4 days in panel (**d**). *$P < 0.05$ compared to the EHEC wild-type group. **f** The representative SEM images of intestinal Caco-2 cell treated with Mock control, EHEC wild-type (EHEC), and the isogenic EHEC with *rfaD* deletion (*ΔrfaD)* for 2 h. **g** The quantification results of EHEC-induced microvillar effacement in Caco-2 cell. EHEC significantly decreased the fraction of microvilli compared to mock control, and cell treated with *ΔrfaD* showed similar fraction of microvilli compared to the mock control. All experiment with ROI = 3 of each 425 μm² region. *$P < 0.05$, **$P < 0.01$ compared to the EHEC wild-type group. **h** The representative SEM images of the intestinal Caco-2 cell treated with mock control, EHEC wild-type (EHEC), and EHEC and the CDK-1 inhibitor RO3306 (10 nM). **i** The quantification results of EHEC-induced microvillar effacement in Caco-2 cells. RO3306 significantly reversed the microvillar effacement induced by EHEC. All experiment with ROI = 3 of each 425 μm² region. ***$P < 0.001$ compared to the EHEC wild-type group. **j** The representative SEM images of intestinal Caco-2 cell treated mock or EHEC with Scramble RNAi control or the *shDIAPH1*. **k** The quantification results of EHEC-induced microvillar effacement in Caco-2 cells. *shDIAPH1* significantly abolished the microvillar effacement induced by EHEC. All experiment with ROI = 4 of each 425 μm² region. ***$P < 0.001$ compared to the RNAi mock group. Arrows indicate bacterial cells, and scale bars represent 5 μm in panels (**f**, **h**, and **j**). All quantitative data are presented as mean ± SEM, and each dot represented an independent result in the bar chart. All data statistics based on: *$P < 0.05$, **$P < 0.01$, and ***$P < 0.001$ by unpaired *t* test (two-tailed). n.s. Not significant. Source data are provided as a Source Data file.

---

Furthermore, an emerging theme is that a central signal axis such as the epithelial CDK-1-formin signal may be a major mediator through which A/E pathogen members exert pleiotropic pathogenic effects.

## Methods

**Bacterial and *C. elegans* strains.** The *C. elegans* strains, bacterial strains, plasmids, and primers used in this study are listed in Supplementary Tables S1, S2, S3, and S4, respectively. The EHEC mutants were generated by the lambda Red recombinase system[32]. The Bristol N2 is the WT *C. elegans* strain. The *C. elegans* strains with the *dkIs247(act-5p::mCherry::HA::act-5)* transgene were used to monitor the microvillar actin rearrangement[7]. *Caenorhabditis elegans* strains were maintained on nematode growth medium (NGM) agar plates using the standard non-pathogenic *E. coli* strain OP50 and synchronized with alkaline hypochlorite solution[33].

**Microvillar ACT-5 cellular localization.** Synchronized late L4–young adult stage *C. elegans* animals with the *mCherry::ACT-5* transgene were fed with *E. coli* OP50 or various EHEC strains for 4 days at 20 °C and the mislocalization (redistribution of the subcellular localization) of the mCherry::ACT-5 signals from the apical membrane to the cytoplasm of the intestinal cells was examined[7]. The infected animals were paralyzed by 25 nM sodium azide and images were captured by an Olympus FV1000 confocal microscope. Each animal was given an arbitrary score from 0 to 3, and 3 showed the most severe phenotype. Animals with a score of 2 or 3 were considered as positive for the mCherry::ACT-5 mislocalization phenotype, and these scores were not given in blind.

**Colocalization analyses.** The quantification of colocalization was analyzed by ImageJ. We used the Image-Color-Split Channels menu command to get a separate z stack for the two fluorescences of confocal microscope image. Then, use the Image-Color-Channels Tool menu command to add color to two channels. To analyze the quantification of colocalization, we used the Plugins-JACoP menu command and checked all the algorithms. Then, set an appropriate threshold to minimize the background and clicked the analyze button to get a value of Pearson's correlation coefficient. The interpretations for the results of Pearson's correlation coefficient were +1 for perfect correlation, 0 for no correlation, and −1 for perfect anti-correlation[34]. The quantification of colocalization was analyzed by three independent confocal images.

**Transmission electron microscopy.** First, synchronized late L4 to young adult stage N2 animals were fed with *E. coli* OP50 control or EHEC strain EDL933 for

4 days at 20 °C. Afterwards, animals were collected, treated with 100 μg/mL gentamicin for 1.5 h, the antibiotics were washed off by M9 buffer, and then they were fixed and embedded in EPON[7]. Thin sections were cut and collected on copper grids. Sections were post stained with uranyl acetate and lead citrate and viewed using a Hitachi transmission electron microscope (H-7650, Tokyo, Japan).

**RNA interference.** *Escherichia coli* strains HT115 transformed with RNAi plasmids were spread on NG-IC plates, which are NGM plates with 50 μg/mL carbenicillin and 1 mM isopropyl-β-D-1-thiogalactopyranoside and incubated at 25 °C overnight to induce the double-strand RNA expression. Synchronized *C. elegans* L1 larva was cultured on bacteria strain HT115 containing L4440 empty vector and RNAi clone expressing double-strand RNA of the gene of interest at 20 °C until L1 larva had grown to L4–young adult stage. All RNAi assays were performed on EHEC strain EDL933 mixed with RNAi bacteria with 1:1 ratio and were spread on NG-IC plates (30 μL per plate) at 20 °C. The *C. elegans* RNAi feeding clones were from the Ahringer[35] or Vidal[36] RNAi libraries.

**Quantitative real-time PCR.** Synchronized populations of N2 WT animals were grown to late L4–young adult stage and fed with *E. coli* OP50, EHEC strain EDL933 for 4 days at 20 °C, respectively. Total RNA was then extracted using RNAzol (Sigma, R4533), and reverse transcribed with M-MLV reverse transcriptase (Promega) using random hexamer primers. The complementary DNA (cDNA) was then subjected to qRT-PCR analysis using SYBR green detection (Applied Biosystems, 4368708) on a 7500 Fast Real-Time PCR System (Applied Biosystems). The experiment was performed using three independent sets of cDNA. The relative expression between samples was normalized to the *gpd-3* as the reference gene[7].

**Specific CDK-1 inhibitor treatment in vivo.** CDK-1 inhibitor, RO3306 (Abcam, ab141491), was added into a 3.5 cm NGM plate with a final concentration of 1 μM. Synchronized *mC::ACT-5* L1 larva was cultured on the plate containing 1 μM RO3306 with food source *E. coli* OP50. After *mC::ACT-5* animals were grown from L4 to the young adult stage, animals were then transferred to OP50, and EDL933 plates containing 1 μM RO3306 for 4 days. The percentage of animals with ACT-5 mislocalization was visualized and calculated.

**Fluorescence recovery after photobleaching.** The *mC::ACT-5* L4 larva were transferred to OP50-GFP and EDL933-GFP mixed with RNAi bacteria in a 1:1 ratio on NG-IC plates (30 μL per plate) at 20 °C. After 4 days of EDL933 treatment, *mC::ACT-5* animals were immobilized with 1 mM levamisole (Merck, 359302.1566) and fixed on 1% agarose gel pad. Photobleaching and image gain were performed on the FV1000 Confocal Imaging System (Olympus) by using the

×100 objective. The 405-nm laser was used to bleach the selective region of mC:: ACT-5 signal, with mCherry fluorescence detected by the 570 nm excitation laser.

**EHEC EDL933 killing assay**. *Escherichia coli* was cultured in Luria-Bertani (LB) broth overnight at 37 °C. Thirty to fifty synchronized late L4–young adult stage *C. elegans* animals were transferred to each NGM agar plate, which was spread with the 30 μL bacterial overnight cultures and kept at 20 °C. Animals were monitored daily and transferred to fresh plates daily until no more progeny were produced. Animals that did not respond to gentle prodding by platinum wire and displayed no pharyngeal pumping were scored as dead. Survival was monitored over time until all animals had died[7]. The experiment was performed independently three times with ~100 animals per *E. coli* strain each time. Survival analysis was performed using Prism 6.0 (GraphPad Software, La Jolla, CA). The Mantel–Cox log-rank test was used to assess the statistical significance of the difference in survival, and *P* values <0.05 were considered significant.

**EHEC infection in human cells**. In preparation for infection, individual EHEC strain EDL933 colonies from fresh plates were grown in 2 mL of LB for up to 8 h. Ten microliters of bacterial culture was transferred to Dulbecco's modified Eagle's medium (DMEM) overnight at 37 °C with 5% $CO_2$ to induce the expression of EHEC T3SS. Before infection, bacteria were resuspended in DMEM to the following multiplicity of infection (MOI): EHEC EDL933 and Caco-2 cells at 500:1 and infected for 0.5 h. After infection, the plates were washed by phosphate-buffered saline (PBS) twice before image acquisition.

**Specific CDK-1 and pan-DIAPH inhibitors treatment in vitro**. Caco-2 cells were seeded in 6-well plates. When the cells reached 70–80% confluence, the specific CDK-1 inhibitor RO3306 (Abcam, ab141491) or the pan-DIAPH inhibitor SMIFH2 (Sigma-Aldrich, 340316-62-3) was added into the well with final concentrations of 10 nM or 50 μM, respectively. After EHEC infection for 2 h, the EHEC-induced microvillar effacement was observed by SEM.

**Scanning electron microscopy**. A total of $1.5 \times 10^5$ Caco-2 cells were seeded on the 6-well plate containing $22 \times 22\ mm^2$ cover glasses and grown into monolayers. After EHEC infection, cell monolayers were washed twice by PBS and fixed by 2.5% glutaraldehyde (Sigma-Aldrich, 111-30-8) solution overnight at 4 °C. After fixation, samples were washed twice by MQ water for 5 min each time. The glasses were rinsed in 1% tannin acid (Sigma-Aldrich, 1401-55-4) at room temperature for 1 h. After washing twice by MQ water, the glasses were dehydrated by using 30%, 50%, 70%, 95%, and 99% ethanol for 5, 5, 5, 10, and 10 min, respectively. Glasses were then transferred to a new well and hexamethyldisilazane (Sigma-Aldrich, 999-97-3) was added to the samples. The samples were mounted onto the pin stub mounts with carbon-coated conductive tape and coated with gold for 6 min to increase electrical conductivity. Next, the apical surface of cells was viewed using an OXFORD INCA 350 scanning electron microscope at 15 kV in the 5000× field. Every sample was randomly scanned and snapped at least three fields (regions of interest (ROI) = 3). The area of microvilli in the image was quantified by using ImageJ software (https://imagej.nih.gov/)[37]. In brief, the 5000× field image was opened. The analyzed area was enclosed by a rectangle. The threshold range that should cover the area of microvilli was adjusted according to the brightness of each image. The area of microvilli should be segmented against the background by a threshold value. The chosen area was measured and the showing result contained total area enclosed by a rectangle and interested area fraction (%Area). The interested area fraction was the area fraction of microvilli. If the image contained EHEC, the area of bacteria should be excluded. The region was outlined by the Wand tool and set an adequate threshold value. The chosen bacteria area fraction should be excluded from the interested area fraction after measurement.

**Transcriptomic analysis of the EHEC response genes**. L4–young adult stage N2 worms were fed on EHEC strain EDL933 or control OP50 plates for 6 h. Animals were washed twice by M9, then washed twice with DEPC (diethylpyrocarbonate) water, and collected and resuspended with 1 mL RNAzol (Sigma, R4533). Total RNA was extracted and cleaned-up with RNeasy Mini Kit (Qiagen, 74104). RNA-seq was performed with Illumina HiSeq 2000. The RNA-seq reads (Reads), compared the reads with the genome, and calculated the number of reads corresponding to each thousand base pairs in each million reads of each gene (Reads/kb million reads, RPKM). Comparing the expression levels of the two groups of bacteria, the expression level difference was more than two times, and the genes with false discovery rate <0.001 are regarded as genes with different expression levels. The accession number of the results uploaded in GEO is GSE142507.

**Western blot**. First, ~750 synchronized late L4 to young adult stage animals were plated on NGM plates seeded with EHEC strain EDL933 or control plates seeded with *E. coli* OP50 for 4 days at 20 °C. Polyclonal antibody to GFP (Abcam, ab6556), monoclonal antibody to SUN-1 (phospho-S43) (University of Vienna, Austria), and monoclonal antibody to α-tubulin (Sigma-Aldrich, T6199) were used for detection, Caco-2. In brief, $3 \times 10^5$ cells were seeded to a 10-cm cell culture dish. When monolayer cells became 70–80% confluent 4–6 days after the inoculum, they

were infected by EHEC strain EDL933 at MOI (500:1) for 0.5 h. Monoclonal antibody to GAPDH (Abcam, ab181602), polyclonal antibody to histone H3 (phospho-S10) (Abcam, ab47297), monoclonal antibody to DIAPH1 (Cell Signaling, 14634), polyclonal antibody to DIAPH2 (Cell Signaling, 5474), polyclonal antibody to DIAPH3 (Sigma-Aldrich, SAB1409850), and monoclonal antibody to polyclonal phospho-PP1α (Thr320) (Cell Signaling, 2581) were used for detection.

**Cell culture**. HeLa cells (ATCC, CCL-2) and Caco-2 cells (ATCC, CRL-2102) were cultured in DMEM (Life Technologies) containing high glucose and L-glutamine supplemented with 10% fetal bovine serum at 37 °C in a humidified incubator with 5% $CO_2$. A total of $3 \times 10^5$ cells were seeded to a 10-cm cell culture dish and were routinely subcultured when they reached 70–80% confluence. The culture media were changed every other day. Monolayer cells became 70–80% confluent 4–6 days after the inoculum, and the cultures were stopped after 25 passages.

**Generation of *CCNB3* stable knockdown cells**. Caco-2 cells were seeded in 6-well plates in each well at a density of $1 \times 10^5$ cells per well. After 24 h, cells were transfected with 2 μg of the shRNA expression construct (shCCNB3, pWF424) using Lipofectamine 2000 (Thermo Fisher Scientific, 11668027) according to the manufacturer's instructions. The cells were collected 48 h after transfection, replated into 10 cm Petri dishes, and then selected in growth medium containing 300 μg/ml hygromycin B for about 2 weeks. Two independent hygromycin B-resistant colonies were isolated for further characterization and analysis.

**RNAi of DIAPH1, DIAPH2, and DIAPH3**. *DIAPH1*, *DIAPH2*, and *DIAPH3* knockdown of Caco-2 cell was conducted by lentiviruses from the National RNAi Core Facility at Academia Sinica in Taiwan. On day 1, $1.5 \times 10^5$ Caco-2 cells were seeded in 6-well plates. The next day, 2 MOI lentiviruses were added to Caco-2 cells in DMEM containing 8 μg/mL polybrene and the cell was incubated at 37 °C with 5% $CO_2$. After 24 h, DMEM was substituted with a new growth medium containing 5 μg/mL puromycin. Puromycin selection requires at least 48 h. Growth medium with 5 μg/mL puromycin was replaced every few days. The infected cells were assayed until growing into the monolayer. *diaph1* was infected by lentiviruses, respectively, containing the following plasmids: TRCN0000118680 that targets CCC TATC AAG AGA TTA AGA AT, TRCN0000118681 that targets GCA TGC CCT ATC AAG AGA TTA, TRCN0000118678 that targets GC CCA GAA TCT CTC AAT CTT T, and TRCN0000303715 that targets GCC GCT GCT GGA TGG ATT AAA. *diaph2* was infected by lentiviruses, respectively, containing following plasmids: TRCN0000293811 that targets ACG TTA TTC TGG AGG TTA ATG, TRCN0000293812 that targets ATG ATG TGC GTG ACC GAA TTA, TRCN0000083889 that targets CCT ACA AAG AAG AAA GTG AAA, and TRCN0000083892 that targets GCACCCTGTCT TCACAAGAAT. *diaph3* was infected by lentiviruses, respectively, containing the following plasmids: TRCN0000154182 that targets CCT TCG GAT TTA ACC TTA GCT, TRCN0000150850 that targets GCA TGA CAA GTT TGT GAC AAA, TRCN0000150903 that targets GCT CAG TGC TAT TCT CTT TAA, and TRCN0000151280 that targets CGT GTC AGA ATA GCT AAA GAA.

**Immunofluorescence staining**. HeLa cells were seeded on the coverslips in 6-well plates with $\sim 2 \times 10^5$ cells. Following infection by EDL933-GFP, the wells were washed three times with PBS and then fixed with 3.7% paraformaldehyde at room temperature for 0.5 h. The wells were again washed with PBS twice for 2 min. The samples were quenched with 0.1 M glycine for 15 min. After washing with PBS twice, the samples were blocked and permeabilized by 1% bovine serum albumin + 0.2% Triton X-100 for 15 min. The coverslips were air-dried and overlaid with Bodipy 558/568 phalloidin (Invitrogen, B3475) for 20 min and DAPI (4′,6-diamidino-2-phenylindole) in PBS for 5 min in the dark at room temperature. The coverslips were washed in PBS and mounted on microscope slides by using Fluorescent Mounting Medium. The samples were ready to observe by immunofluorescence microscopy (Nikon Ti-U).

**Flow cytometry**. Caco-2 cells were seeded at a density of 30,000 cells/6-well plates in DMEM with 10% fetal bovine serum. On the third day of culture, cells were synchronized by double-thymidine block as described previously[38]. Cells were grown in 2 mM thymidine for 18 h in a cell incubator, and then released into fresh medium without thymidine (Sigma-Aldrich, 50-89-5) for 9 h. Next, cells were incubated in the presence of 2 mM thymidine for 12 h. After discarding the thymidine-containing medium, cells were released into the fresh DMEM medium and synchronized at the G1 phase. Before infection, cells were incubated at 37 °C for 2 h to enrich the S-phase population. When cells were synchronized at the S phase, cells were infected with EHEC EDL933 immediately with the MOI:EHEC and Caco-2 cells at 50:1 for 0.5–1 h. After infection, cells were washed twice by PBS and detached from plates by using a trypsin-EDTA solution. Cell lysates were moved to a 15 mL tube and centrifuged at $2000 \times g$ for 3 min. The supernatant was then removed and 3 mL of 70% ethanol was added to the tube. The samples were stored at −20 °C. On the day of flow cytometry analysis, the sample was centrifuged at $2000 \times g$ for 3 min and the supernatant was then removed. Mixed reagent (1 mL; 20 μL propidium iodide + 20 μL 5% Triton X-100 + 20 μL 1 mg/mL RNase A + 940 μL PBS) was added to the tube and the sample was passed through a cell filter before analysis. Finally, the cell cycle phases analysis of each sample was

conducted using a FACSCalibur flow cytometer (BD Biosciences), and the cell cycle distribution was analyzed by the FlowJo software, which fits the best Gaussian distribution curve to each peak G0/G1, G2/M, and then calculated the S phase.

**Phos-tag gel analysis.** The phosphorylated CYK-1::GFP proteins were separated by Phos-tag sodium dodecyl-sulfate polyacrylamide gel electrophoresis. The proteins were denatured with SDS sample buffer at boiling water for 10 min and then cool down at the bench before protein electrophoresis. The Phos-tag gel contains 50 μM Phos-tag AAL107 (NARD, Wako cat# 300-93523) and 1 mM $MnCl_2$. The proteins in the gel were treated with transfer buffer containing 10 mM EDTA for 10 min four times and then washed with transfer buffer for 10 min four times before transferred to PVDF by the method of wet transfer protocol at 4 °C for 1.5 h. The proteins on the PVDF were blocked with 5% milk and detected by anti-GFP antibody (Abcam, ab6556).

**Data analysis.** All experiments were performed a minimum of three times independently. For quantitative data, the $N$ of experiments performed, and the $n$ of samples (e.g., animals or ROIs) examined per $N$ are provided in the figure legend. Data are presented as mean ± SEM. All statistical analysis between two values was compared with an unpaired $t$ test (two-tailed). Survival analysis was performed using Prism 6.0 (GraphPad Software, La Jolla, CA), and the Mantel–Cox log-rank test was used to assess the statistical significance of the difference in survival. Statistically significant differences of $P < 0.05$, $P < 0.01$, and $P < 0.001$ are represented by *, **, and ***, respectively.

**Reporting summary.** Further information on research design is available in the Nature Research Reporting Summary linked to this article.

## Data availability

All the data supporting the finding of this study are available within the article and its Supplementary information files. The RNA-sequencing data can be accessed at the GEO database under the accession number GSE142507. Source data are provided with this paper.

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

## Acknowledgements

We thank the Caenorhabditis Genetics Center (CGC), which is supported by the National Institutes of Health (United States) Office of Research Infrastructure Programs (P40 OD010440), for the *C. elegans* strains. We are grateful to David Pruyne from State University of NY Upstate Medical University for the *cyk-1::GFP* plasmid. We thank Ken Sato (Gunma University, Japan) for the mCherry::ACT-5 worm. We are grateful to Verena Jantsch (University of Vienna, Austria) for the p-SUN-1[ser43] antibody. We are grateful for the assistance from the Taiwan *C. elegans* core facility (CECF), funded by the

Minister of Science and Technology (MOST 108-2319-B-002-004-) Taiwan. We also thank the National RNAi Core Facility at Academia Sinica in Taiwan for providing shRNA reagents and related services. We thank Miranda Loney for editing the manuscript. This work is supported by the Ministry of Science and Technology (MOST) grants (105-2321-B-006-011-, 106-2321-B-006-005-, 107-2628-B-006-003-, and 108-2628-B-006-005-) to CSC. The funder had no role in study design, data collection and analysis, decision to publish, or preparation of the manuscript.

## Author contributions

C.-R.H. and C.-S.C. conceived and designed the experiments. C.-R.H., C.-J.K., C.-W.H., Y.-T.C., B.-Y.L., C.-T.L., P.-L.C., Y.-W.C., T.-M.L., W.-T.C., and H.-C.H. performed the experiments. C.-R.H., C.-J.K., and C.-S.C. analyzed the data. Y.-W.C., T.-M.L., and W.-T.C. contributed reagents/materials/analysis tools. C.-R.H., C.-J.K., and C.-S.C. wrote the paper.

## Competing interests

The authors declare no competing interests.
