## [Peer Review File · Nature Communications]

Reviewers' comments:

Reviewer #1 (Remarks to the Author):

Using *C. elegans* as a model system, the authors show that enterohemorrhagic *E. coli* EHEC infection of the intestine triggers loss of microvilli by redirecting microvillar actin (particularly the ACT-5 isoform) into non-microvillar structures assembled by the formin CYK-1 and the profilin PFN-1. Additional key results shown are that effect is at least in part through increased mitotic B cyclin expression and apparent B cyclin/CDK-1-mediated phosphorylation of CYK-1. The authors implicate bacterial *rfaD* and lipopolysaccharides in inducing these effects. Finally, the authors demonstrate several aspects of this pathway are conserved in human-derived Caco-2 cells. Overall, the results of this study are quite novel, and are likely to be of interest to a number of different fields, including studies of cytoskeletal organization, cell signaling, and pathogenesis. The statistical analyses appear appropriate, although as mentioned below, the authors should provide additional information on sample sizes and the nature of data being presented. However, some additional controls need to be performed to support some of the authors' conclusions, and several speculative aspects of their conclusions could be tested relatively easily. I also have additional minor suggestions for providing better clarity in the text:

Main concerns:

(1) One key finding is that based on studies of transgenic mutants, CYK-1 behaves as though activated by phosphorylation of its DAD. However, these mutant proteins were expressed from extrachromosomal arrays, which can be highly variable for their levels of expression. Additionally, introduction of mutations in the proteins could alter protein stability. Thus, several controls are needed.

- First, to help alleviate concerns about random variability of expression among different arrays, the authors should confirm their results in at least one additional independent line for each transgene isolated.
- Second, the authors should examine whether expression levels in the intestine appear comparable based on GFP fluorescence. (Note - I am not saying the expression levels must be the same, as based on their speculation in the Discussion, they may very well be quite different. I am just saying the authors should document to what degree the expression levels are similar or different)
- Third, the authors need to perform western blots on extracts of worms bearing the different CYK-1::GFP transgenes, showing the entire blot (as opposed to the single band in Fig.S4c) to show to what degree the fusion protein is degraded into smaller GFP-containing fragments. Again, it is important to know whether any particular transgene is more or less subject to degradation, as this can factor in to conclusions of the effects of phosphorylation.

(2) The authors show in Fig.S4 that CYK-1::GFP protein levels are elevated with EHEC infection, but *cyk-1* mRNA levels are not affected. However, it is unclear whether the authors are examining protein/mRNA from endogenous *cyk-1*, from *cyk-1::gfp* driven by the *cyk-1* promoter, or *cyk-1::gfp* driven by the *app-1* promoter. This needs to be clarified. In particular, it would not be a valid comparison to examine by western blot only intestine-expressed CYK-1::GFP, but *cyk-1* mRNA from the entire animal. The ideal test would be CYK-1::GFP specifically expressed in the intestine, and mRNA for *gfp*, which should only be expressed as part of the *app-1p*-driven *cyk-1::gfp* mRNA. (In the case that the authors now observed the levels of transgene mRNA are affected by EHEC, an important follow-up question would be whether this is an effect of the exogenous promoter or 3'utr associated with the *cyk-1::gfp* transgene.)

(3) In the Discussion, the authors state (line 466), "CYB-3-CDK-1 signal upregulates the diaphanous-related formin 1 DIAPH1/CYK-1, protein level probably through the phosphorylation of CYK-1...". The authors need to dampen this conclusion, as they have not demonstrated that phosphorylation affects the level of CYK-1 protein. Alternatively, they could directly test by western blot whether the increase in CYK-1::GFP protein levels after EHEC infection (Fig.S4c) depends on *cyb-3*, or if this is an effect that is independent of B cyclin/CDK. Similarly, the

authors could determine whether expression levels of the phosphomimetic and non-phosphorylatable mutants are influenced by EHEC, and whether this is CYB-3-dependent. These are critical to teasing out how simple or complicated this pathway of regulation is, and the authors have all the tools on-hand to test these.

(4) The authors use SMIFH2 to test whether formins are involved in EHEC-induced microvillar effacement in Caco-2 cells. This is problematic for two reasons. First, SMIFH2 is a pan-formin inhibitor that does not discriminate between formin isoforms, contrary to the authors' statements that it is a DIAPH1 inhibitor (e.g. page 20, line 449; page 27, line 208; page 41, line 923). Second, SMIFH2 is known to also inhibit p53, and has also been very recently shown to inhibit a broad range of myosin isoforms when present at concentrations comparable to those used to inhibit formins (Sellers et al. 2020 Biophys J 118(3), p125a). The authors must back off their conclusions that they have demonstrated that formin activity contributes to EHEC effects in Caco-2 cells. In particular they should not state (line 473) that they have "confirmed that human cyclin B3/CCNB3, CDK1, and diaphanous related formin 1/DIAPH1 are indispensable for the EHEC-induced microvillar effacement in human intestinal Caco-2 cells". Demonstrating a role for formins would require elimination of one or more isoforms. If the authors wished to pursue results that support their suggestion that formins will be involved in Caco-2 cells, I would recommend trying simultaneous targeting of DIAPH1-3, as CYK-1 is of similar relatedness to all three isoforms, and these three formins have been shown to have varying degrees of overlapping functions in different contexts.

(5) The ability of worms to grow from L1 stage to adulthood in the presence of R03306 or with knockdown of mitotic cyclins. Can the authors comment on this? Has it been previously demonstrated that mitotic cyclin/CDK activity is dispensable for postembryonic development? Do the authors have any evidence that these treatments are effective? For example, are there expected phenotypes observed, such as endoduplications, or failure of mitoses in the M-lineage?

I also have more minor questions regarding the authors' results that don't require changes, but I hope they will consider addressing in revisions of their text:

(6) Can the authors comment on why two different B-cyclins are required for EHEC? It would seem as though the presence of just one would be sufficient, but knockdown of either cyb-3 or cyb-1 blocks redistribution of ACT-5 in response to EHEC. Can overexpression of one B cyclin (e.g. cyb-1) compensate for supporting EHEC-induced remodeling absence of the other (e.g. in cyb-3(RNAi))?

(7) In Fig.2c, cyb-3 RNAi appears to enhance microvillar association of ACT-5 even on OP50. Do the authors want to comment on whether that is the case? Would this suggest CYB-3 is normally playing a role in the intestine even in the absence of EHEC?

(8) In Fig.S7c, the authors show monitor cell cycle stage (based on DNA content) in Caco-2 cells arrested in G1, released into S phase for 2 hours, and then for 30 or 60 min, infected (or not) with EHEC. I am not an expert at studying the cell cycle, but I wonder whether these results consistent with what would be expected for an induction of mitotic B cyclin activity? Fig.S7c suggests DNA synthesis is completed more quickly with EHEC treatment, but I might have expected that S phase would prematurely terminate and DNA content would never reach G2/M. Could the authors comment on how premature induction of mitotic B cyclin activity is known to affect the progression of S phase and DNA replication?

Finally, suggestions to improve clarity of presentation:

(9) For quantitative data, the authors should provide in the figures or legends both the N of experiments performed, and the n of samples (e.g. animals, cells, ROIs) examined per N. Also, each figure should define what is shown - e.g. mean value \pm standard error, etc.

(9) Many of the images provided with the manuscript are extremely dark, even when viewed on-screen, and need to be brightened. This includes Fig. 1b, 1f, 2c, 2d, 2e, 2f, 3a, 3c, 5a, 5c, 6c, 6e, 6g, S4a, S4f, S5a, S5c, S6a, S7a. Where possible, single-color images should be rendered as black

and white for easier viewing.

(10) In Fig.S7a, it is not clear to the casual reader what is shown. It would be helpful if the authors provided outlines of the worms in the figure, and a better description of what the reader is meant to get from viewing that panel.

(11) "Expression" of a protein is typically used to describe which cells or tissues a protein is produced in, or at what level the protein is present, but not typically where a protein is localized in a cell. Where the authors refer to "expression" of ACT-5 or CYK-1, I would recommend they consider using the more precise term "subcellular localization" or "subcellular distribution".

(12) The authors should provide additional information on how the SEM quantitation was performed. For example, were specific ImageJ plug-ins utilized? How many ROI's were examined, and of what dimensions? How many cells?

(13) Fig.S4, "tubulin" is misspelled.

Reviewer #2 (Remarks to the Author):

The paper "Host CDK1-formin signal confers microvillar effacement by an attaching and effacing (A/E) pathogen" dissects a potential molecular mechanism through which Enterohemorrhagic E. coli (EHEC) causes the effacement of intestinal microvilli. This is a fascinating paper, focusing on dissecting a putative new pathway behind microvillar effacement, using a live, in vivo system with powerful genetic tools. They use ACT-5::mcherry localization as an effacement phenotype in targeted RNAi screens, using RNAseq data to guide their targets. Then, they rely on in vivo cell biological techniques and genetics to dissect the epistasis of the pathway involved in effacement. Additionally, they find two putative EHEC mutants important for inducing effacement in both C. elegans and Caco-2 cells. Finally, they used drug inhibition of CDK-1 and DIAPH1 to suggest that a similar cyclin-formin pathway is involved in EHEC effacement of human cells. Overall, the authors have systematically dissected a novel pathway downstream of EHEC infection that may play an important role in the virulence of the pathogen. And the breadth and range of experiments they have conducted is impressive. Given that the mechanisms of microvillar effacement by EHEC are currently unknown, the discovery of a cyclin/CDK and formin pathway involved in effacement opens up a new avenue to research into inhibiting EHEC infection and/or pathogenesis. However, there are several major issues that the authors should address. For example, they over-rely on a transgenic strain of C. elegans overexpressing ACT-5::mCherry as a proxy for effacement in the majority of the paper. The mislocalization of ACT-5 transgenic proteins might be an artefact and not representative of endogenous protein. Similarly, they claim multiple times that specific proteins interact in vivo using strains that are transgenically overexpressing proteins (ACT-5 and LifeAct, ACT-5 and CYK-3, etc). It is known in the C. elegans field that overexpression of proteins will inherently cause mislocalization, so their data may not be indicative of endogenous protein-protein co-localization. Furthermore, the authors heavily rely on two important assays as a proxy for effacement in C. elegans and Caco-2 cells, percent ectopic ACT-5 expression and area fraction of microvilli, respectively. Yet they do not give enough information in the manuscript on how they quantify these assays, or on the number of samples or independent replicates that were analyzed. Also, given the importance of ACT-5 filamentation by formin after EHEC infection, they should show convincing data that ACT-5 is indeed polymerized basally in the cytoplasm and that endogenous ACT-5 and formin interact in vivo (or in vitro). Also, there are many examples where the authors overinterpret their results and they should be careful that the language used in their interpretations fits with the observed data. Finally, there are several key positive and negative controls that are missing. All of these issues are expanded upon below:

Major issues:

1. It is unclear in this manuscript what the authors mean by "ectopic ACT-5 expression". Is it the loss of ACT-5 at the apical side, or is it the presence of ACT-5 punctae in the cytoplasm, or both? This is important to know especially considering the authors use this quantification in every figure. What is 100% in this assay, how many animals were calculated. How do they bin animals as having ectopic expression? What if an animal has both phenotypes or partial penetrance of a phenotype? These need to be clarified. For example, Figure 5C (T1273A) images show ACT-5 that is both apical and in granules. Was this worm counted as positive or negative for "ACT-5 ectopic expression." Without a clearcut set of rules for the most important assay in the manuscript, it can be subject to biases. Therefore, it is also important to know whether these experiments were conducted blind, and should be indicated in the methods.

2. The authors are overly dependent on a single phenotype as a proxy for effacement, the mislocalization of mCherry::ACT5 in a single transgenic worm strain. Transgenic *C. elegans* are made by inserting multiple copies of a transgene into an unknown location. Therefore, there could be an artefact from overexpression, especially considering that ACT-5 is only used apically in the microvilli. For example, the overexpression of mcherry::ACT-5 may lead to aberrant aggregates of the protein (punctae) that the authors mistake for ectopically polymerized actin, which may be exacerbated only upon EHEC infection. The authors should show the mislocalization of endogenous ACT-5 protein upon exposure to EHEC in N2, using fluorescent immunohistochemistry (there is an antibody specific to *C. elegans* ACT-5 available), and show rescue of endogenous ACT-5 localization upon cdk-1/cyb-1 knockout/knockdown.

3. It is not convincing that the mis-localized ACT-5 is filamentous actin from their LifeAct data (Fig 5f, left). First, there is no quantification of this data, just pictures. Second, the authors use a transgenic strain LifeAct::GFP strain, which is overexpressed and may be mislocalized with mcherry::ACT-5 (see above comment 2). Third, it has been recently reported that LifeAct can alter F-actin assembly in a dose-dependent manner (Flores et al., 2019, Scientific Reports). By contrast, upon *N. parisii* infection of *C. elegans*, ACT-5 is basolaterally mislocalized and clearly filamentous, as one can see filaments of ACT-5::YFP (Estes, et al, 2011 PLoS Pathogens). There needs to be stronger evidence to claim that the mislocalized ACT-5 is filamentous, especially considering the proposed pathway involves actin filamenters.

4. Similarly, since the manuscript is overly dependent on ACT-5 mislocalization as a proxy for effacement, it is unclear if cyk-1, cdk-1 or cyb-3 mutants (or knockdowns) are truly not effaced after EHEC infection. Or WT worms when deleting bacterial virulent factors rfaD or stx1. They should show at least a few EM examples of loss of effacement in these mutants.

5. For the RO3306 drug, they need a control to show the drug actually inhibiting CDK-1 kinase activity.

6. They need to show RNAi of cdk-1, cyb-3, cyk-1, and pfn-1 do not lead to increased lifespan in the absence of EHEC. These are important negative controls for Fig 3h.

7. Part of the logic in the Introduction regarding attachment and effacement is not clear, especially in regards to the order in which these processes occur during EHEC infection and their independence from each other. Does attachment precede effacement? Or are they linked? If so, couldn't the T3SS effectors that recruit actin polymerizes N-WASP and Arp2/3 also be playing a role in effacement, since actin polymerization for attachment may inherently lead to defacement of existing actin in the microvilli? This should be clarified. Similarly, the logic of the sentence in the Introduction on lines 84-87 is not clear. It is not obvious to link effacement of microvilli to actin nucleators, since disruption of other cell processes could give a similar phenotype (like disrupting membrane trafficking or cell polarity)? This logic should be clarified.

8. Results section (lines 313-331) is again over-relying on transgenic strains for interpretation of their data. The authors imply that CYK-1 interacts with ACT-5 from the colocalization of CYK::GFP with mCherry::ACT-5 in punctae (Fig S4a). First, colocalization is not indicative of interaction. Second, the colocalization might be due to transgenic overexpression and mislocalization (see comment 2 above). The authors need to show endogenous ACT-5 colocalization with endogenous

CYK-1, or conduct co-IP for evidence of interaction. Similarly, an increase in CYK-1::GFP transgenic protein is not a true proxy for an increase in endogenous CYK-1. The authors claim that CYK-1 levels increase upon EHEC infection while mRNA levels are unchanged. But they are looking at CYK-1::GFP levels (transgenic), not endogenous protein levels. Simultaneously, it is not clear whether the qPCR data is looking at *cyk-1* mRNA, *cyk-1::gfp* mRNA, or both (Fig. S4c-e). All of this data should clearly be looking at endogenous CYK-1 protein levels and endogenous mRNA levels upon EHEC infection. Finally, the sentence on line 329-331 is a clear overstatement of the data. The authors have shown no evidence that CYK-1 is sequestering ACT-5 basolaterally.

9. For Fig 5a-c, again, overexpression of transgenic proteins often leads to protein mislocalization that can be misinterpreted as colocalization. There should be some *in vitro* evidence or evidence of endogenous colocalization. Also, the authors should show quantification of the colocalization in Figures 5a and 5c.

10. Similar to comment 9, the data in Fig 6c and 6d should look at endogenous protein expression and colocalization to confirm the *rfaD* EHEC mutant diminishes colocalization, otherwise this could be a transgenic overexpression effect. Furthermore, the effect that *rfaD* is playing in effacement in the CYB-3/CYK-1 pathway would be bolstered by looking at the mutant in the *cyb-3* or *cyk-1* overexpression lines.

11. For the Caco-2 cells, it is unclear how 'area fraction of microvilli' was quantified? How many cells were examined per figure? How many individual replicates were conducted? There's no info on this in the methods or figure legends.

12. For Figure S6a-b, they should verify the knockdown. Also, this experiment seems key to their main hypothesis in human cells so it should be a main figure.

13. For Figure 6g-j, they should confirm that R0336 and SMIFH2 do not kill EHEC, leading to a misinterpretation of the results.

14. Mention of figure S7a is in the Discussion and should be in the Results. It is not clear what Fig S7a is showing or how that correlates to the number of nuclei.

Minor issues:

1. The sentence in the Abstract on lines 36-40 is a run-on sentence and pretty confusing.
2. Typo on line 57 "in chromosome"
3. Figure S1a is not informative and not needed
4. There is no strain information for the *cyb-3* overexpression transgene in the manuscript or the supplement, which prevent the reviewers from knowing how the authors are overexpressing the gene.
5. Lines 274-276 is an overstatement. The data suggests that *cyk-1* and *pfn-1* are involved in EHEC-induced ectopic expression, not required since there is still a large amount of ectopic expression.
6. In the Methods, there is no description of the how EHEC is grown or added to the lifespan assays. What's the dose? Or are they grown on lawns of EHEC? If so how do you do RNAi before lifespan? How many generations before transfer?
7. Define the y-axis in Fig 1c.
8. Should indicate the microvilli in infected samples for the TEM, since they are still there (Fig. 1F, right).
9. Figure 2f is a misleading image. A truly representative image of EHEC at 3dpi would show ACT-5 mislocalization as its above 50% in their graph
10. Figure 1f is used twice.

Reviewer #3 (Remarks to the Author):

Enteropathogenic *E. coli* (EPEC) are an important pathogen in children particularly in developing nations. These pathogens cause dysbiosis by attaching and effacing (A/E) the lining of the intestinal lumen, generating pedestal-like structures important for attachment. In this fascinating study, the nematode *C. elegans* is used as a model to discover that a principal mechanism by which A/E bacteria accomplish this is through mobilization of diaphanous-related form 1-mediated deformation of the intestinal microvilli, a mechanism which appears to be conserved with humans. The authors first show that the *C. elegans* intestinal microvilli-specific protein actin ACT-5 becomes mislocalized away from the microvilli to inside intestinal cells during EHEC infection, while other lumen markers retain their apical localization. Different EHEC strains show this effect. Strain ATC43888, specifically lacking in Shiga-like toxin, showed this effect to a lesser extent. An *in vivo* marker of filamentous actin, LifeAct::GFP, showed that the internal ACT-5 was composed in part of F-actin. TEM results support the effacement of the microvilli by the EHEC strains.

The authors used RNA-Seq to identify 97 candidate genes upregulated by EHEC that might mediate the redistribution of ACT-5. By using RNAi to knock down the expression of these individually, they found that 10 of them had a significant effect in reducing the proportion of animals with abnormal ACT-5 expression, the strongest of which was *cyb-3* which encodes a type B cyclin. CYB-3 binds to and activates CDK-1 to regulate the cell cycle. Remarkably knockdown of this one gene was enough to almost completely restore normal localization of ACT-5 during EHEC infection. Correspondingly, overexpression of CYB-3 could exacerbate the ACT-5 mislocalization suggesting a cause-effect relationship.

Knockdown or mutation of *cdk-1* also suppressed ectopic ACT-5 in EHEC treated animals consistent with a role for both CDK-1 and CYB-3. Knockdown of the *cdc-25.1/2* activators of the CDK-1 kinase activity, or treatment with the CDK-1 inhibitor R03306, also reduced ACT-5 mislocalization. Conversely, knockdown of CDK-1 inhibitors (which would enhance CDK-1 activity) resulted in earlier onset of ACT-5 mislocalization.

The authors test the effect of knockdown of genes whose products mediate actin nucleation and find that various actin modifiers (*arx-2*, *ani-1*, *cap-1*, *cap-2*), when absent, result in a stronger ACT-5 mislocalization. In particular, CYK-1 (the ortholog of human diaphanous related formin 1) and PFN-1 (actin binding profilin that accelerates actin polymerization activity of CYK-1) are both required for EHEC-induced effacement. Epistasis analysis found that these roles are downstream of CYB-3.

The authors then show that the ACT-5 mislocalization phenotype correlates with life span (susceptibility to EHEC infection).

In vivo tagged forms of ACT-5 and CYK-1 show colocalization during EHEC infection. CYK-1 protein levels are increased (though not mRNA) in response to EHEC infection suggesting that EHEC induces CYK-1 protein increases which then sequesters ACT-5 protein. This hypothesis was supported by results using FRAP assays.

Further localization studies suggest EHEC causes localization of ACT-5, CYK-1 and CDK-1. The authors also show that missense changes of a candidate phosphorylated threonine at position 1273 of CYK-1 support phosphorylation at this amino acid as being critical for colocalization with ACT-5.

In order to test for which factors in EHEC strains mediate A/E lesions in *C. elegans*, the authors test virulence determinants for the ACT-5 mislocalization phenotype. Mutations in most did not affect ACT-5 mislocalization, but loss of two genes – *stx1* (encoding Shiga-like toxin 1) had a weak effect, while loss of *rfaD* (which encodes an enzyme in lipopolysaccharide biosynthesis) had a strong effect. Following up with these, it was found that the *rfaD* mutant failed to induce *cyb-3* mRNA increases, and did not show localization of CYK-1 and ACT-5, suggesting *rfaD* and EHEC LPS are the major virulence determinants for ACT-5 mislocalization.

Moving to human Caco-2 cells, the authors find that EHEC *rfaD* mutants also fails to induce microvillar effacement compared with normal EHEC in this model. They also find that CDK-1 and Cyclin B3 are similarly required for EHEC-induced microvillar effacement. In parallel with the *C. elegans* results, increased protein levels of formin/DIAPH1 were also observed, and an inhibitor of DIAPH1 also abolished EHEC-induced microvillar effacement.

The findings are highly significant and I think would be of broad interest as they shed light on the mechanisms by which EHEC cause A/E lesions. The authors were able to use an invertebrate far removed from humans, the nematode *C. elegans*, in which to do this work. This validates the use

of *C. elegans* as a model for EHEC dysbiosis. The data within *C. elegans* (which I am most competent to judge) are standard methods for protein expression and localization and gene knockdown and have been presented well. The authors have also provided evidence that similar mechanisms may be at work in the human intestine, using Caco-2 cells. These latter experiments really need to be examined in an in vivo context to be assured of a truly conserved mechanism of A/E dysbiosis. Because of the novelty of the finding and the quality of the work, I recommend acceptance of the paper. I have no major comments (based on the *C. elegans* work) and have only minor comments.

Minor Comments:

line 468 Theronine – Threonine

one 495 – Thronine - Threonine

line 485 – "therefore producing more than 20 nuclei in 485 every *C. elegans* animal." This is a confusion of two effects that happen in the *C. elegans* gut. In the first larval stage, 14 additional nuclear divisions occur without cytokinesis, and a doubling of nuclear content accompanies each larval molt, resulting in 34 total nuclei of content 32C, in the same 20 cells that were present at hatching.

Reviewer #4 (Remarks to the Author):

Host CDK1-formin signal confers microvillar effacement by an attaching and effacing (A/E) pathogen
Huang et al.

EHEC is a pathogenic *E. coli* causing symptoms including intestinal bleeding. Although the molecular mechanisms underlying EHEC attachment were well studied, little is known about the mechanisms of intestinal microvilli effacement. Here, using *C. elegans* as a model, the authors identified host factors involved in the process, revealing that Cyclin B/CDK1-formin axis is critical for ACT-5 ectopic expression, a key process underlying intestinal microvilli effacement. They also found other factors involved in the process. Finally, they showed that the mechanisms are evolutionarily conserved from *C. elegans* to human.

The authors discovered a physiologically relevant novel signaling pathway using their original model. Their genetic evidence strongly supports the involvement of CyclinB/CDK1 and formin in this process. However, the molecular basis for Cyclin B/CDK1-formin axis is inconclusive. Thus, I recommend publishing this work in Nature Communications after addressing major issues described below. Overall, this work includes the data sets revealing a novel mechanism. This reviewer strongly encourages the authors to definitively prove their conclusion.

Major concerns:

1. The rationale for their point mutation strategy in Figure 5 is missing. The authors mutated four CDK phosphorylation consensus sites including only Thr residues, but not other S/T-P sites. Reasonable explanations should be clearly described in a text.
2. An alternative explanation should be excluded. Figure 5 could be explained by the hypothetical mechanism that CYK-1 activation (open conformation of the molecule), but not phosphorylation, is responsible for ACT-5 ectopic expression. This is because DAD domain mutations are known to affect open-close conformation in multiple formin family proteins. To exclude this possibility, another constitutively negative/active mutant CYK-1 should be expressed and their effect on ACT-5 ectopic expression should be investigated.
3. To directly show CYK-1 is phosphorylated in CDK1-dependent manner, at least one of the following should be presented: 1) mass spectrometry analysis of CYK-1 phosphorylation to show CDK consensus sites are indeed phosphorylated (including T1273), 2) CYK-1 immunoprecipitation and detection with phospho-S/T-P antibody, 3) CDK1-dependent mobility shift of CYK-1 and loss of mobility shift in T1273A, or 4) CYK-1 in vitro phosphorylation by Cyclin B/CDK1.
4. Another alternative explanation is that other molecule mediates CycB/CDK1-formin signaling axis. To exclude this possibility, direct binding between CycB/CDK1 and CYK1 should be shown. If

the authors cannot show such evidence, at least alternative explanation should be clearly described in a text.

5. To show that T1273 is important for CYK-1 protein levels, western blotting of T1273A, T1273E and control should be shown.

Minor concerns:

1. In Figure 1, there are two panel f.

2. In Figure 1b, numerical analyses of PGP-1::GFP and IFB-2::CFP should be performed, as in Figure 1c for mC::ACT-5

3. In Figure S2, the protein levels or mRNA levels of each factors should be shown to confirm knockdown.

Response to the reviewers

Note, in general we are grateful for the request for revisions by the reviewers since we feel these clarify the manuscript.

Reviewer 1:

Using *C. elegans* as a model system, the authors show that enterohemorrhagic *E. coli* EHEC infection of the intestine triggers loss of microvilli by redirecting microvillar actin (particularly the ACT-5 isoform) into nonmicrovillar structures assembled by the formin CYK-1 and the profilin PFN-1. Additional key results shown are that effect is at least in part through increased mitotic B cyclin expression and apparent B cyclin/CDK-1-mediated phosphorylation of CYK-1. The authors implicate bacterial *rfaD* and lipopolysaccharides in inducing these effects. Finally, the authors demonstrate several aspects of this pathway are conserved in human-derived Caco-2 cells. Overall, the results of this study are quite novel, and are likely to be of interest to a number of different fields, including studies of cytoskeletal organization, cell signaling, and pathogenesis. The statistical analyses appear appropriate, although as mentioned below, the authors should provide additional information on sample sizes and the nature of data being presented. However, some additional controls need to be performed to support some of the authors' conclusions, and several speculative aspects of their conclusions could be tested relatively easily. I also have additional minor suggestions for providing better clarity in the text:

Response: We are grateful for the positive comments of the reviewer and the overall enthusiasm for the manuscript. We believe that the additional clarifications requested by the reviewer strengthen the paper, and we are appreciative of these suggestions.

Main concerns:

(1) One key finding is that based on studies of transgenic mutants, CYK-1 behaves as though activated by phosphorylation of its DAD. However, these mutant proteins were expressed from extrachromosomal arrays, which can be highly variable for their levels of expression. Additionally, introduction of mutations in the proteins could alter protein stability. Thus, several controls are needed.

- First, to help alleviate concerns about random variability of expression among different arrays, the authors should confirm their results in at least one additional independent line for each transgene isolated.
- Second, the authors should examine whether expression levels in the intestine appear comparable based on GFP fluorescence. (Note - I am not saying the expression levels must be the same, as based on their speculation in the Discussion, they may very well be quite different. I am just saying the authors should document to what degree the expression levels are similar or different)
- Third, the authors need to perform western blots on extracts of worms bearing the different CYK-1::GFP transgenes, showing the entire blot (as opposed to the single band in Fig.S4c) to show to

what degree the fusion protein is degraded into smaller GFP-containing fragments. Again, it is important to know whether any particular transgene is more or less subject to degradation, as this can factor in to conclusions of the effects of phosphorylation.

Response: We are grateful to have the comments and suggestions by the reviewer. First: We have created additional transgenic lines for the *cyk-1::gfp*(wt #2), *cyk-1::gfp*(T1273A #2), and *cyk-1::gfp*(T1273E #2) and analyzed them as shown in the new Fig. 6e. Third: Moreover, the expression level of CYK-1::GFP have been analyzed and quantified by western blotting (and the entire blot) as shown in the new Fig. 6f and 6g. Our results clear showed that EHEC can significantly increase the protein level of CYK-1::GFP(wt #1) and CYK-1::GFP(T1273E #1) but not CYK-1::GFP(T1273A #1). Second: Quantification of the GFP intensity of CYK-1::GFP has been analyzed and shown in Fig. 5b. The protein level of CYK-1::GFP has been analyzed by western blotting as shown in Fig. 5c and 5d.

(2) The authors show in Fig.S4 that CYK-1::GFP protein levels are elevated with EHEC infection, but *cyk-1* mRNA levels are not affected. However, it is unclear whether the authors are examining protein/mRNA from endogenous *cyk-1*, from *cyk-1::gfp* driven by the *cyk-1* promoter, or *cyk-1::gfp* driven by the *app-1* promoter. This needs to be clarified. In particular, it would not be a valid comparison to examine by western blot only intestine-expressed CYK-1::GFP, but *cyk-1* mRNA from the entire animal. The ideal test would be CYK-1::GFP specifically expressed in the intestine, and mRNA for *gfp*, which should only be expressed as part of the *app-1* driven *cyk-1::gfp* mRNA. (In the case that the authors now observed the levels of transgene mRNA are affected by EHEC, an important follow-up question would be whether this is an effect of the exogenous promoter or 3'utr associated with the *cyk-1::gfp* transgene.)

Response: We are grateful to have the comment and suggestion by the reviewer. We have re-performed q-RT-PCR analysis for the *cyk-1::gfp* mRNA. The new results are shown in new Fig. 5e. Moreover, we have revised the text as "Our result demonstrated that the protein levels of CYK-1::GFP was significantly increased in EHEC-infected *C. elegans*" on line 355-356, and "Interestingly, the mRNA level of *cyk-1::gfp* was not increased in EHEC-treated animal (Fig. 5e)." on line 357-358.

(3) In the Discussion, the authors state (line 466), "CYB-3-CDK-1 signal upregulates the diaphanous-related formin 1 DIAPH1/CYK-1, protein level probably through the phosphorylation of CYK-1...". The authors need to dampen Page 5 of 11 this conclusion, as they have not demonstrated that phosphorylation affects the level of CYK-1 protein. Alternatively, they could directly test by western blot whether the increase in CYK-1::GFP protein levels after EHEC infection (Fig.S4c) depends on *cyb-3*, or if this is an effect that is independent of B cyclin/CDK. Similarly, the authors could determine whether expression levels of the phosphomimetic and non-phosphorylatable mutants are influenced by EHEC, and whether this is CYB-3-dependent. These

are critical to teasing out how simple or complicated this pathway of regulation is, and the authors have all the tools on-hand to test these.

Response: We are grateful to have the comments and suggestions pointed out. We have performed the western blot with *cyb-3* RNAi as shown in the new Fig 5f and 5g. Our results suggested that the upregulation of CYK-1::GFP protein level is *cyb-3* dependent. Thus, we added a sentence "Moreover, we also demonstrated that this EHEC-induced upregulation of CYK-1::GFP protein level is *cyb-3* dependent (Fig. 5f and quantified in Fig. 5g)." on line 358-360. We have also shown that the up regulation of CYK-1::GFP protein level is dependent on Threonine¹²⁷³ modification. The expression levels of CYK-1::GFP variants (phosphomimetic and non-phosphorylatable mutants) were analyzed and quantified by western blot (new Fig. 6f and quantified in 6g). Our results clearly showed that EHEC can increase the protein level of CYK-1::GFP(WT) and CYK-1::GFP(T1273E) but not CYK-1::GFP(T1273A). We therefore revised the sentence in the Discussion section as "This CYB-3-CDK-1 signal upregulates the diaphanous-related formin 1, DIAPH1/CYK-1, protein level probably through phosphorylation on Threonine¹²⁷³ in the diaphanous autoregulatory domain. However, we cannot exclude an alternative explanation that the Threonine¹²⁷³ to Alanine or Glutamate mutations may affect the open-close conformation and thus protein level of CYK-1." on line 554-559.

(4) The authors use SMIFH2 to test whether formins are involved in EHEC-induced microvillar effacement in Caco-2 cells. This is problematic for two reasons. First, SMIFH2 is a pan-formin inhibitor that does not discriminate between formin isoforms, contrary to the authors' statements that it is a DIAPH1 inhibitor (e.g. page 20, line 449; page 27, line 208; page 41, line 923). Second, SMIFH2 is known to also inhibit p53, and has also been very recently shown to inhibit a broad range of myosin isoforms when present at concentrations comparable to those used to inhibit formins (Sellers et al. 2020 Biophys J 118(3), p125a). The authors must back off their conclusions that they have demonstrated that formin activity contributes to EHEC effects in Caco-2 cells. In particular they should not state (line 473) that they have "confirmed that human cyclin B3/CCNB3, CDK1, and diaphanous related formin 1/DIAPH1 are indispensable for the EHEC-induced microvillar effacement in human intestinal Caco-2 cells". Demonstrating a role for formins would require elimination of one or more isoforms. If the authors wished to pursue results that support their suggestion that formins will be involved in Caco-2 cells, I would recommend trying simultaneous targeting of DIAPH1-3, as CYK-1 is of similar relatedness to all three isoforms, and these three formins have been shown to have varying degrees of overlapping functions in different contexts.

Response: We are grateful to have the comments and suggestions by the reviewer. We have analyzed the protein and mRNA levels of *DIAPH1*, *DIAPH2*, and *DIAPH3* (Fig S6d-f). We found that only the *DIAPH1* protein level is significantly upregulated by EHEC in

human Caco-2 cells. Moreover, specifically knockdown of *DIAPH1* and *DIAPH2*, but not *DIAPH3*, by shRNA can significantly suppress the EHEC-induced microvillar effacement on Caco-2 cells (Fig. 7j and quantitative analysis in 7k, and S5i-5l). Taken together, our results demonstrated that although only *DIAPH1* protein level is subjected to be regulated by EHEC, however the *DIAPH1* and *DIAPH2* are both required for the microvillar effacement on human Caco-2 cells induced by EHEC infection. We have also moved the SMIFH2 data into the new Fig. S5g-5h, and indicated it is a pan-DIAPH inhibitor in the text. Therefore, according to these new data we have revised the sentences in the results section from line 504 to 519. We have deleted the sentence “We also reconfirmed that human cyclin B3/CCNB3, CDK1, and diaphanous related formin 1/*DIAPH1* are indispensable for the EHEC-induced microvillar effacement in human intestinal Caco-2 cells.”, and revised sentences as “We also found that human cyclin B3/CCNB3, CDK1, *DIAPH1*, and *DIAPH2* are required for the EHEC-induced microvillar effacement in human intestinal Caco-2 cells. ” on line 563-565 in the Discussion section.

(5) The ability of worms to grow from L1 stage to adulthood in the presence of R03306 or with knockdown of mitotic cyclins. Can the authors comment on this? Has it been previously demonstrated that mitotic cyclin/CDK activity is dispensable for postembryonic development? Do the authors have any evidence that these treatments are effective? For example, are there expected phenotypes observed, such as endoduplications, or failure of mitoses in the M-lineage? I also have more minor questions regarding the authors' results that don't require changes, but I hope they will consider addressing in revisions of their text:

Response: We are grateful to have this comment by the reviewer. Completely inhibit or knockout of *cyb-3* and *cdk-1* can be lethal to the *C. elegans* animal. We believe that our *cyb-3* and *cdk-1* RNAi and treatments of RO3306 are at the conditions of “partial knock-down/inhibition” of the CDK1 activity. Therefore, we can get L4 to young adult stage animals, although a little bit longer than WT/control, for experiments after these treatments. Moreover, the efficacy of the treatment of RO3306 in worms are demonstrated in the new Fig. 3h and 3i. Our results showed that EHEC can significantly induce phospho-SUN-1^{ser43} level, which is CDK-1 dependent, and RO3306 significantly inhibited the p-SUN-1^{ser43} level in worms (line 268-270).

I also have more minor questions regarding the authors' results that don't require changes, but I hope they will consider addressing in revisions of their text:

(6) Can the authors comment on why two different B-cyclins are required for EHEC? It would seem as though the presence of just one would be sufficient, but knockdown of either *cyb-3* or *cyb-1* blocks redistribution of ACT-5 in response to EHEC. Can overexpression of one B cyclin (e.g. *cyb-1*) compensate for supporting EHEC-induced remodeling absence of the other (e.g. in *cyb-*

3(RNAi))?

Response: We are grateful to have this comment pointed out. We have tested to RNAi of *cyb-1* in the *cyb-3* o/e animals. Our new results showed that *cyb-1* RNAi can suppress the EHEC-induced ACT-5 redistribution in the *cyb-3* o/e animals. We thus added the sentences in the text as” Giving that both CYB-1 and CYB can activate CDK-1, to test whether the *cyb-1* and *cyb-3* are interchangeable in the EHEC-induced ACT-5 mislocalization, we fed WT and *cyb-3* o/e animals with *cyb-1* RNAi and EHEC for 3 days. Our results showed that *cyb-1* RNAi not only can significantly suppress the EHEC-induced ACT-5 mislocalization in WT animals (in agreement with the Fig. 3b) but also inhibit the ACT-5 mislocalization in the *cyb-3* o/e animals (Fig. 3e). These data demonstrated the non-redundant roles of *cyb-1* and *cyb-3* in the EHEC-induced ACT-5 mislocalization.” in line 250-257.

(7) In Fig.2c, *cyb-3* RNAi appears to enhance microvillar association of ACT-5 even on OP50. Do the authors want to comment on whether that is the case? Would this suggest CYB-3 is normally playing a role in the intestine even in the absence of EHEC?

Response: We are grateful to have this comment pointed out. We have performed the image analysis on the mCherry intensity of these worms. Our results demonstrated that the mCherry::ACT-5 signals in the *cyb-3* knock-down animals, although showed an upward trend, are not statistically different (by *t-test*) from the L4440 control group. We, therefore, replace the images in the Fig. 2c to represent our quantification results. And, the quantification data is shown here:

(8) In Fig.S7c, the authors show monitor cell cycle stage (based on DNA content) in Caco-2 cells arrested in G1, released into S phase for 2 hours, and then for 30 or 60 min, infected (or not) with EHEC. I am not an expert at studying the cell cycle, but I wonder whether these results consistent with what would be expected for an induction of mitotic B cyclin activity? Fig.S7c suggests DNA synthesis is completed more quickly with EHEC treatment, but I might have expected that S phase would prematurely terminate and DNA content would never reach G2/M. Could the authors comment on how premature induction of mitotic B cyclin activity is known to affect the progression of S phase and DNA replication?

Response: We are grateful to have this comment pointed out. We synchronized the cell cycle population by double-thymidine block before EHEC infection, because we aimed to maximize the population of S phase cells before EHEC infection. Our data suggested that S phase is not permanently terminated and the cells can enter into the G2/M phase after the removal of thymidine from the culture media, giving that we can observe the G2/M population increased after 2 hr pulse in the control group. Moreover, the activity of CDK-1 increases in the G2/M phase, and our results clearly showed that under this condition, EHEC can significantly increase the G2/M population compared to mock control.

Finally, suggestions to improve clarity of presentation:

(9) For quantitative data, the authors should provide in the figures or legends both the N of experiments performed, and the n of samples (e.g. animals, cells, ROIs) examined per N. Also, each figure should define what is shown - e.g. mean value \pm standard error, etc.

Response: We are grateful to have the comments pointed out. We have added, n, N, and the define of error bar in the figure legend according the regulation of Nature Communications format into every figure.

(9) Many of the images provided with the manuscript are extremely dark, even when viewed on-screen, and need to be brightened. This includes Fig. 1b, 1f, 2c, 2d, 2e, 2f, 3a, 3c, 5a, 5c, 6c, 6e, 6g, S4a, S4f, S5a, S5c, S6a, S7a. Where possible, single-color images should be rendered as black and white for easier viewing.

Response: We are grateful to have this suggestion. The brightness of these figures has been adjusted.

(10) In Fig.S7a, it is not clear to the casual reader what is shown. It would be helpful if the authors provided outlines of the worms in the figure, and a better description of what the reader is meant to get from viewing that panel.

Response: We are grateful to have this suggestion. The DIC images of the worms has been added, and the figure legend has also been revised. The text shown as “The *app-1p::mCherry-HistoneH2B* transgenic animals were infected with the EHEC for four days. Each red dot represented a nucleus in the intestinal cells. Scale bar represents 100 μ m.” in the figure legend.

(11) "Expression" of a protein is typically used to describe which cells or tissues a protein is produced in, or at what level the protein is present, but not typically where a protein is localized in a

cell. Where the authors refer to "expression" of ACT-5 or CYK-1, I would recommend they consider using the more precise term "subcellular localization" or "subcellular distribution".

Response: We are grateful to have this suggestion. The word "expression" has been revised as "subcellular localization" or "subcellular distribution" accordingly in the text.

(12) The authors should provide additional information on how the SEM quantitation was performed. For example, were specific ImageJ plug-ins utilized? How many ROI's were examined, and of what dimensions? How many cells?

Response: We are grateful to have this suggestion. The detail method of SEM analysis has been revised in the Methods section. The sentence "ROI= 3 (or according to the corresponding number) of each 425 μm^2 region" has been added in the figure legend.

(13) Fig.S4, "tubulin" is misspelled.

Response: We are grateful to have this comment pointed out. The typo has been corrected.

Reviewer 2:

The paper "Host CDK1-formin signal confers microvillar effacement by an attaching and effacing (A/E) pathogen" dissects a potential molecular mechanism through which Enterohemorrhagic E. coli (EHEC) causes the effacement of intestinal microvilli. This is a fascinating paper, focusing on dissecting a putative new pathway behind microvillar effacement, using a live, in vivo system with powerful genetic tools. They use ACT-5::mcherry localization as an effacement phenotype in targeted RNAi screens, using RNAseq data to guide their targets. Then, they rely on in vivo cell biological techniques and genetics to dissect the epistasis of the pathway involved in effacement. Additionally, they find two putative EHEC mutants important for inducing effacement in both C. elegans and Caco-2 cells. Finally, they used drug inhibition of CDK-1 and DIAPH1 to suggest that a similar cyclin-formin pathway is involved in EHEC effacement of human cells.

Overall, the authors have systematically dissected a novel pathway downstream of EHEC infection that may play an important role in the virulence of the pathogen. And the breadth and range of experiments they have conducted is impressive. Given that the mechanisms of microvillar effacement by EHEC are currently unknown, the discovery of a cyclin/CDK and formin pathway involved in effacement opens up a new avenue to research into inhibiting EHEC infection and/or pathogenesis. However, there are several major issues that the authors should address. For example, they over-rely on a transgenic strain of C. elegans overexpressing ACT-5::mCherry as a proxy for effacement in the majority of the paper. The mislocalization of ACT-5 transgenic proteins might be an artefact and not representative of endogenous protein. Similarly, they claim multiple times that

specific proteins interact in vivo using strains that are transgenically overexpressing proteins (ACT-5 and LifeAct, ACT-5 and CYK-3, etc). It is known in the *C. elegans* field that overexpression of proteins will inherently cause mislocalization, so their data may not be indicative of endogenous protein-protein co-localization. Furthermore, the authors heavily rely on two important assays as a proxy for effacement in *C. elegans* and *caco-2* cells, percent ectopic ACT-5 expression and area fraction of microvilli, respectively. Yet they do not give enough information in the manuscript on how they quantify these assays, or on the number of samples or independent replicates that were analyzed. Also, given the importance of ACT-5 filamentation by formin after EHEC infection, they should show convincing data that ACT-5 is indeed polymerized basally in the cytoplasm and that endogenous ACT-5 and formin interact in vivo (or in vitro). Also, there are many examples where the authors overinterpret their results and they should be careful that the language used in their interpretations fits with the observed data. Finally, there are several key positive and negative controls that are missing. All of these issues are expanded upon below:

Response: We are grateful for the positive comments of the reviewer and the overall enthusiasm for the manuscript. We believe that the additional clarifications requested by the reviewer strengthen the paper, and we are appreciative of these suggestions.

Major issues:

1. It is unclear in this manuscript what the authors mean by "ectopic ACT-5 expression". Is it the loss of ACT-5 at the apical side, or is it the presence of ACT-5 punctae in the cytoplasm, or both? This is important to know especially considering the authors use this quantification in every figure. What is 100% in this assay, how many animals were calculated. How do they bin animals as having ectopic expression? What if an animal has both phenotypes or partial penetrance of a phenotype? These need to be clarified. For example, Figure 5C (T1273A) images show ACT-5 that is both apical and in granules. Was this worm counted as positive or negative for "ACT-5 ectopic expression." Without a clearcut set of rules for the most important assay in the manuscript, it can be subject to biases. Therefore, it is also important to know whether these experiments were conducted blind, and should be indicated in the methods.

Response: We are grateful to have the comments and suggestions pointed out by the reviewer. We have added the detailed description to score the EHEC-induced "mislocalization" or "change in subcellular localization" of mCherry::ACT-5 signals in the new Fig. 1b and quantification in 1c. Basically, after EHEC infection for 4 days animals showed various degree of the mCherry::ACT-5 signals that are mislocalized/redistributed from the apical surface into the cytosol of basolateral site. We give each animal an arbitrary score from 0 to 3, and 3 showed the most severe phenotype. We considered animal with the score of 2 or 3 are as positive for the mCherry::ACT-5 mislocalization phenotype. These experiments were not conducted in blind and have been indicated in the methods section. Moreover, as suggested by you and the reviewer 2, we have revised the

term “ectopic expression” to mislocalization”, “change in subcellular localization”, or “redistribution” in the text.

2. The authors are overly dependent on a single phenotype as a proxy for effacement, the mislocalization of mCherry::ACT5 in a single transgenic worm strain. Transgenic *C. elegans* are made by inserting multiple copies of a transgene into an unknown location. Therefore, there could be an artefact from overexpression, especially considering that ACT-5 is only used apically in the microvilli. For example, the overexpression of mcherry::ACT-5 may lead to aberrant aggregates of the protein (punctae) that the authors mistake for ectopically polymerized actin, which may be exacerbated only upon EHEC infection. The authors should show the mislocalization of endogenous ACT-5 protein upon exposure to EHEC in N2, using fluorescent immunohistochemistry (there is an antibody specific to *C. elegans* ACT-5 available), and show rescue of endogenous ACT-5 localization upon *cdk-1/cyb-1* knockout/knockdown.

Response: We are grateful to have this comment pointed out by the reviewer. We have tried several times to contact Dr. J.A. Waddle, who published the only paper that described the usage of ACT-5 antibody in MBC 2005, for the ACT-5-specific antibody. However, we did not get any response from him. After checked the website of his university, we found that he has probably retired, and thus we are not able to show the distribution of endogenous ACT-5 currently. However, in order to minimize the effects of the tagged fluorescent proteins and the position and copy number effects of the *act-5* transgenes. We have tested the EHEC-induced ACT-5 mislocalization by 2 additional independent transgenic animals, namely *GFP::ACT-5* and *YFP::ACT-5* (Fig S1a and S1b and, quantification in S1c). Moreover, the expression levels of these transgenes have also been examined (Fig S1d). Together, our results showed that EHEC can induce similar phenotype in these three independent transgenic animals with different expression levels of *act-5* transgenes, suggesting the redistribution of ACT-5 subcellular localization-induced by EHEC is not due to the effect of the tagged fluorescent proteins nor the position and copy number effects of the *act-5* transgenes in the *C. elegans* genome.

3. It is not convincing that the mis-localized ACT-5 is filamentous actin from their LifeAct data (Fig 5f, left). First, there is no quantification of this data, just pictures. Second, the authors use a transgenic strain LifeAct::GFP strain, which is overexpressed and may be mislocalized with mcherry::ACT-5 (see above comment 2). Third, it has been recently reported that LifeAct can alter F-actin assembly in a dose-dependent manner (Flores et al., 2019, Scientific Reports). By contrast, upon *N. parisii* infection of *C. elegans*, ACT-5 is basolaterally mislocalized and clearly filamentous, as one can see filaments of ACT-5::YFP (Estes, et al, 2011 PLoS Pathogens). There needs to be stronger evidence to claim that the mislocalized ACT-5 is filamentous, especially considering the proposed pathway involves actin filamenters.

Response: We are grateful to have the comments pointed out by the reviewer. First, our LifeAct experiments only aimed to examine whether these mislocalized mCherry::ACT-5s are F-actin, thus we did not quantify these data. Second, we tried to interpret our data by additional experiments to minimize the possible transgenic effects, as described above. Third, LifeAct is the gold standard currently to monitor F-actin without disrupting the structure and dynamics of F-actin assembly *in vivo*. The report (Flores et al., 2019, Scientific Reports) pointed out by the reviewer has generated discussions in the PubPeer scientific community. The major argument is that the “green fluorescent protein” they used in their report is from a different species than the “GFP” we and others used. Thus, the authors (Flores et al., 2019, Scientific Reports) have corrected the title of their paper from “Lifeact-GFP” to “Lifeact-TagGFP2”. Forth, the paper (Estes, et al, 2011 PLoS Pathogens) pointed by the reviewer stated that “upon *N. parisii* infection of *C. elegans*, ACT-5 is basolaterally mislocalized and clearly filamentous, as one can see filaments of ACT-5::YFP”. We have read the article carefully, the data from this paper only demonstrated that the ACT-5::YFP signals are filamentous (thin-thread like), they did not demonstrate that if these actin signals are F-actin (actin polymer), rather than G-actin (actin monomer). We can also observe thin-thread like signals of ACT-5 in the EHEC-infected animals, as shown in new Fig S1a. Therefore, in order to reconfirm that the mCherry::ACT-5 signals are F-actins, which are dynamic actin polymers, we use Phalloidin-GFP to stain the F-actin in the EHEC-infected animals (new Fig. S1e). Our data clearly showed that the apical and cytosolic mCherry::ACT-5 signals are colocalized with the Phalloidin-GFP signals by confocal microscopy analysis. Together, our results suggested that the mislocalized mCherry::ACT-5s are F-actin.

4. Similarly, since the manuscript is overly dependent on ACT-5 mislocalization as a proxy for effacement, it is unclear if *cyk-1*, *cdk-1* or *cyb-3* mutants (or knockdowns) are truly not effaced after EHEC infection. Or WT worms when deleting bacterial virulent factors *rfaD* or *stx1*. They should show at least a few EM examples of loss of effacement in these mutants.

Response: We are grateful to have the suggestion by the reviewer. The TEM analysis for N2 animals treated with OP50, EHEC, EHEC $\Delta rfaD$, and EHEC $\Delta stx1$; and the TEM images *rrf-3(pk1426)* animals treated with EHEC and RNAi control L4440, *cdk-1* RNAi, *cyb-3* RNAi, and *cyk-1* RNAi, has been performed and shown as new Fig. S4. Our results showed that EHEC $\Delta rfaD$ does not induce microvillar effacement in *C. elegans*, EHEC $\Delta stx1$ still can induce microvillar effacement but is significantly less severe compared to EHEC wild-type. Moreover, RNAi of *cdk-1*, *cyb-3*, and *cyk-1* can significantly suppress the EHEC-induced microvillar effacement compared to the L4440 RNAi control.

5. For the RO3306 drug, they need a control to show the drug actually inhibiting CDK-1 kinase activity.

Response: We are grateful to have the suggestion pointed out. The efficacy for the treatment of RO3306 in worms are demonstrated in the new Fig. 3h and 3i. Our results showed that EHEC can significantly induce phospho-SUN-1^{ser43} level, which is CDK-1 dependent, and RO3306 significantly inhibited the p-SUN-1^{ser43} level in worms. Moreover, the efficacy for the treatment of RO3306 in Caco-2 cells were also monitored (Fig. S5e and quantitative analysis in S5f). Our results showed that EHEC can significantly induce phospho-PP1 α ^{Thr302} level, which is a CDK-1 phosphorylation substrate, and RO3306 significantly inhibited the p-PP1 α ^{Thr302} level in Caco-2 cells.

6. They need to show RNAi of *cdk-1*, *cyb-3*, *cyk-1*, and *pfn-1* do not lead to increased lifespan in the absence of EHEC. These are important negative controls for Fig 3h.

Response: We are grateful to have the suggestion pointed out. The normal lifespans of N2 animals knock-down *cdk-1*, *cyb-3*, *cyk-1*, or *pfn-1* were similar to that of L4440 RNAi control (new Fig. S3a).

7. Part of the logic in the Introduction regarding attachment and effacement is not clear, especially in regards to the order in which these processes occur during EHEC infection and their independence from each other. Does attachment precede effacement? Or are they linked? If so, couldn't the T3SS effectors that recruit actin polymerizes N-WASP and Arp2/3 also be playing a role in effacement, since actin polymerization for attachment may inherently lead to defacement of existing actin in the microvilli? This should be clarified. Similarly, the logic of the sentence in the Introduction on lines 84-87 is not clear. It is not obvious to link effacement of microvilli to actin nucleators, since disruption of other cell processes could give a similar phenotype (like disrupting membrane trafficking or cell polarity)? This logic should be clarified.

Response: We are grateful to have the comments pointed out. A/E lesions is characterized by the effacement of microvilli “firstly” and “then following the formation of actin pedestals” for the tight attachment of bacteria. We have revised the sentences in the Introduction section on line 58-60 as “A/E lesions are characterized by the effacement of intestinal microvilli firstly and then following the rearrangement of the host actin cytoskeleton to form pedestal-like structures for the attachment of EHEC.”. The logic of the sentence in the Introduction on lines 84-87 is to link the introduction of various actin nucleators in the following sentence. We did not exclude the possibility that other cell processes could give a similar phenotype. Thus, we have dampened our wording and revised the sentence into “Given the complexity and inherent dynamic nature of microvilli, the EHEC-induced microvillar effacement may likely involve yet unidentified cellular processes, including the key actin nucleators in a similar manner as N-WASP and Arp2/3 that are employed in the pedestal formation for EHEC attachment.” on line 80-84.

8. Results section (lines 313-331) is again over-relying on transgenic strains for interpretation of their data. The authors imply that CYK-1 interacts with ACT-5 from the colocalization of CYK::GFP with mCherry::ACT-5 in punctae (Fig S4a). First, colocalization is not indicative of interaction. Second, the colocalization might be due to transgenic overexpression and mislocalization (see comment 2 above). The authors need to show endogenous ACT-5 colocalization with endogenous CYK-1, or conduct co-IP for evidence of interaction. Similarly, an increase in CYK-1::GFP transgenic protein is not a true proxy for an increase in endogenous CYK-1. The authors claim that CYK-1 levels increase upon EHEC infection while mRNA levels are unchanged. But they are looking at CYK-1::GFP levels (transgenic), not endogenous protein levels. Simultaneously, it is not clear whether the qPCR data is looking at *cyk-1* mRNA, *cyk-1::gfp* mRNA, or both (Fig. S4c-e). All of this data should clearly be looking at endogenous CYK-1 protein levels and endogenous mRNA levels upon EHEC infection. Finally, the sentence on line 329-331 is a clear overstatement of the data. The authors have shown no evidence that CYK-1 is sequestering ACT-5 basolaterally.

Response: We are grateful to have the comments pointed out. First, we have dampened our interpretation and revised the text as "The CYK-1::GFP signals were mostly colocalized with mC::ACT-5 in animals infected by EHEC, which suggests that CYK-1 might interact with ACT-5 in the basolateral cytosol to induce ACT-5 mislocalization during EHEC infection." on line 348-350. Second, for the issue about the transgene effects, we have described in the above responses. Third, we are not able to generate and acquire antibodies currently for analyzing the endogenous ACT-5 and endogenous CYK-1 in *C. elegans*. Thus, fourth, we have reanalyzed the mRNA level of *cyk-1::gfp* per your and the reviewer 1's suggestions, the new results are shown in new Fig. 5e. Moreover, we have dampened our interpretation and revised the text as "the protein levels of CYK-1::GFP was significantly increased in EHEC-infected *C. elegans*" on line 355-356, and "the mRNA level of *cyk-1::gfp* was not increased in EHEC-treated animal" on line 357-358. Finally, we have deleted the statement that "CYK-1 is sequestering ACT-5 basolaterally" and revised the text as " Together, these results suggested that EHEC can induce CYK-1::GFP protein level in the basolateral cytoplasm by a, yet unknown, post-transcriptional regulation that dependent on CYB-3/CDK-1 signal." on line 360-362.

9. For Fig 5a-c, again, overexpression of transgenic proteins often leads to protein mislocalization that can be misinterpreted as colocalization. There should be some in vitro evidence or evidence of endogenous colocalization. Also, the authors should show quantification of the colocalization in Figures 5a and 5c.

Response: We are grateful to have the comments and suggestions pointed out. First, for the transgene effects, we have described the issue in the above responses. Second, we

have dampened our interpretation and revised the text as” These data, together the above mention reports that formin can polymerize actin and CDK1 can phosphorylate formin, suggested a scenario that ACT-5, CYK-1, and CDK-1 might interact with one another during the EHEC-induced microvillar actin rearrangement *in vivo*.” On line 382-385. Third, “*in vitro*” interaction, such as analyzed by IP-WB analysis, does not support the “*in vivo*” interaction between two proteins. Thus, in order to strengthen our results, we have performed the fluorescence resonance energy transfer (FRET) analysis between the signals of CDK-1:BFB and CYK-1::GFP. Our FRET analysis results suggested that CDK-1:BFB and CYK-1::GFP may, at least in part, interact with each other in the intestinal cells of *C. elegans* after EHEC infection (Fig. 6b). Finally, our confocal image analysis only aimed to examine whether these fluorescent signals colocalized in the intestinal cells after EHEC infection, thus we did not quantify these data.

10. Similar to comment 9, the data in Fig 6c and 6d should look at endogenous protein expression and colocalization to confirm the *rfaD* EHEC mutant diminishes colocalization, otherwise this could be a transgenic overexpression effect. Furthermore, the effect that *rfaD* is playing in effacement in the CYB-3/CYK-1 pathway would be bolstered by looking at the mutant in the *cyb-3* or *cyk-1* overexpression lines.

Response: We are grateful to have the comment and suggestion pointed out. First, the issues about the transgene effect and endogenous protein analysis have been addressed as above responses. Second, we have performed the EHEC-induced mC::ACT-5 mislocalization assay in *cyb-3* o/e worms fed with EHEC wild-type and the EHEC $\Delta rfaD$ mutant, and the new results are shown in Fig 7c. The new results strengthen our interpretation that *rfaD* and the CYB-3/CDK-1 may function in the same pathway to induce mC::ACT-5 mislocalization.

11. For the Caco-2 cells, it is unclear how 'area fraction of microvilli' was quantified? How many cells were examined per figure? How many individual replicates were conducted? There's no info on this in the methods or figure legends.

Response: We are grateful to have the suggestions pointed out. The detail method of SEM analysis has been revised in the Methods section. Moreover, the sentence “ROI= 3 (or according to the corresponding number) of each 425 μm^2 region” has been added in the figure legend.

12. For Figure S6a-b, they should verify the knockdown. Also, this experiment seems key to their main hypothesis in human cells so it should be a main figure.

Response: We are grateful to have the suggestion pointed out. The knockdown efficacies

of each RNAi in the Figs. S2, S6c, and S6j have been confirmed by the qRT-PCR analysis for the corresponding mRNA.

13. For Figure 6g-j, they should confirm that R0336 and SMIFH2 do not kill EHEC, leading to a misinterpretation of the results.

Response: We are grateful to have the suggestion pointed out. Our growth analysis showed that neither RO3306 nor SMIFH2 can inhibit the growth of EHEC cells (new Fig S6i).

14. Mention of figure S7a is in the Discussion and should be in the Results. It is not clear what Fig S7a is showing or how that correlates to the number of nuclei.

Response: We are grateful to have the suggestions pointed out. First, we have moved the mention of Fig. S7 into the Results section on line 520-540. Second, the DIC images of the worms has been added into Fig S7a, and the figure legend has also been revised. The text shown as "The *app-1p::mCherry-HistoneH2B* transgenic animals were infected with the EHEC for four days. Each red dot represented a nucleus in the intestinal cells. Scale bar represents 100 μm ." in the figure legend.

Minor issues:

1. The sentence in the Abstract on lines 36-40 is a run-on sentence and pretty confusing.

Response: We are grateful to have this comment pointed out. We have deleted this sentence from the Abstract section.

2. Typo on line 57 "in chromosome"

Response: We are grateful to have this comment pointed out. The typo has been corrected.

3. Figure S1a is not informative and not needed

Response: We are grateful to have this suggestion. The Fig. S1a has been removed from the revised manuscript.

4. There is no strain information for the *cyb-3* overexpression transgene in the manuscript or the supplement, which prevent the reviewers from knowing how the authors are overexpressing the gene.

Response: We are grateful to have the comment and suggestion pointed out. The information about the strain JNC100 carries the *cyb-3* overexpression transgene was added into the text on line 218. The reference for JNC100 has also been cited and the detail description of JNC100 can be found in the Supplemental Table S3.

5. Lines 274-276 is an overstatement. The data suggests that *cyk-1* and *pfn-1* are involved in EHEC-induced ectopic expression, not required since there is still a large amount of ectopic expression.

Response: We are grateful to have this comment and suggestion pointed out. We have dampened our interpretation and revised the text as "Together, our results reconfirmed that *cyk-1* and *pfn-1* are involved for the EHEC-induced ACT-5 mislocalization." on line 302-304.

6. In the Methods, there is no description of the how EHEC is grown or added to the lifespan assays. What's the dose? Or are they grown on lawns of EHEC? If so how do you do RNAi before lifespan? How many generations before transfer?

Response: We are grateful to have the comments pointed out. First, the descriptions about the culture of EHEC and added to the lifespan assays are on line 679-683. 30 μ l of EHEC overnight cultures was used to spread on the NGM plates. Then, the *C. elegans* animals were cultured on lawns of EHEC for the lifespan assay. Second the descriptions about RNAi are on line 625-634. *C. elegans* animals were placed on RNAi plates from L1 larva to L4-young adult stage firstly, therefore it is one generation knockdown. We then transferred these L4-young adult animals onto the plates with EHEC and RNAi bacteria in 1:1 ratio and were spread on NG-IC plates (30 μ L per plate) at 20°C.

7. Define the y-axis in Fig 1c.

Response: We are grateful to have the suggestion pointed out. The old Fig. 1c has been moved to as new Fig. 1e. The y-axis represents the % of animals with mislocalization of mC::ACT-5, PGP-1::GFP, or IFB-2::CFP in the experiment shown in new Fig.1d.

8. Should indicate the microvilli in infected samples for the TEM, since they are still there (Fig. 1F, right).

Response: We are grateful to have the suggestions pointed out. The intestinal microvilli of EHEC-infected worms were much shorter as indicated by the hash mark (#) in the new Fig 1i. Moreover, the effacements of microvilli in the EHEC-treated animals are indicated by the asterisks (*).

9. Figure 2f is a misleading image. A truly representative image of EHEC at 3dpi would show ACT-5 mislocalization as its above 50% in their graph

Response: We are grateful to have this comment. We believed that the images in Fig. 2f we picked and shown are representative confocal images of “wild-type (WT)” animals infected with EHEC at 20°C for “3 days”, giving that the % of animals with ACT-5 mislocalization is less than 10% as shown in the Fig. 2h. Moreover, the representative confocal images of “cyb-3 o/e” animals infected with EHEC at 20°C for “3 days” are also correct, giving that the % of animals with ACT-5 mislocalization is above 50% in the graph (Fig. 2h).

10. Figure 1f is used twice.

Response: We are grateful to have this comment pointed out. The typo has been corrected.

Reviewer 3:

Enteropathogenic *E. coli* (EPEC) are an important pathogen in children particularly in developing nations. These pathogens cause dysbiosis by attaching and effacing (A/E) the lining of the intestinal lumen, generating pedestal-like structures important for attachment. In this fascinating study, the nematode *C. elegans* is used as a model to discover that a principal mechanism by which A/E bacteria accomplish this is through mobilization of diaphanous-related form 1-mediated deformation of the intestinal microvilli, a mechanism which appears to be conserved with humans. The authors first show that the *C. elegans* intestinal microvilli-specific protein actin ACT-5 becomes mislocalized away from the microvilli to inside intestinal cells during EHEC infection, while other lumen markers retain their apical localization. Different EHEC strains show this effect. Strain ATC43888, specifically lacking in Shiga-like toxin, showed this effect to a lesser extent. An *in vivo* marker of filamentous actin, LifeAct::GFP, showed that the internal ACT-5 was composed in part of F-actin. TEM results support the effacement of the microvilli by the EHEC strains. The authors used RNA-Seq to identify 97 candidate genes upregulated by EHEC that might mediate the redistribution of ACT-5. By using RNAi to knock down the expression of these individually, they found that 10 of them had a significant effect in reducing the proportion of animals with abnormal ACT-5 expression, the strongest of which was *cyb-3* which encodes a type B cyclin. CYB-3 binds to and activates CDK-1 to regulate the cell cycle. Remarkably knockdown of this one gene was enough to almost completely restore normal localization of ACT-5 during EHEC infection. Correspondingly, overexpression of CYB-3 could exacerbate the ACT-5 mislocalization suggesting a cause-effect relationship.

Knockdown or mutation of *cdk-1* also suppressed ectopic ACT-5 in EHEC treated animals

consistent with a role for both CDK-1 and CYB-3. Knockdown of the *cdc-25.1/2* activators of the CDK-1 kinase activity, or treatment with the CDK-1 inhibitor R03306, also reduced ACT-5 mislocalization. Conversely, knockdown of CDK-1 inhibitors (which would enhance CDK-1 activity) resulted in earlier onset of ACT-5 mislocalization.

The authors test the effect of knockdown of genes whose products mediate actin nucleation and find that various actin modifiers (*arx-2*, *ani-1*, *cap-1*, *cap-2*), when absent, result in a stronger ACT-5 mislocalization. In particular, CYK-1 (the ortholog of human diaphanous related formin 1) and PFN-1 (actin binding profilin that accelerates actin polymerization activity of CYK-1) are both required for EHEC-induced effacement. Epistasis analysis found that these roles are downstream of CYB-3.

The authors then show that the ACT-5 mislocalization phenotype correlates with life span (susceptibility to EHEC infection).

In vivo tagged forms of ACT-5 and CYK-1 show colocalization during EHEC infection. CYK-1 protein levels are increased (though not mRNA) in response to EHEC infection suggesting that EHEC induces CYK-1 protein increases which then sequesters ACT-5 protein. This hypothesis was supported by results using FRAP assays.

Further localization studies suggest EHEC causes localization of ACT-5, CYK-1 and CDK-1. The authors also show that missense changes of a candidate phosphorylated threonine at position 1273 of CYK-1 support phosphorylation at this amino acid as being critical for colocalization with ACT-5.

In order to test for which factors in EHEC strains mediate A/E lesions in *C. elegans*, the authors test virulence determinants for the ACT-5 mislocalization phenotype. Mutations in most did not affect ACT-5 mislocalization, but loss of two genes – *stx1* (encoding Shiga-like toxin 1) had a weak effect, while loss of *rfaD* (which encodes an enzyme in lipopolysaccharide biosynthesis) had a strong effect. Following up with these, it was found that the *rfaD* mutant failed to induce *cyb-3* mRNA increases, and did not show localization of CYK-1 and ACT-5, suggesting *rfaD* and EHEC LPS are the major virulence determinants for ACT-5 mislocalization.

Moving to human Caco-2 cells, the authors find that EHEC *rfaD* mutants also fails to induce microvillar effacement compared with normal EHEC in this model. They also find that CDK-1 and Cyclin B3 are similarly required for EHEC-induced microvillar effacement. In parallel with the *C. elegans* results, increased protein levels of formin/DIAPH1 were also observed, and an inhibitor of DIAPH1 also abolished EHEC-induced microvillar effacement.

The findings are highly significant and I think would be of broad interest as they shed light on the mechanisms by which EHEC cause A/E lesions. The authors were able to use an invertebrate far removed from humans, the nematode *C. elegans*, in which to do this work. This validates the use of *C. elegans* as a model for EHEC dysbiosis. The data within *C. elegans* (which I am most competent to judge) are standard methods for protein expression and localization and gene knockdown and have been presented well. The authors have also provided evidence that similar mechanisms may be at work in the human intestine, using Caco-2 cells. These latter experiments really need to be

examined in an in vivo context to be assured of a truly conserved mechanism of A/E dysbiosis. Because of the novelty of the finding and the quality of the work, I recommend acceptance of the paper. I have no major comments (based on the *C. elegans* work) and have only minor comments.

Response: We are grateful for the positive comments of the reviewer and the overall enthusiasm for the manuscript. We believe that the additional clarifications requested by the reviewer strengthen the paper, and we are appreciative of these suggestions.

Minor Comments:

line 468 Therionine – Threonine

Response: We are grateful to have this comment pointed out. The typo has been corrected.

one 495 – Thrionine – Threonine

Response: We are grateful to have this comment pointed out. The typo has been corrected.

line 485 – "therefore producing more than 20 nuclei in 485 every *C. elegans* animal." This is a confusion of two effects that happen in the *C. elegans* gut. In the first larval stage, 14 additional nuclear divisions occur without cytokinesis, and a doubling of nuclear content accompanies each larval molt, resulting in 34 total nuclei of content 32C, in the same 20 cells that were present at hatching.

Response: We are grateful to have this comment pointed out. The sentence has been revised as "However, it has been reported that the intestinal cells can undergo nuclear endoreduplication, *i.e.*- in the first larval stage 14 additional nuclear divisions occur without cytokinesis, and a doubling of nuclear content accompanies each larval molt, resulting in 34 total nuclei in the same 20 intestinal cells that were present at hatching in every *C. elegans* animal." on line 522-526.

Reviewer 4:

Host CDK1-formin signal confers microvillar effacement by an attaching and effacing (A/E) pathogen Huang et al.

EHEC is a pathogenic *E.coli* causing symptoms including intestinal bleeding. Although the molecular mechanisms underlying EHEC attachment were well studied, little is known about the mechanisms of intestinal microvilli effacement. Here, using *C. elegans* as a model, the authors

identified host factors involved in the process, revealing that Cyclin B/CDK1-formin axis is critical for ACT-5 ectopic expression, a key process underlying intestinal microvilli effacement. They also found other factors involved in the process. Finally, they showed that the mechanisms are evolutionarily conserved from *C. elegans* to human.

The authors discovered a physiologically relevant novel signaling pathway using their original model. Their genetic evidence strongly supports the involvement of CyclinB/CDK1 and formin in this process. However, the molecular basis for Cyclin B/CDK1-formin axis is inconclusive. Thus, I recommend publishing this work in Nature Communications after addressing major issues described below. Overall, this work includes the data sets revealing a novel mechanism. This reviewer strongly encourages the authors to definitively prove their conclusion.

Response: We are grateful for the positive comments of the reviewer and the overall enthusiasm for the manuscript. We believe that the additional clarifications requested by the reviewer strengthen the paper, and we are appreciative of these suggestions.

Major concerns:

1. The rationale for their point mutation strategy in Figure 5 is missing. The authors mutated four CDK phosphorylation consensus sites including only Thr residues, but not other S/T-P sites. Reasonable explanations should be clearly described in a text.

Response: We are grateful to have this suggestion pointed out. We chose these sites to test because they are in the important functional domains, *i.e.*- FH2, FH3, and DAD domains, of CYK-1. The sentence has been revised as “We identified three S/T-P sites in the important functional domains of CYK-1, *i.e.*- Threonine⁶¹¹ in the FH3 domain, Threonine¹²³² in the FH2 domain, and Threonine¹²⁷³ in the diaphanous autoregulatory domain (DAD) in CYK-1, as the potential phosphorylation sites of CDK-1” on line 388-391.

2. An alternative explanation should be excluded. Figure 5 could be explained by the hypothetical mechanism that CYK-1 activation (open conformation of the molecule), but not phosphorylation, is responsible for ACT-5 ectopic expression. This is because DAD domain mutations are known to affect open-close conformation in multiple formin family proteins. To exclude this possibility, another constitutively negative/active mutant CYK-1 should be expressed and their effect on ACT-5 ectopic expression should be investigated.

Response: We are grateful to have this comment pointed out. However, we are not able to find the information about the construction of constitutively negative/active CYK-1 mutant. Thus, the limitation/alternative explanation pointed by the reviewer has been added to the Discussion section, as “However, we can not exclude an alternative explanation that the Threonine¹²⁷³ to Alanine or Glutamate mutations may affect the open-close conformation of CYK-1.” on line 556-558.

3. To directly show CYK-1 is phosphorylated in CDK1-dependent manner, at least one of the following should be presented: 1) mass spectrometry analysis of CYK-1 phosphorylation to show CDK consensus sites are indeed phosphorylated (including T1273), 2) CYK-1 immunoprecipitation and detection with phospho-S/T-P antibody, 3) CDK1-dependent mobility shift of CYK-1 and loss of mobility shift in T1273A, or 4) CYK-1 in vitro phosphorylation by Cyclin B/CDK1.

Response: We are grateful to have this comment pointed out. In order to further confirm that EHEC-induced phosphorylation of CYK-1::GFP we performed Phos-tag gel analysis of the CYK-1::GFP protein (Fig 6i). Our results demonstrated that EHEC can induce the phospho-CYK-1::GFP level, and this CYK-1::GFP phosphorylation signal is *cdk-1*-dependnet. Thus, we added the sentences as "Finally, to examine whether EHEC induces phosphorylation of CYK-1, we performed phospho-tag gel analysis of the CYK-1::GFP protein (Fig. 6i, and quantification in 6j). Our results showed that EHEC can significantly induce the level of phospho-CYK-1::GFP, and this phosphorylation signal of CYK-1::GFP induced by EHEC is *cdk-1*-dependnet." on line 423-427.

4. Another alternative explanation is that other molecule mediates CycB/CDK1-formin signaling axis. To exclude this possibility, direct binding between CycB/CDK1 and CYK1 should be shown. If the authors cannot show such evidence, at least alternative explanation should be clearly described in a text.

Response: We are grateful to have this comment pointed out. In order to further confirm that EHEC-induced the interaction between CDK-1::BFP and CYK-1::GFP *in vivo*, we performed the fluorescence resonance energy transfer (FRET) analysis (Fig. 6b). Our FRET results suggested that EHEC may induce the interaction between CDK-1::BFP and CYK-1::GFP *in vivo*. Thus we added the sentences as "In order to reconfirm that EHEC-induced the interaction between CDK-1::BFP and CYK-1::GFP *in vivo*, we performed the fluorescence resonance energy transfer (FRET) analysis (Fig. 6b). Our FRET analysis results suggested that CDK-1::BFP and CYK-1::GFP may, at least in part, interact with each other in the intestinal cells of *C. elegans in vivo* after EHEC infection. These data, together the above mention reports that formin can polymerize actin and CDK1 can phosphorylate fomin, suggested a scenario that ACT-5, CYK-1, and CDK-1 might interact with one another during the EHEC-induced microvillar actin rearrangement *in vivo*." on line 377-385.

5. To show that T1273 is important for CYK-1 protein levels, western blotting of T1273A, T1273E and control should be shown.

Response: We are grateful to have the comments and suggestions by the reviewer. The expression level of these CYK-1::GFP variants have been analyzed and quantified by western blot as shown in the new Fig. 6f and 6g. Our results clear showed that EHEC can significantly increase the protein level of CYK-1::GFP(wt #1) and CYK-1::GFP(T1273E #1) but not CYK-1::GFP(T1273A #1).

Minor concerns:

1. In Figure 1, there are two panel f.

Response: We are grateful to have this comment pointed out. The typo has been corrected in the new Fig. 1.

2. In Figure 1b, numerical analyses of PGP-1::GFP and IFB-2::CFP should be performed, as in Figure 1c for mC::ACT-5

Response: We are grateful to have this suggestion pointed out. The numerical analyses of PGP-1::GFP and IFB-2::CFP have been performed and presented in the new Fig. 1e.

3. In Figure S2, the protein levels or mRNA levels of each factors should be shown to confirm knockdown.

Response: We are grateful to have this comment and suggestion pointed out. The knockdown efficacies of each RNAi in the Fig. S2 have been confirmed by the qRT-PCR analysis for the corresponding mRNA.

REVIEWER COMMENTS

Reviewer #1 (Remarks to the Author):

The authors have addressed my concerns regarding their previous manuscript. I have only several editorial suggestions/comments to address a few minor mistakes that arose during addition of new material in this manuscript version. These all require only minor edits to either the text or figure labels:

The authors re-analyzed the effects of individual DIAPH formins, and confirmed DIAPH1 has the strongest effect on microvillar effacement. In the text (line 518) and Fig.S6 legend (line 227), they state DIAPH2 but not DIAPH3 also affect microvillar effacement. However, the figure indicates DIAPH3 has an effect but DIAPH2 does not, suggesting either the text or figure labels are incorrect. Also, line 516 callout to Fig.S6i-l has been mistyped as Fig.Si-l. Finally, Fig.S6 legend line 220 refers to SMIFH2 as a pan-DIAPH inhibitor, when it is actually believed to be a pan-formin inhibitor that affects all formins, whether or not they are DIAPH homologs.

Line 176, 178, the authors refer to "Phalloidin-GFP" regarding results in Fig.S1e. Presumably they mean "Alexa Fluor 488 Phalloidin" which they cite in the supplemental methods. This is indeed green, but it is not "GFP" (green fluorescent protein), and the authors should correct this in the text, Fig.S1e, and Fig.S1e legend.

Minor mis-spelling:

Line 250 I suggest "Giving" be replaced with "Given".

Line 251 "CYB" should read "CYB-3".

Line 427 "dependent" is mis-spelled.

Line 507 "the" has been mis-spelled "he".

Reviewer #2 (Remarks to the Author):

The paper "Host CDK1-formin signal confers microvillar effacement by an attaching and effacing (A/E) pathogen" dissects a potential molecular mechanism through which Enterohemorrhagic *E. coli* (EHEC) causes the effacement of intestinal microvilli. The authors have done a good amount of work to address the reviewer's concerns, and their main discoveries are novel and impactful. Also, they have added many positive and negative controls that were asked for as well as changed some of the language regarding their interpretations of the findings. There are two main issues, however, that remain unresolved, as outline below:

Major issues:

1. Point 3 regarding filamentous actin: the authors mentioned that they also observed thin thread-like signals in Fig. S1a (similar to the case of *N. parisii* infection), but the image looks like a background of GFP signals amongst GFP-negative but granules, rather than thin thread-like actin. Perhaps there is a better representative image to show the authors' claim. In addition, the use of phalloidin-GFP would be a good support for the authors' claim of mislocalized mC::ACT-5 are F-actin, but the signal of phalloidin-GFP looks almost like background. We would expect that phalloidin-GFP would stain clear, bright GFP lines along the intestine in the control, and the overlapping signals would be yellow, rather than overwhelmingly red as in Fig. S1e the authors showed. Overall, the authors have not convincingly shown that mislocalized ACT-5 is F-actin with these two additional pieces of data.

2. Point 9 regarding protein interactions: In vitro data, such as IP-WB analysis, in fact does not necessarily mean the 2 proteins interact in vivo, but if the 2 purified proteins do not interact in an IP-WB analysis, then it would be unlikely the 2 proteins directly interact in vivo. Furthermore, the FRET signal and efficiency the authors provide are likely due to chance because in the one image provided they show two tiny spots of putative FRET which represents less than 1-3 percent of the total GFP and BFP overlap signal (merge) in Fig. 6B. (i.e. perhaps the 2 proteins happen to be in close proximity in certain clusters and result in the FRET signal observed). This brings up the

larger point of showing single images without quantification. The authors state in the rebuttal "Finally, our confocal image analysis only aimed to examine whether these fluorescent signals colocalized in the intestinal cells after EHEC infection, thus we did not quantify these data." This misses the larger reason reviewers ask for quantification. Authors may be presenting only the best image for publication that may not be representative of the phenotype under examination, quantification is a way to quantitatively show how often a phenomenon is seen in the multiple images that are often taken. The FRET image is a perfect example. Here the authors show one image of a very small amount of FRET and claim it is evidence of CYK-1 and CDK-1 interaction in vivo and suggest that it can be extrapolated to all samples. Based on this, I stand by my request for quantification of the confocal co-localization in Fig. 5A, 6A and 6D.

Minor issues:

1. Figure S1d. It's unclear what "fold change" they're measuring.
2. Lines 119-120, should say, "was abnormal".
3. There are other minor typos in the new text that should be addressed.

Reviewer #4 (Remarks to the Author):

My concerns were addressed appropriately in the revised manuscript. Now this reviewer recommends this manuscript for publication.

Response to the reviewers

Note, we are grateful for the request of second revision by the editor and reviewers since we feel these clarify and strengthen the manuscript.

Reviewer #1 (Remarks to the Author):

The authors have addressed my concerns regarding their previous manuscript. I have only several editorial suggestions/comments to address a few minor mistakes that arose during addition of new material in this manuscript version. These all require only minor edits to either the text or figure labels:

Response: We are grateful to the positive comments and constructive suggestions for the manuscript, and the overall enthusiasm from the reviewer. We believe that the additional clarifications requested by the reviewer strengthen the paper, and we are appreciative of these suggestions.

The authors re-analyzed the effects of individual DIAPH formins, and confirmed DIAPH1 has the strongest effect on microvillar effacement. In the text (line 518) and Fig.S6 legend (line 227), they state DIAPH2 but not DIAPH3 also affect microvillar effacement. However, the figure indicates DIAPH3 has an effect but DIAPH2 does not, suggesting either the text or figure labels are incorrect.

Also, line 516 callout to Fig.S6i-l has been mistyped as Fig.Si-l. Finally, Fig.S6 legend line 220 refers to SMIFH2 as a pan-DIAPH inhibitor, when it is actually believed to be a pan-formin inhibitor that affects all formins, whether or not they are DIAPH homologs.

Response: We are grateful to have the comments and suggestions pointed out by the reviewer. (1) Our quantitative data in Fig. 7K and Fig. S6l indicated that knockdown of both DIAPH1 and DIAPH2 can totally abolish the EHEC-induced microvillar effacement in Caco-2 cells. However, EHEC can still induce significant microvillar effacement in the DIAPH3 knockdown group compared to the mock group, suggesting that the DIAPH3 did not participate in the EHEC-induced microvillar effacement phenotype. Thus, the statements in the text are correct. Nevertheless, we have revised the sentence to "Finally, specifically knockdown of DIAPH1 and DIAPH2, but not DIAPH3, by shRNA can totally abolish the EHEC-induced microvillar effacement on Caco-2 cells (line 517)" to avoid the confusion. (2) The typo has been corrected (line 519). (3) We have changed the words to "pan-formin inhibitor" in the text (line 514).

Line 176, 178, the authors refer to "Phalloidin-GFP" regarding results in Fig.S1e. Presumably they mean "Alexa Fluor 488 Phalloidin" which they site in the supplemental methods. This is indeed green, but it is not "GFP" (green fluorescent protein), and the authors should correct this in the text,

Fig.S1e, and Fig.S1e legend.

Response: We are grateful to have the comments and suggestions pointed out by the reviewer. We have corrected the words "Phalloidin-GFP" to "Alexa Fluor 488 Phalloidin" in text and Fig. S1.

Minor mis-spelling:

Line 250 I suggest "Giving" be replaced with "Given".

Line 251 "CYB" should read "CYB-3".

Line 427 "dependent" is mis-spelled.

Line 507 "the" has been mis-spelled "he".

Response: We are grateful to have the comments and suggestions pointed out by the reviewer. The typos have been corrected in the text.

Reviewer #2 (Remarks to the Author):

The paper "Host CDK1-formin signal confers microvillar effacement by an attaching and effacing (A/E) pathogen" dissects a potential molecular mechanism through which Enterohemorrhagic E. coli (EHEC) causes the effacement of intestinal microvilli. The authors have done a good amount of work to address the reviewer's concerns, and their main discoveries are novel and impactful. Also, they have added many positive and negative controls that were asked for as well as changed some of the language regarding their interpretations of the findings. There are two main issues, however, that remain unresolved, as outline below:

Response: We are grateful to the positive comments and constructive suggestions for the manuscript, and the overall enthusiasm from the reviewer. We believe that the additional clarifications requested by the reviewer strengthen the paper, and we are appreciative of these suggestions.

Major issues:

1. Point 3 regarding filamentous actin: the authors mentioned that they also observed thin thread-like signals in Fig. S1a (similar to the case of *N. parisii* infection), but the image looks like a background of GFP signals amongst GFP-negative but granules, rather than thin thread-like actin. Perhaps there is a better representative image to show the authors' claim. In addition, the use of phalloidin-GFP would be a good support for the authors' claim of mislocalized mC::ACT-5 are F-actin, but the signal of phalloidin-GFP looks almost like background. We would expect that phalloidin-GFP would stain clear, bright GFP lines along the intestine in the control, and the overlapping signals would be yellow, rather than overwhelmingly red as in Fig. S1e the authors

showed. Overall, the authors have not convincingly shown that mislocalized ACT-5 is F-actin with these two additional pieces of data.

Response: We are grateful to have the comments pointed out by the reviewer. Our data did demonstrate that most of the mislocalized mC::ACT-5s colocalize with Phalloidin and LifeAct signals, however we can not exclude the possibility that some of these mislocalized ACT-5 are protein aggregates as pointed out by the reviewer previously. Thus, we have dampened our conclusions and provide possible alternative explanations in the main text, as “However, we can not exclude the possibility that some of the mislocalized ACT-5 are simply mC:ACT-5 protein aggregates.” in line 180-181.

2. Point 9 regarding protein interactions: In vitro data, such as IP-WB analysis, in fact does not necessarily mean the 2 proteins interact in vivo, but if the 2 purified proteins do not interact in an IP-WB analysis, then it would be unlikely the 2 proteins directly interact in vivo. Furthermore, the FRET signal and efficiency the authors provide are likely due to chance because in the one image provided they show two tiny spots of putative FRET which represents less than 1-3 percent of the total GFP and BFP overlap signal (merge) in Fig. 6B. (i.e. perhaps the 2 proteins happen to be in close proximity in certain clusters and result in the FRET signal observed). This brings up the larger point of showing single images without quantification. The authors state in the rebuttal “Finally, our confocal image analysis only aimed to examine whether these fluorescent signals colocalized in the intestinal cells after EHEC infection, thus we did not quantify these data.” This misses the larger reason reviewers ask for quantification. Authors may be presenting only the best image for publication that may not be representative of the phenotype under examination, quantification is a way to quantitatively show how often a phenomenon is seen in the multiple images that are often taken. The FRET image is a perfect example. Here the authors show one image of a very small amount of FRET and claim it is evidence of CYK-1 and CDK-1 interaction in vivo and suggest that it can be extrapolated to all samples. Based on this, I stand by my request for quantification of the confocal co-localization in Fig. 5A, 6A and 6D.

Response: We are grateful to have the comments and suggestions by the reviewer. We have quantified the confocal colocalization in Fig 5a as shown in the new Fig 5b. And the quantifications of the confocal colocalization in Fig. 6a and 6d are also provided as the new Fig. 6b and Fig. 6f.

Minor issues:

1. Figure S1d. It's unclear what “fold change” they're measuring.
2. Lines 119-120, should say, “was abnormal”.
3. There are other minor typos in the new text that should be addressed.

Response: We are grateful to have the comments and suggestions by the reviewer. (1) It

indicated the relative mRNA expression levels of the three act-5 transgenes. We have changed it into “Relative expression levels of act-5 mRNA”. (2) The typo has been corrected. (3) The typos have been corrected.

Reviewer #4 (Remarks to the Author):

My concerns were addressed appropriately in the revised manuscript. Now this reviewer recommends this manuscript for publication.

Response: We are grateful to the positive comments and constructive suggestions for the manuscript, and the overall enthusiasm from the reviewer. We believe that the additional clarifications requested by the reviewer clarify and strengthen the paper, and we are appreciative of these suggestions.

REVIEWERS' COMMENTS

Reviewer #2 (Remarks to the Author):

Overall, the paper "Host CDK1-formin signal confers microvillar effacement by an attaching and effacing (A/E) pathogen" describes a very interesting potential molecular mechanism through which Enterohemorrhagic *E. coli* (EHEC) causes the effacement of intestinal microvilli. I commend the authors on the amount of work to address the reviewer's concerns, and their novel and impactful discoveries. However, there are still two main points that the authors have failed to convincingly show, and these are restated from the prior review:

1. The single image from Fig. S1e is being over-interpreted by the authors. In the one image provided, phalloidin-GFP is in ~90% of the intestinal tissue. They are claiming that the overlap of mCherry-ACT-5 in the intestine with some of the phalloidin-GFP is indicative of ACT-5 being f-actin. The most obvious interpretation is that this localization is completely due to chance, just because of how widespread the GFP signal. If they had another intestinal-specific mCherry-tagged protein as a control in this experiment (not-actin), then you would conclude that that this non-actin protein was co-localizing with phalloidin using the authors' interpretation criteria. This image, in my opinion, should be removed from the paper and all ensuing interpretations from it.

2. Similarly, the one image provided for FRET is being over-interpreted. Again, the authors show one image of a very small amount of FRET and claim it is evidence of CYK-1 and CDK-1 interaction *in vivo* and suggest that it can be extrapolated to all samples. The FRET signal and efficiency the authors provide are likely due to chance because in the one image provided they show two tiny spots of putative FRET which represents less than 1-3 percent of the total GFP and BFP overlap signal (merge) in Fig. 6c. (i.e. perhaps the 2 proteins happen to be in close proximity in certain clusters and result in the FRET signal observed). The authors have not shown convincing evidence of CYK-1 and CDK-1 interaction via FRET and this image should be further bolstered by more convincing evidence or removed from the manuscript.

3. Finally, the new data provided (Fig 5b, 6b and 6f) should provide information on how that samples were quantified in the methods and how many images were qualified.

Response to the reviewer

We are grateful for the constructive comments and suggestions by the reviewer 2, since we feel these clarify and strengthen the manuscript.

Reviewer #2 (Remarks to the Author):

Overall, the paper “Host CDK1-formin signal confers microvillar effacement by an attaching and effacing (A/E) pathogen” describes a very interesting potential molecular mechanism through which Enterohemorrhagic E. coli (EHEC) causes the effacement of intestinal microvilli. I commend the authors on the amount of work to address the reviewer’s concerns, and their novel and impactful discoveries. However, there are still two main points that the authors have failed to convincingly show, and these are restated from the prior review:

Response: We are grateful to the positive comments and constructive suggestions for the manuscript, and the overall enthusiasm from the reviewer. We believe that the additional clarifications requested by the reviewer strengthen the paper, and we are appreciative of these suggestions.

1. The single image from Fig. S1e is being over-interpreted by the authors. In the one image provided, phalloidin-GFP is in ~90% of the intestinal tissue. They are claiming that the overlap of mCherry-ACT-5 in the intestine with some of the phalloidin-GFP is indicative of ACT-5 being f-actin. The most obvious interpretation is that this localization is completely due to chance, just because of how widespread the GFP signal. If they had another intestinal-specific mCherry-tagged protein as a control in this experiment (not-actin), then you would conclude that that this non-actin protein was co-localizing with phalloidin using the authors interpretation criteria. This image, in my opinion, should be removed from the paper and all ensuing interpretations from it.

Response: We are grateful to have the comments pointed out by the reviewer. We have removed the Fig. S1e from the manuscript and removed or toned down the corresponding conclusions.

2. Similarly, the one image provided for FRET is being over-interpreted. Again, the authors show one image of a very small amount of FRET and claim it is evidence of CYK-1 and CDK-1 interaction in vivo and suggest that it can be extrapolated to all samples. The FRET signal and efficiency the authors provide are likely due to chance because in the one image provided they show two tiny spots of putative FRET which represents less than 1-3 percent of the total GFP and BFP overlap signal (merge) in Fig. 6c. (i.e. perhaps the 2 proteins happen to be in close proximity in certain clusters and result in the FRET signal observed). The authors have not shown convincing evidence of CYK-1 and CDK-1 interaction via FRET and this image should be further bolster by more convincing evidence or removed from the manuscript.

Response: We are grateful to have the comments pointed out by the reviewer. We have removed the FRET data from the manuscript and removed or toned down the corresponding conclusions.

3. Finally, the new data provided (Fig 5b, 6b and 6f) should provide information on how that samples were quantified in the methods and how many images were qualified.

Response: We are grateful to have the suggestions by the reviewer. We have added the detail materials and methods for the quantification of colocalization for the confocal image signals into the Method section and corresponding figure legends.